# Factors controlling the competition between *Phaeocystis* and diatoms in the Southern Ocean and implications for carbon export fluxes

Cara Nissen[1] and Meike Vogt[1]

[1]Institute for Biogeochemistry and Pollutant Dynamics, ETH Zürich, Universitätstrasse 16, 8092 Zürich, Switzerland

**Correspondence:** C. Nissen (cara.nissen@usys.ethz.ch)

**Abstract.** The high-latitude Southern Ocean phytoplankton community is shaped by the competition between *Phaeocystis* and silicifying diatoms, with the relative abundance of these two groups controlling primary and export production, the production of dimethylsulfide, the ratio of silicic acid and nitrate available in the water column, and the structure of the food web. Here, we investigate this competition using a regional physical-biogeochemical-ecological model (ROMS-BEC) configured at eddy-permitting resolution for the Southern Ocean south of 35° S. We improved ROMS-BEC by adding an explicit parameterization of *Phaeocystis* colonies, so that the model, together with the previous addition of an explicit coccolithophore type, now includes all biogeochemically relevant Southern Ocean phytoplankton types. We find that *Phaeocystis* contribute 46±21% (1$\sigma$ in space) and 40±20% to annual NPP and POC export south of 60° S, respectively, making them an important contributor to high-latitude carbon cycling. In our simulation, the relative importance of *Phaeocystis* and diatoms is mainly controlled by spatio-temporal variability in temperature and iron availability. In addition, in more coastal areas, such as the Ross Sea, the higher light sensitivity of *Phaeocystis* at low irradiances promotes the succession from *Phaeocystis* to diatoms. Differences in the biomass loss rates, such as aggregation or grazing by zooplankton, need to be considered to explain the simulated seasonal biomass evolution and carbon export fluxes.

## 1 Introduction

Phytoplankton production in the Southern Ocean (SO) regulates not only the uptake of anthropogenic carbon in marine food-webs, but also controls global primary production via the lateral export of nutrients to lower latitudes (e.g. Sarmiento et al., 2004; Palter et al., 2010). The amount and stoichiometry of these laterally exported nutrients is determined by the combined action of multiple types of phytoplankton with differing ecological niches and nutrient requirements. Yet, despite their important role, the drivers of phytoplankton biogeography and competition and the relative contribution of different phytoplankton groups to SO carbon cycling are still poorly quantified. Today, the SO phytoplankton community is largely dominated by silicifying diatoms that efficiently fix and transport carbon from the surface ocean to depth (e.g. Swan et al., 2016) and have been suggested to be the major contributor to SO carbon export (Buesseler, 1998; Smetacek et al., 2012). However, calcifying coccolithophores and dimethylsulfide (DMS) producing *Phaeocystis* have been found to contribute in a significant way to total phytoplankton biomass in summer/fall at subantarctic (Balch et al., 2016; Nissen et al., 2018) and in spring/summer at high

latitudes, respectively (Smith and Gordon, 1997; Arrigo et al., 1999; DiTullio et al., 2000; Poulton et al., 2007; Arrigo et al., 2017), thus suggesting that the succession and competition of different plankton groups governs biogeochemical cycles at the (sub)regional scale. As climate change is expected to differentially impact the competitive fitness of different phytoplankton groups and ultimately their contribution to total net primary production (NPP; IPCC, 2014; Constable et al., 2014; Deppeler and Davidson, 2017), with a likely increase in the relative importance of coccolithophores and *Phaeocystis* in a warming world at the expense of diatoms (Bopp et al., 2005; Winter et al., 2013; Rivero-Calle et al., 2015), the resulting change in SO phytoplankton community structure is likely to affect global nutrient and carbon distributions, ocean carbon uptake, and marine food web structure (Smetacek et al., 2004). While a number of recent studies have elucidated the importance of coccolithophores for subantarctic carbon cycling (e.g. Rosengard et al., 2015; Balch et al., 2016; Nissen et al., 2018; Rigual Hernández et al., 2020), few estimates quantify the role of present and future high-latitude SO phytoplankton community structure for ecosystem services such as NPP and carbon export (e.g. Wang and Moore, 2011; Yager et al., 2016).

*Phaeocystis* blooms in the SO have been regularly observed in early spring at high SO latitudes (especially in the Ross Sea, see e.g. Smith et al., 2011), thus preceding those of diatoms (Green and Sambrotto, 2006; Peloquin and Smith, 2007; Alvain et al., 2008; Arrigo et al., 2017; Ryan-Keogh et al., 2017), and *Phaeocystis* can dominate over diatoms in terms of carbon biomass at regional and sub-annual scales (e.g. Smith and Gordon, 1997; Alvain et al., 2008; Leblanc et al., 2012; Vogt et al., 2012; Ben Mustapha et al., 2014). Nevertheless, *Phaeocystis* is not routinely included as a phytoplankton functional type (PFT) in global biogeochemical models, possibly a result of the limited number of biomass validation data (Vogt et al., 2012) and its complex life cycle (Schoemann et al., 2005). In particular, *Phaeocystis* is difficult to model because traits linked to biogeochemistry-related ecosystem services, such as size and carbon content, vary due to its complex multi-stage life cycle. Its alternation between solitary cells of a few $\mu$m in diameter and gelatinous colonies of several mm to cm in diameter (e.g. Rousseau et al., 1994; Peperzak, 2000; Chen et al., 2002; Bender et al., 2018) directly impacts community biomass partitioning and the relative importance of aggregation, viral lysis, and grazing for *Phaeocystis* biomass losses, its susceptibility to zooplankton grazing relative to that of diatoms (Granéli et al., 1993; Smith et al., 2003), and ultimately the export of particulate organic carbon (POC; Schoemann et al., 2005). With *Phaeocystis* colonies typically dominating over solitary cells during the SO growing season (Smith et al., 2003) and with larger cells being more likely to form aggregates and less likely to be grazed by microzooplankton (Granéli et al., 1993; Caron et al., 2000; Schoemann et al., 2005; Nejstgaard et al., 2007), *Phaeocystis* biomass loss via aggregation possibly increases in relative importance at the expense of grazing as more colonies are formed and colony size increases (Tang et al., 2008). Altogether, this implies a complex seasonal variability in the magnitude and pathways of carbon transfer to depth as the phytoplankton community changes throughout the year, which is expensive to comprehensively assess through in situ studies and therefore calls for marine ecosystem models.

Across those marine ecosystem models including a *Phaeocystis* PFT, the representation of its life cycle differs in terms of complexity (Pasquer et al., 2005; Tagliabue and Arrigo, 2005; Wang and Moore, 2011; Le Quéré et al., 2016; Kaufman et al., 2017; Losa et al., 2019). While some models include rather sophisticated parametrizations to describe life cycle transitions (accounting for nutrient concentrations, light levels, and a seed population, see e.g. Pasquer et al., 2005; Kaufman et al., 2017), the majority includes rather simple transition functions (accounting for iron concentrations only, see Losa et al., 2019)

or only the colonial life stage of *Phaeocystis* (Tagliabue and Arrigo, 2005; Wang and Moore, 2011; Le Quéré et al., 2016). Despite these differences, all of the models see improvements in the simulated SO phytoplankton biogeography as compared to observations upon the implementation of a *Phaeocystis* PFT. In particular, Wang and Moore (2011) find that *Phaeocystis* contributes substantially to SO integrated annual NPP and POC export (23% and 30% south of 60° S, respectively; Wang and Moore, 2011), implying that models not accounting for *Phaeocystis* possibly overestimate the role of diatoms for high-latitude phytoplankton biomass, NPP, and POC export (Laufkötter et al., 2016). Overall, the link between ecosystem composition, ecosystem function, and global biogeochemical cycling in general (e.g. Siegel et al., 2014; Guidi et al., 2016; Henson et al., 2019) and the contribution of *Phaeocystis* to SO export of POC in particular are still under debate. While some have found blooms of *Phaeocystis* to be important vectors of carbon transfer to depth through the formation of aggregates (Asper and Smith, 1999; DiTullio et al., 2000; Ducklow et al., 2015; Yager et al., 2016; Asper and Smith, 2019), others suggest their biomass losses to be efficiently retained in the upper ocean by local circulation (Lee et al., 2017) and degraded in the upper water column through bacterial and zooplankton activity (Gowing et al., 2001; Accornero et al., 2003; Reigstad and Wassmann, 2007; Yang et al., 2016), making *Phaeocystis* a minor contributor to SO POC export. This demonstrates the major existing uncertainty in how the high-latitude phytoplankton community structure impacts carbon export fluxes.

In general, the relative importance of different phytoplankton types for total phytoplankton biomass is controlled by a combination of top-down factors, i.e. processes impacting phytoplankton biomass loss such as grazing by zooplankton, aggregation of cells and subsequent sinking, or viral lysis, and bottom-up factors, i.e. physical and biogeochemical variables impacting phytoplankton growth (Le Quéré et al., 2016). The observed spatio-temporal differences in the relative importance of *Phaeocystis* and diatoms in the SO are thought to be largely controlled by differences in light and iron levels, but the relative importance of the different bottom-up factors appears to vary depending on the time and location of the sampling (Arrigo et al., 1998, 1999; Goffart et al., 2000; Sedwick et al., 2000; Garcia et al., 2009; Tang et al., 2009; Mills et al., 2010; Feng et al., 2010; Smith et al., 2011, 2014). Concurrently, while available models agree with the observations on the general importance of light and iron levels, differences in the dominant bottom-up factors controlling the distribution of *Phaeocystis* at high SO latitudes across models are possibly a result of differences in how this phytoplankton type is parametrized (Tagliabue and Arrigo, 2005; Pasquer et al., 2005; Wang and Moore, 2011; Le Quéré et al., 2016; Kaufman et al., 2017; Losa et al., 2019). In this context, whether the model explicitly represents both *Phaeocystis* life stages (Pasquer et al., 2005; Kaufman et al., 2017; Losa et al., 2019) or only the colonial stage (Wang and Moore, 2011; Le Quéré et al., 2016) is key, as single cells are known to have lower iron requirements than *Phaeocystis* colonies (Veldhuis et al., 1991). Besides bottom-up factors, some observational studies suggest that top-down factors are important in controlling the relative importance of *Phaeocystis* and diatoms as well. For instance, van Hilst and Smith (2002) suggest grazing by zooplankton to be an important factor explaining the observed distributions of these two phytoplankton types in the SO, likely resulting from the generally lower grazing pressure on *Phaeocystis* colonies than on diatoms (Granéli et al., 1993; Smith et al., 2003). Yet, further evidence suggests a role for other biomass loss processes such as aggregation and subsequent sinking (Asper and Smith, 1999; Ducklow et al., 2015; Asper and Smith, 2019). Altogether, this calls for a comprehensive quantitative analysis of the relative importance of bottom-up and top-down factors in controlling

the competition between *Phaeocystis* and diatoms over the course of the SO growing season and its ramifications for carbon transfer to depth.

In this study, we investigate the competition between *Phaeocystis* and diatoms and its implications for carbon cycling using a regional coupled physical-biogeochemical-ecological model configured at eddy-permitting resolution for the SO (ROMS-BEC, Nissen et al., 2018). To address the missing link between SO phytoplankton biogeography, ecosystem function, and the global carbon cycle, we have added *Phaeocystis* colonies as an additional PFT to the model, so that it includes all known biogeochemically relevant phytoplankton types of the SO (e.g. Buesseler, 1998; DiTullio et al., 2000). Using available observations, such as satellite-derived chlorophyll concentrations, carbon biomass and pigment data, we first validate the simulated phytoplankton distributions and community structure across the SO and then particularly focus on the temporal variability of diatoms and *Phaeocystis* in the high-latitude SO. After assessing the relative importance of bottom-up and top-down factors in controlling the contribution of *Phaeocystis* colonies and diatoms to total phytoplankton biomass over a complete annual cycle in the high-latitude SO, we show that the spatially and temporarilly varying phytoplankton community composition leaves a distinct, PFT-specific imprint on upper ocean carbon cycling and POC export across the SO.

## 2   Methods

### 2.1   ROMS-BEC with explicit *Phaeocystis* colonies

We use a quarter-degree SO setup of the Regional Ocean Modeling System ROMS (latitudinal range from 24° S-78° S, 64 topography-following vertical levels, time step to solve the primitive equations is 1600 s; Shchepetkin and McWilliams, 2005; Haumann, 2016), coupled to the biogeochemical model BEC (Moore et al., 2013), which was recently extended to include an explicit represenation of coccolithophores and thoroughly validated in the SO setup (Nissen et al., 2018). BEC resolves the biogeochemical cycling of all macronutrients (C, N, P, Si), as well as the cycling of iron (Fe), the major micronutrient in the SO. The model includes four PFTs – diatoms, coccolithophores, small phytoplankton/SP, and $N_2$-fixing diazotrophs – and one zooplankton functional type (Moore et al., 2013; Nissen et al., 2018). Here, we extend the version of Nissen et al. (2018) to include an explicit parameterization of colonial *Phaeocystis antarctica*, which is the only bloom-forming species of *Phaeocystis* occurring in the SO (Schoemann et al., 2005) and which typically dominates over solitary cells when SO *Phaeocystis* biomass levels are highest (Smith et al., 2003). For the remainder of this manuscript, we will refer to the new PFT as "*Phaeocystis*". Generally, model parameters for *Phaeocystis* in the *Baseline* setup are chosen to represent the colonial form of *Phaeocystis* whenever information is available in the literature (see e.g. review by Schoemann et al., 2005), and model parameters were tuned to maximize the model-data agreement in the spatio-temporal variability of the phytoplankton community structure between ROMS-BEC and all available observations (see also section 2.3.1). By only simulating the colonial form of *Phaeocystis*, we assume enough solitary cells of *Phaeocystis* to be available for colony formation at any time as part of the SP PFT. As for the other phytoplankton PFTs, growth by *Phaeocystis* is limited by surrounding temperature, nutrient, and light conditions as outlined in the following (see appendix B for a complete description of the model equations describing phytoplankton growth).

As the new PFT in ROMS-BEC represents a single species of *Phaeocystis*, we use an optimum function rather than an Eppley curve (Eppley, 1972) to describe its temperature-limited growth rate $\mu^{\mathrm{PA}}(T)$ (d$^{-1}$, Schoemann et al., 2005):

$$\mu^{\mathrm{PA}}(T) = \mu^{\mathrm{PA}}_{\mathrm{max}} \cdot e^{-\left(\frac{T-T_{\mathrm{opt}}}{\tau}\right)^2} \tag{1}$$

In the above equation, the maximum growth rate ($\mu^{\mathrm{PA}}_{\mathrm{max}}$) is 1.56 d$^{-1}$ at an optimum temperature ($T_{\mathrm{opt}}$) of 3.6° C and the temperature interval ($\tau$) is 17.51° C and 1.17° C at temperatures below and above 3.6° C, respectively. With these parameters, the simulated growth rate of *Phaeocystis* in ROMS-BEC is zero at temperatures above ∼8° C (in agreement with laboratory experiments with *Phaeocystis antarctica*, see Buma et al., 1991) and higher than that of diatoms for temperatures between ∼0-4° C (Fig. A1a). We acknowledge that the range of temperatures for which the growth of *Phaeocystis* exceeds that of
diatoms is possibly underestimated, as the temperature-limited growth rate by diatoms in ROMS-BEC is overestimated at low temperatures compared to available laboratory data (see Fig. A1a & Eq. B5). Yet, we note that temperature-limited growth by diatoms in the model is tuned to fit the data at the global range of temperatures, in particular for the competition with coccolithophores at subantarctic latitudes (Nissen et al., 2018).

Half-saturation constants for macronutrient limitation are scarce for *P. antarctica* (Schoemann et al., 2005), and macronutri-
ent limitation of *Phaeocystis* is therefore chosen to be identical to that of diatoms in ROMS-BEC (Table 1). As the availability of the micronutrient Fe generally limits phytoplankton growth in the high-latitude SO (Martin et al., 1990a, b) and accordingly in ROMS-BEC (Fig. S1), this choice is not expected to significantly impact the simulated competition between diatoms and *Phaeocystis* in this area. In contrast, differences in the half-saturation constants with respect to dissolved Fe concentrations (k$_{\mathrm{Fe}}$) of *Phaeocystis* and diatoms critically impact the competitive success of *Phaeocystis* relative to diatoms throughout the
145    year (see e.g. Sedwick et al., 2000, 2007). Here, due to their larger size, we assume a higher k$_{\mathrm{Fe}}$ for *Phaeocystis* (0.2 $\mu$mol m$^{-3}$) than for diatoms (0.15 $\mu$mol m$^{-3}$, Table 1). We note however, that the k$_{\mathrm{Fe}}$ of *Phaeocystis* has been reported to vary over one order magnitude depending on the ambient light level (0.045-0.45 $\mu$mol m$^{-3}$, see Fig. A1b and Garcia et al., 2009), with lowest values at optimum light levels of around 80 W m$^{-2}$. Due to the limited number (3) of reported light levels in Garcia et al. (2009) and the associated uncertainty when fitting the data, we refrain from using this k$_{\mathrm{Fe}}$-light-dependency in the *Base-*
*line* simulation, but explore the sensitivity of the simulated seasonality of *Phaeocystis* and diatom biomass to a polynomial fit describing the k$_{\mathrm{Fe}}$ of *Phaeocystis* as a function of the light intensity (see Fig. A1b and section 2.2). As a result of the tuning exercise aiming to maximize the fit of *all* simulated PFT biomass fields to available observations, the k$_{\mathrm{Fe}}$ of the other PFTs in ROMS-BEC are increased by 25% in this study as compared to in Nissen et al. (2018, see Table 1). For diatoms, this change leads to a better agreement of the k$_{\mathrm{Fe}}$ used in ROMS-BEC with values suggested for large SO diatoms by Timmermans et al.
(2004), but we acknowledge that the chosen value here is still at the lower end of their suggested range (0.19-1.14 $\mu$mol m$^{-3}$). In ROMS-BEC, phytoplankton Fe uptake relative to the uptake of C varies as a function of seawater Fe levels and decreases linearly below a critical concentration which is specific to each PFT's k$_{\mathrm{Fe}}$ (see Eq. B11). In concert with the seasonal evolution of upper ocean Fe levels, the Fe:C ratios of all PFTs are highest in winter and lowest in summer (not shown). As a result of their higher k$_{\mathrm{Fe}}$ in the model, *Phaeocystis* generally have lower Fe:C uptake ratios than diatoms. We note that we currently do
not include any luxury uptake of Fe by *Phaeocystis* into their gelatinous matrix (Schoemann et al., 2001). Serving as a storage

of additional Fe accessible to the *Phaeocystis* colony when Fe in the seawater gets low, this luxury uptake is thought to relieve it from Fe limitation when Fe concentrations become growth limiting (see discussion in Schoemann et al., 2005). We therefore probably overestimate the Fe limitation of *Phaeocystis* growth in ROMS-BEC.

*P. antarctica* blooms are typically found where and when waters are turbulent and the mixed layer is deep (in comparison to blooms dominated by diatoms, see e.g. Arrigo et al., 1999; Alvain et al., 2008), suggesting that *Phaeocystis* is better in coping with low light levels than diatoms (e.g. Arrigo et al., 1999). In agreement with laboratory experiments (Tang et al., 2009; Mills et al., 2010; Feng et al., 2010), we therefore choose a higher $\alpha_{PI}$, i.e. a higher sensitivity of growth to increases of photosynthetically active radiation (PAR) at low PAR levels, for *Phaeocystis* than for diatoms in ROMS-BEC (see Table 1). Our value (0.63 mmol C m$^2$ (mg Chl W s)$^{-1}$) corresponds to the average value compiled from available laboratory experiments (Schoemann et al., 2005).

In addition to environmental conditions directly impacting phytoplankton growth rates, loss processes such as grazing, non-grazing mortality, and aggregation impact the simulated biomass levels at any point and time (Moore et al., 2002). Grazing on *Phaeocystis* varies across zooplankton size classes, as a consequence of *Phaeocystis* life forms spanning several orders of magnitude in size (Schoemann et al., 2005). Furthermore, *Phaeocystis* colonies are surrounded by a membrane (Hamm et al., 1999), potentially serving as protection from zooplankton grazing. While small copepods have been shown to graze less on *Phaeocystis* once they form colonies, other larger zooplankton appear to continue grazing on *Phaeocystis* colonies at unchanged rates (Granéli et al., 1993; Caron et al., 2000; Schoemann et al., 2005; Nejstgaard et al., 2007). Based on a size-mismatch assumption of the single grazer in ROMS-BEC and *Phaeocystis* colonies, we assume a lower maximum grazing rate on *Phaeocystis* than on diatoms (3.6 d$^{-1}$ and 3.8 d$^{-1}$, respectively, see $\gamma_{g,max}$ in Table 1). Upon grazing, we assume the fraction of the grazed phytoplankton biomass that is transformed to sinking POC via zooplankton fecal pellet production to be higher for larger and ballasted cells than for small, unballasted cells. Consequently, the fraction of grazing routed to POC increases from grazing on SP or diazotrophs to coccolithophores, *Phaeocystis*, and diatoms ($r_g$ in Table 1). Consistent with Nissen et al. (2018), we keep a Holling Type II ingestion functional response here (Holling, 1959) and compute grazing on each prey separately (Eq. B14). We refer to Nissen et al. (2018) for a discussion of the relative merits and pitfalls for using Holling Type II versus III.

Non-grazing mortality (such as viral lysis) has been shown to increase under environmental stress for *Phaeocystis* colonies, causing colony disruption and ultimately cell death (van Boekel et al., 1992; Schoemann et al., 2005). To account for processes causing colony disintegration and for grazing by higher trophic levels not explicitly included in ROMS-BEC, *Phaeocystis* in ROMS-BEC experience a higher mortality rate than diatoms (0.18 d$^{-1}$ and 0.12 d$^{-1}$, respectively, see $\gamma_{m,0}$ in Table 1 & Eq. B16). Thereby, the chosen non-grazing mortality rate of *Phaeocystis* assumed in the model is still lower than the estimated rate of viral lysis for *Phaeocystis* in the North Sea by van Boekel et al. (1992, 0.25 d$^{-1}$), but we note that data on non-grazing mortality of *P. antarctica* are currently lacking (Schoemann et al., 2005). Furthermore, based on the assumption that for a given biomass concentration, larger cells are more likely than smaller cells to form aggregates and to subsequently stop photosynthesizing and sink as POC, we use a higher quadratic loss rate for *Phaeocystis* (0.005 d$^{-1}$) than for diatoms (0.001 d$^{-1}$) in the model (see $\gamma_{a,0}$ in Table 1 & Eq. B18).

In summary, the spatio-temporal variability of the relative importance of *Phaeocystis* and diatoms in ROMS-BEC is controlled by the interplay of the environmental conditions and loss processes, which differentially impact the growth and loss rates of these two PFTs and consequently their competitive fitness in the model. In the following, we will describe the model setup and the simulations that were performed to assess the competition between *Phaeocystis* and diatoms throughout the year in the high-latitude SO. The simulations include a set of sensitivity experiments, with the aim to assess the impact of choices of single parameters or parameterizations on the simulated *Phaeocystis* biogeography.

## 2.2 Model setup and sensitivity simulations

With few exceptions, we use the same ROMS-BEC model setup as described in detail in Nissen et al. (2018): At the open northern boundary, we use monthly climatological fields for all tracers (Carton and Giese, 2008; Locarnini et al., 2013; Zweng et al., 2013; Garcia et al., 2014b, a; Lauvset et al., 2016; Yang et al., 2017), and the same data sources are used to initialize the model simulations. At the ocean surface, the model is forced with a 2003-normal year forcing for momentum, heat, and freshwater fluxes (Dee et al., 2011). Satellite-derived climatological total chlorophyll concentrations are used to initialize phytoplankton biomass and to constrain it at the open northern boundary in the model (NASA-OBPG, 2014b), and the fields are extrapolated to depth following Morel and Berthon (1989). Due to the addition of *Phaeocystis*, the partitioning of total chlorophyll onto the different phytoplankton PFTs is adjusted compared to Nissen et al. (2018): 90% is attributed to small phytoplankton, 4% to diatoms and coccolithophores, respectively, and 1% to diazotrophs and *Phaeocystis*, respectively. This partitioning is motivated by the phytoplankton community structure at the open northern boundary at 24° S, where small phytoplankton typically dominate and *P. antarctica* are only a minor contributor to phytoplankton biomass (see e.g. Schoemann et al., 2005; Swan et al., 2016). *Phaeocystis* is initialized with a carbon-to-chlorophyll ratio of 60 mg C (mg chl)$^{-1}$ (same as small phytoplankton and coccolithophores), whereas diatoms are initialized with a ratio of 36 mg C (mg chl)$^{-1}$ (Sathyendranath et al., 2009).

We first run a 30 year long physics-only spin-up, followed by a 10 year long spin-up in the coupled ROMS-BEC setup. Our *Baseline* simulation for this study is then run for an additional 10 years, of which we analyze a daily climatology over the last 5 full seasonal cycles. i.e. from 1 July of year 5 until 30 June of year 10. Apart from having added *Phaeocystis* and adjusted the parameters of the other PFTs as described in section 2.1, the setup of the *Baseline* simulation in this study is thereby identical to the *Baseline* simulation in Nissen et al. (2018). We will evaluate the model's performance with respect to the simulated phytoplankton biogeography in section 3.1 and in the supplementary material.

Furthermore, we perform two sets of sensitivity experiments (22 simulations in total), in order to 1) assess the sensitivity of the simulated *Phaeocystis* biogeography and the competition of *Phaeocystis* and diatoms to chosen parameters and parameterizations (competition experiments, runs 1-8 in Table 2) and 2) systematically assess the sensitivity of the simulated biomass distributions to chosen *Phaeocystis* parameter values (parameter experiments, runs 9-22). For the former set, we set the parameters and parameterizations of *Phaeocystis* to those used for diatoms in ROMS-BEC (runs 1-8 in Table 2). Generally, the differences in parameters between *Phaeocystis* and diatoms affect either the simulated biomass accumulation rates (runs TEMPERATURE, ALPHA$_{PI}$, IRON, and THETA_N_MAX) or loss rates (runs GRAZING, AGGREGATION, and MORTALITY).

By successively eradicating the differences between *Phaeocystis* and diatoms, these simulations allow us to directly quantify the impact of parameter differences on the simulated relative importance of *Phaeocystis* for total phytoplankton biomass. To assess the impact of iron-light interactions on the competitive success of *Phaeocystis* at high SO latitudes, we ultimately run a simulation in which the half-saturation constant of iron ($k_{Fe}$) of *Phaeocystis* is a function of the light intensity, following a polynomial fit of available laboratory data (VARYING_kFE, Fig. A1b; Garcia et al., 2009). For the second set of experiments,

we systematically vary *Phaeocystis* growth and loss parameters by ±50%, and the results of these experiments are discussed in detail in section S2 of the supplementary material. All sensitivity experiments use the same physical and biogeochemical spin-up as the *Baseline* simulation and start from the end of year 10 of the coupled ROMS-BEC spin-up. Each simulation is then run for an additional 10 years, of which the average over the last 5 full seasonal cycles is analyzed in this study.

## 2.3  Data and diagnostics used in the model assessment

### 2.3.1  Evaluating the simulated phytoplankton community structure

We compare the simulated spatio-temporal variability in phytoplankton biomass and community structure to available observations of phytoplankton carbon biomass concentrations from the MAREDAT initiative (O'Brien et al., 2013; Leblanc et al., 2012; Vogt et al., 2012), satellite-derived total chlorophyll concentrations (Fanton d'Andon et al., 2009; Maritorena et al., 2010), DMS measurements (Curran and Jones, 2000; Lana et al., 2011), the ecological niches suggested for SO phytoplankton

taxa (Brun et al., 2015), and the CHEMTAX climatology based on high performance liquid chromatography (HPLC) pigment data (Swan et al., 2016). The latter provides seasonal estimates of the mixed layer average community composition, which we compare to the seasonally and top 50 m averaged model output of each phytoplankton's contribution to total chlorophyll biomass. The CHEMTAX analysis splits the phytoplankton community into diatoms, nitrogen fixers (such as *Trichodesmium*), pico-phytoplankton (such as *Synechococcus* and *Prochlorococcus*), dinoflagellates, cryptophytes, chlorophytes (all three com-

bined into the single group "Others" here), and haptophytes (such as coccolithophores and *Phaeocystis*). As noted in Swan et al. (2016), the differentiation between coccolithophores and *Phaeocystis* in the CHEMTAX analysis is difficult and prone to error. Possibly, this is due to the large variability in pigment composition of *Phaeocystis* in response to varying environmental conditions, especially regarding light and iron levels (Smith et al., 2010; Wright et al., 2010). Coccolithophores have been reported to only grow very slowly at low temperatures (below ∼8° C, Buitenhuis et al., 2008), and in the SO, their abundance in the high

latitudes south of the polar front is very low (Balch et al., 2016). Therefore, whenever the climatological temperature in the World Ocean Atlas 2013 (Locarnini et al., 2013) is below 2° C at the time and location of the respective HPLC observation, we re-assign data points identified as "Hapto-6" (hence e.g. *Emiliania huxleyi*) in the CHEMTAX analysis to "Hapto-8" (hence e.g. *Phaeocystis antarctica*). Throughout the manuscript, this new category ("Hapto-8 re-assigned") is indicated separately in the respective figures, and leads to a better correspondence of the functional types included in the CHEMTAX-based climatology

by Swan et al. (2016) and the PFTs in ROMS-BEC.

To assess the controlling factors of the simulated PFT distributions in our model, we analyze the simulated summer (December-March; DJFM) top 50 m average biomass distribution of the different model PFTs south of 40° S in environmental niche space.

To that aim, we bin the simulated carbon biomass concentrations of *Phaeocystis*, diatoms, and coccolithophores in ROMS-BEC as a function of the temperature [° C], nitrate concentration [mmol m$^{-3}$], iron concentration [$\mu$mol m$^{-3}$], and mixed layer pho-
tosynthetically active radiation (MLPAR; W m$^{-2}$). Subsequently, we compare the simulated ecological niche to that observed for abundant SO species of each model PFT (such as *Phaeocystis antarctica*, *Fragilariopsis kerguelensis*, *Thalassiosira* sp., or *Emiliania huxleyi*, see Brun et al., 2015). In section 3.3 of this manuscript, only the results for *Phaeocystis* and diatoms will be shown, the corresponding figures for coccolithophores can be found in the supplementary material (Fig. S2 & S3). While this analysis informs on possible links between the competitive fitness of a PFT and the environmental conditions it
lives in, the assessment is limited to a qualitative inter-comparison due to difficulties in comparing a model PFT to individual phytoplankton species, a sampling bias towards the summer months and the low latitudes, and the neglect of loss processes such as zooplankton grazing to explain biomass distributions. As a consequence, the ecological niche analysis does not allow for the assessment of any temporal variability in PFT biomass concentrations.

In order to assess the simulated seasonality and the seasonal succession of *Phaeocystis* and diatoms, we identify the bloom
peak as the day of peak chlorophyll concentrations throughout the year. Besides the timing of the bloom peak, phytoplankton phenology is typically characterized by metrics such as the day of bloom initiation or the day of bloom end (see e.g. Soppa et al., 2016). In this regard, the timing of the bloom start is known to be sensitive to the chosen identification methodology (Thomalla et al., 2015). At high latitudes, the identification of the bloom start based on remotely sensed chlorophyll concentrations is additionally impaired by the large number of missing data in all seasons (even in the summer months, a large part of the SO is
sampled by the satellite in less than 5 of the 21 available years, see Fig. S4), complicating any comparison of the high-latitude satellite-derived bloom start with output from models such as ROMS-BEC. To minimize the uncertainty due to the low data coverage in the region of interest for this study, and as the seasonal succession of *Phaeocystis* and diatoms in the high-latitude SO is mostly inferred from the timing of observed maximum abundances in the literature (e.g. Peloquin and Smith, 2007; Smith et al., 2011), we focus our discussion of the simulated bloom phenology on the timing of the bloom peak (Hashioka
et al., 2013). To evaluate the model's performance, we compare the timing of the total chlorophyll bloom peak in the *Baseline* simulation of ROMS-BEC to the bloom timing derived from climatological daily chlorophyll data from Globcolor (climatology from 1998-2018 based on the daily 25 km chlorophyll product, see Fanton d'Andon et al., 2009; Maritorena et al., 2010).

### 2.3.2 Phytoplankton competition and succession

In ROMS-BEC, phytoplankton biomass $P^i$ (mmol C m$^{-3}$, $i \in \{PA, D, C, SP, N\}$) is determined by the balance between
growth ($\mu^i$) and loss terms (grazing by zooplankton $\gamma_g^i$, non-grazing mortality $\gamma_m^i$, and aggregation $\gamma_a^i$, see appendix B for a full description of the model equations). Here, in order to disentangle the factors controlling the relative importance of *Phaeocystis* and diatoms for total phytoplankton biomass throughout the year, we use the metrics first introduced by Hashioka et al. (2013) and then applied to assess the competition of diatoms and coccolithophores in ROMS-BEC in Nissen et al. (2018). Same as in Nissen et al. (2018), the relative growth ratio $\mu_{rel}^{ij}$ of phytoplankton $i$ and $j$ (e.g. diatoms and *Phaeocystis*) is defined as the
ratios of their specific growth rates ($\mu^i$, d$^{-1}$), which in turn depends on environmental dependencies regarding the temperature

$T$, nutrients $N$, and irradiance $I$, following:

$$\mu_{\text{rel}}^{\text{DPA}} = \log \frac{\mu^{\text{D}}}{\mu^{\text{PA}}}$$

$$= \underbrace{\log \frac{f^{\text{D}}(T) \cdot \mu_{\max}^{\text{D}}}{\mu_{\text{T}}^{\text{PA}}}}_{\beta_T} + \underbrace{\log \frac{g^{\text{D}}(N)}{g^{\text{PA}}(N)}}_{\beta_N \sim \beta_{\text{Fe}}} + \underbrace{\log \frac{h^{\text{D}}(I)}{h^{\text{PA}}(I)}}_{\beta_I} \tag{2}$$

In the above equation, the specific growth rate $\mu^{\text{i}}$ of each phytoplankton $i$ is calculated as a multiplicative function of a

temperature-limited growth rate ($f^{\text{D}}(T) \cdot \mu_{\max}^{\text{D}}$ for diatoms and $\mu_{\text{T}}^{\text{PA}}$ for *Phaeocystis*; see Eq. B5 & Eq. 1), a nutrient limitation

term ($g^{\text{i}}(N)$, limitation of each nutrient is calculated using a Michaelis-Menten function, and the most-limiting one is then

used here; see Eq. B8), and a light limitation term ($h^{\text{i}}(I)$; see Eq. B9 and Geider et al., 1998). Further, $\beta_T$, $\beta_N$, and $\beta_I$ describe

the logarithmic ratio of the limitation by temperature, nutrients, and light of growth by diatoms and *Phaeocystis*. Thereby,

these terms denote the log-normalized contribution of each environmental factor to the simulated relative growth ratio. At

high-latitudes south of $60°$ S, the ratio of the nutrient limitation of growth $\beta_N$ corresponds to that of the iron limitation $\beta_{\text{Fe}}$ in

our model (Fig. S1). Consequently, environmental conditions regarding temperature, iron, and light decide whether the relative

growth ratio is positive or negative at a given location and point in time, i.e., which of the two phytoplankton types has a higher

specific growth rate and hence a competitive advantage over the other regarding growth.

Similarly, the relative grazing ratio $\gamma_{\text{g,rel}}^{\text{ij}}$ of phytoplankton $i$ and $j$ (e.g. diatoms and *Phaeocystis*) is defined as the ratio of

their specific grazing rates ($\gamma_{\text{g}}^{\text{i}}$, $\text{d}^{-1}$) following:

$$\gamma_{\text{g,rel}}^{\text{DPA}} = \log \frac{\frac{\gamma_{\text{g}}^{\text{PA}}}{P^{\text{PA}}}}{\frac{\gamma_{\text{g}}^{\text{D}}}{P^{\text{D}}}} \tag{3}$$

In ROMS-BEC, grazing on each phytoplankton $i$ is calculated using a Holling Type II ingestion function (Nissen et al., 2018).

As described in section 2.1, *Phaeocystis* and diatoms in ROMS-BEC do not only differ in parameters describing the zooplank-

ton grazing pressure they experience, but in parameters describing their non-grazing mortality and aggregation losses as well.

Therefore, in accordance with the relative grazing ratio defined above, we define the relative mortality ratio ($\gamma_{\text{m,rel}}^{\text{ij}}$) and the rel-

ative aggregation ratio ($\gamma_{\text{a,rel}}^{\text{ij}}$) of phytoplankton $i$ and $j$ (e.g. diatoms and *Phaeocystis*) as the ratio of their specific non-grazing

mortality rates ($\gamma_{\text{m}}^{\text{i}}$, $\text{d}^{-1}$) and aggregation rates ($\gamma_{\text{a}}^{\text{i}}$, $\text{d}^{-1}$), respectively, following:

$$\gamma_{\text{m,rel}}^{\text{DPA}} = \log \frac{\frac{\gamma_{\text{m}}^{\text{PA}}}{P^{\text{PA}}}}{\frac{\gamma_{\text{m}}^{\text{D}}}{P^{\text{D}}}} \tag{4}$$

$$\gamma_{\text{a,rel}}^{\text{DPA}} = \log \frac{\frac{\gamma_{\text{a}}^{\text{PA}}}{P^{\text{PA}}}}{\frac{\gamma_{\text{a}}^{\text{D}}}{P^{\text{D}}}} \tag{5}$$

Since the total specific loss rate ($\gamma_{\text{total}}^{\text{ij}}$, $\text{d}^{-1}$) of phytoplankton $i$ is the addition of its specific grazing, non-grazing mortality,

and aggregation loss rates, the relative total loss ratio $\gamma_{\text{total,rel}}^{\text{ij}}$ of phytoplankton $i$ and $j$ (e.g. diatoms and *Phaeocystis*) is

defined as

$$\gamma_{\text{total,rel}}^{\text{DPA}} = \log \frac{\frac{\gamma_{\text{g}}^{\text{PA}}}{P^{\text{PA}}} + \frac{\gamma_{\text{m}}^{\text{PA}}}{P^{\text{PA}}} + \frac{\gamma_{\text{a}}^{\text{PA}}}{P^{\text{PA}}}}{\frac{\gamma_{\text{g}}^{\text{D}}}{P^{\text{D}}} + \frac{\gamma_{\text{m}}^{\text{D}}}{P^{\text{D}}} + \frac{\gamma_{\text{a}}^{\text{D}}}{P^{\text{D}}}} \qquad (6)$$

If $\gamma_{\text{total,rel}}^{\text{DPA}}$ is positive, the specific total loss rate of *Phaeocystis* is larger than that of diatoms (and accordingly for the individual loss ratios in Eq. 3-5), and loss processes promote the accumulation of diatom biomass relative to that of *Phaeocystis*. While the maximum grazing rate on *Phaeocystis* is lower than that of diatoms, their non-grazing mortality and aggregation losses are higher (see section 2.1 and Table 1). Ultimately, at any given location and point in time, the interaction between the phytoplankton biomass concentrations (impacting the respective loss rates) and environmental conditions (impacting the

respective growth rate) will determine the relative contribution of each phytoplankton type $i$ to total phytoplankton biomass. Here, we use these metrics to assess the controls on the simulated seasonal evolution of the relative importance of *Phaeocystis* and diatoms in the high-latitude SO.

## 3 Results

### 3.1 Phytoplankton biogeography and community composition in the SO

In the 5-PFT *Baseline* simulation of ROMS-BEC, total summer chlorophyll is highest close to the Antarctic continent ($>10$ mg chl m$^{-3}$) and decreases northwards to values $<1$ mg chl m$^{-3}$ close to the open northern boundary (Fig. 1a). While this south-north gradient is in broad agreement with remotely sensed chlorophyll concentrations (Fig. 1b), our model generally overestimates high-latitude chlorophyll levels, which has already been noted for the 4-PFT setup of ROMS-BEC (Nissen et al., 2018). With *Phaeocystis* added, the model overestimates annual mean satellite derived surface chlorophyll biomass estimates

by 18% (40.8 Gg chl in ROMS-BEC between 30-90° S compared to 34.5 Gg chl in the MODIS Aqua chlorophyll product, Table 3, NASA-OBPG, 2014a; Johnson et al., 2013) and satellite derived NPP by 38-42% (17.2 compared to 12.1-12.5 Pg C yr$^{-1}$, Table 3, Behrenfeld and Falkowski, 1997; O'Malley, last access: 16 May 2016; Buitenhuis et al., 2013). This bias is largest south of 60° S, where NPP and surface chlorophyll are overestimated by a factor 1.8-4.4 and 1.8, respectively (Table 3), and the bias is likely due to a combination of underestimated high-latitude chlorophyll concentrations in satellite-derived

products (Johnson et al., 2013) and the missing complexity in the zooplankton compartment in ROMS-BEC, as biases in the simulated physical fields (temperature, light) have been shown to only explain a minor fraction of the simulated high-latitude biomass overestimation (Nissen et al., 2018).

The simulated carbon biomass distributions of colonial *Phaeocystis*, diatoms, coccolithophores, and SP are distinct in the model (Fig. 1c-f, showing top 50 m averages). The simulated summer *Phaeocystis* biomass is highest south of 50° S, with

highest concentrations of 10 mmol C m$^{-3}$ at $\sim$74° S. In the model, average *Phaeocystis* biomass concentrations quickly decline to levels $<0.1$ mmol C m$^{-3}$ north of 50° S (Fig. 1c), a direct result of the restriction of *Phaeocystis* growth to temperatures $< \sim$8° C in the model (Fig. A1a). This is in broad agreement with in situ observations, which suggest highest concentrations ($>20$ mmol C m$^{-3}$) south of $\sim$75° S, and concentrations $<5$ mmol C m$^{-3}$ north of $\sim$65° S (Fig. 1c & Fig. S5a & b). As a

response to the addition of *Phaeocystis* to ROMS-BEC, the simulated high-latitude diatom biomass concentrations decrease compared to the 4-PFT setup of the model (Nissen et al., 2018). In the 5-PFT setup, the model simulates highest diatom biomass south of 60° S with maximum concentrations of $\sim$7 mmol C m$^{-3}$ at 72° S (top 50 m mean; $\sim$17 mmol C m$^{-3}$ in 4-PFT setup) and rapidly declining concentrations north of 60° S (Fig. 1d). Nevertheless, the simulated summer diatom biomass levels are still overestimated compared to carbon biomass estimates (Fig. S5c, Leblanc et al., 2012) and satellite derived diatom chlorophyll estimates (Soppa et al., 2014, comparison not shown). In contrast to both *Phaeocystis* and diatoms, the simulated biomass levels of coccolithophores are highest in the subantarctic (highest concentrations of 3 mmol C m$^{-3}$ on the Patagonian Shelf, Fig. 1e & S3d). Overall, their simulated SO biogeography agrees well with the position of the "Great Calcite Belt" (Balch et al., 2011, 2016) and remains largely unchanged compared to the 4-PFT setup (Nissen et al., 2018).

Taken together, the model simulates a phytoplankton community with substantial contributions of coccolithophores and *Phaeocystis* in the subantarctic and high-latitude SO, respectively (Fig. 2a). CHEMTAX data generally support this latitudinal trend (see Fig. 2b-d and section 2.3.1, Swan et al., 2016). Averaged over 30-90° S (60-90° S), the simulated relative contributions of *Phaeocystis*, diatoms, and coccolithophores to total chlorophyll in summer are 20±28% (33±34%; subarea mean as shown in Fig. 2b & c ±1$\sigma$ in space), 68±33% (64±33%), and 5±17% (<1±2%), respectively, in good agreement with the CHEMTAX climatology (28% (27%), 46% (48%), and 3% (1%), respectively). Acknowledging the uncertainty in the attribution of the group "Other" in the CHEMTAX data to a model PFT ("Other" includes dinoflagellates, cryptophytes, and chlorophytes here, see section 2.3.1), the model also captures the seasonal evolution of the relative importance of *Phaeocystis* and diatoms reasonably well, both averaged over 30-90° S (Fig. 2b) and at high SO latitudes (Fig. 2c-d). The model overestimates the contribution of *Phaeocystis* in fall (39±14% as compared to 24% in CHEMTAX) and spring (51±22% as compared to 28%) between 60-90° S and in the Ross Sea, respectively (Fig. 2c-d), but the limited number of data points available in the CHEMTAX climatology in this area and the uncertainty in the attribution of pigments in CHEMTAX to the *Phaeocystis* PFT in ROMS-BEC have to be noted (see section 2.3.1).

In the 4-PFT setup of ROMS-BEC, the simulated summer phytoplankton community south of 60° S was often almost solely composed of diatoms (Fig. S6 and Nissen et al., 2018), suggesting that the implementation of *Phaeocystis* led to a substantial improvement in the representation of the observed high-latitude community structure (Fig. 2). Concurrently, as the distribution of silicic acid and nitrate is directly impacted by the relative importance of silicifying and non-silicifying phytoplankton, such as *Phaeocystis*, in the community, the addition of *Phaeocystis* to the model led to an improvement in the simulated high-latitude nutrient distributions when comparing to climatological data from the World Ocean Atlas (WOA, Fig. S7d-f, Garcia et al., 2014b). Upon the addition of *Phaeocystis*, the zonal average location of the silicate front, i.e., the latitude at which nitrate and silicic acid concentrations are equal (Freeman et al., 2018), is shifted northward by $\sim$7° C in ROMS-BEC (from 57.1° S in 4-PFT setup to 50° S in 5-PFT setup, see Fig. S8). While this is further north than suggested by WOA data (56.5° S, Fig. S8b and Garcia et al., 2014b), this can certainly be expected to affect the competitive fitness of individual phytoplankton types in the subantarctic and possibly at lower latitudes, which we did not assess further in this study. Overall, our model agrees with observational data that *Phaeocystis* is an important member of the high-latitude phytoplankton community. In the remainder

of the manuscript, we will therefore explore the temporal variability in the relative importance of diatoms and *Phaeocystis* and its implications for SO carbon cycling in more detail.

### 3.2 Phytoplankton phenology and the seasonal succession of *Phaeocystis* and diatoms

Maximum total chlorophyll concentrations are simulated for the first half of December across latitudes in ROMS-BEC (solid blue line in Fig. 3a), and at high SO latitudes south of $60°$ S, total chlorophyll blooms start already in late September in the model (not shown). Thereby, the model-derived timing of total chlorophyll bloom start and peak is 2-3 and 1-2 months earlier, respectively, than satellite-derived estimates (for bloom peak, see black line in Fig. 3a, for bloom start, see e.g. Thomalla et al., 2011). Yet, compared to the 4-PFT setup (dashed blue line in Fig. 3a), the simulated timing of peak chlorophyll levels improved in this study, with peak chlorophyll delayed by on average a week in the model upon the implementation of *Phaeocystis*. The simulated physical biases (i.e., generally too high temperatures and too shallow mixed layer depths, both favoring an earlier onset of the phytoplankton bloom, see Nissen et al., 2018) only partially explain the bias in the simulated timing of maximum chlorophyll levels (see red and green dashed lines in Fig. S9a), suggesting that biological factors must explain the difference between ROMS-BEC and the satellite product. As diatoms dominate the phytoplankton community at peak total chlorophyll concentrations for all latitudinal averages in the model domain (compare their bloom timing in Fig. 3c to Fig. 3a and to the simulated community composition in Fig. 2b-d, but note that *Phaeocystis* often dominate in coastal areas, not shown), the mismatch in timing is likely related to the representation of this PFT in the model, and is possibly at least partly caused by their comparatively high growth rates at low temperatures (see Fig. A1a).

In contrast to diatoms, maximum zonally averaged chlorophyll concentrations of *Phaeocystis* are simulated for late November or early December across most latitudes in the model (only around $70°$ S a peak in late January is simulated, Fig. 3b; note that locally, maximum *Phaeocystis* chlorophyll concentrations exceed 10 mg chl m$^{-3}$, not shown here). Overall, the timing of simulated peak *Phaeocystis* chlorophyll levels corresponds well to the suggested timing of observed maximum seawater DMSP concentrations (peak in November/December in Curran et al., 1998; Curran and Jones, 2000) and the delayed maximum atmospheric DMS concentrations (January/February, e.g. Nguyen et al., 1990; Ayers et al., 1991). This further corroborates the hypothesis that the bias in the timing of maximum total chlorophyll levels in ROMS-BEC is likely caused by how diatoms are parameterized in the model (see e.g. the rather high temperature-limited growth rate of diatoms at low temperatures compared to available laboratory data, see Fig. A1). Taken together, the model simulates a succession from *Phaeocystis* to diatoms close to the Antarctic continent (south of $72°$ S, see also Fig. S9b) and in some parts of the open ocean north of $68°$ S (Fig. 3d & Fig. S9b). The difference in the timing of the bloom peak between the two PFTs is largely <10 days when averaged zonally, but locally exceeds 30 days when looking at individual grid cells in the model (Fig. S9b), in broad agreement with observations, which suggest up to 2 months between the peak chlorophyll concentrations of *Phaeocystis* and diatoms in the Ross Sea (see e.g. Peloquin and Smith, 2007; Smith et al., 2011). Subsequently, we will assess how environmental conditions and biomass loss processes interact to control the competition between *Phaeocystis* and diatoms at high SO latitudes.

## 3.3 Drivers of the high-latitude biogeography and seasonal succession of *Phaeocystis* and diatoms

Relating the observed or simulated PFT biomass concentrations to the concurrent environmental conditions allows for an assessment of the ecological niche of the PFT in question. In ROMS-BEC, *Phaeocystis* and diatoms occupy distinct ecological niches in the *Baseline* simulation, in agreement with their distinct geographic distributions in summer (Fig. 1c-d). Between 40-90° S, the niche center of DJFM average *Phaeocystis* biomass is simulated at a nitrate concentration of 18.8 mmol m$^{-3}$ (inter quartile range (IQR) 16.6-20.5 mmol m$^{-3}$), a temperature of 1.1° C (IQR -0.2-2.6° C), and MLPAR of 27.8 W m$^{-2}$ (IQR 24.3-32 W m$^{-2}$, Fig. 4a & c). Since the diatom PFT in ROMS-BEC represents multiple species (in contrast to the *Phaeocystis* PFT), diatoms occupy a wider niche in temperature (IQR 0.8-8.5° C, niche center at 5° C) and nitrate (IQR 11-19.5 mmol m$^{-3}$, niche center at 15.5 mmol m$^{-3}$) in the model, which is in agreement with the ecological niches of important SO diatom and *Phaeocystis* species derived by Brun et al. (2015) based on presence/absence observations and species distribution models (Fig. 4a & b). In ROMS-BEC, the niche center is only at marginally higher MLPAR for diatoms than for *Phaeocystis* (28.9 W m$^{-2}$ compared to 27.8 W m$^{-2}$, respectively, Fig. 4c & d) and is at higher MLPAR for both PFTs than available observations for important SO species suggest (~10 W m$^{-2}$ and ~20 W m$^{-2}$ for *Phaeocystis* and diatoms, respectively, see Fig. 4c & d). While this bias in the MLPAR niche is consistent with the mixed layer depth bias in ROMS-BEC (~10 m; Nissen et al., 2018), the small difference in the MLPAR niche center between *Phaeocystis* and diatoms implies a minor role for MLPAR in controlling the differences in DJFM average biomass concentrations of these two PFTs (Fig. 1c-d). With regard to iron, the two PFTs do not occupy distinct ecological niches in ROMS-BEC (niche centers at 0.32 $\mu$mol m$^{-3}$ for both PFTs, see Fig. S3). Yet, as all simulated phytoplankton growth is most limited by iron availability in the high-latitude SO compared to the availability of other nutrients (Fig. S1), this suggests that the spatio-temporal averaging applied for the niche analysis here potentially precludes the assessment of the role of iron in the competition between *Phaeocystis* and diatoms, especially on a sub-seasonal scale. We conclude that the simulated ecological niches of *Phaeocystis* and diatoms are largely in agreement with available observations, but acknowledge the difficulties in comparing the ecological niche of a model PFT to those of individual phytoplankton species or groups, a sampling bias towards temperate and tropical species/strains and the overall low data coverage in the high-latitude SO in Brun et al. (2015), and the limitation of this niche analysis to inform about the role of top-down factors and sub-seasonal environmental variability in controlling the simulated biogeography of phytoplankton types.

The temporal evolution of the relative growth ratio, i.e., the ratio of the specific growth rates of diatoms and *Phaeocystis*, informs about the competitive advantage of one PFT over the other throughout the year due to bottom-up factors and can be broken down into the different environmental contributors for each phytoplankton type at any point in time (Eq. 2). In the 5-PFT *Baseline* simulation of ROMS-BEC, the relative growth ratio is only positive ($\mu^{\mathrm{D}} > \mu^{\mathrm{PA}}$) between early December and early February between 60-90° S ($\mu^{\mathrm{D}}$ is on average 5% larger than $\mu^{\mathrm{PA}}$ in summer, but 5-6% smaller in the other seasons, Fig. 5a & c) and only between mid-December and mid-January in the Ross Sea ($\mu^{\mathrm{PA}}$ is up to 38% larger than $\mu^{\mathrm{D}}$ in spring, Fig. 5b & d). Hence, bottom-up factors promote the accumulation of *Phaeocystis* relative to diatom biomass over much of the year, particularly in the Ross Sea. In both areas, as expected from the chosen half-saturation constants ($k_{\mathrm{Fe}}^{\mathrm{PA}} > k_{\mathrm{Fe}}^{\mathrm{D}}$; Table 1),

the iron limitation of *Phaeocystis* growth is stronger than that of diatoms in the model, and iron availability is an advantage for diatoms at all times ($\beta_{\mathrm{Fe}} > 0$; up to 14% stronger iron limitation of *Phaeocystis* in both areas in summer, blue areas in Fig. 5a-d). Yet, the two subareas differ in the simulated temperature and light limitation of growth of *Phaeocystis* and diatoms. Overall, temperature is limiting diatom growth more than *Phaeocystis* growth in both subareas throughout the year ($\beta_{\mathrm{T}} < 0$), but this difference is rather small in summer between 60-90° S (5%, but up to 19% stronger growth limitation in the Ross Sea, red areas in Fig. 5a-d, see also Fig. A1). Similarly, the difference in light limitation between diatoms and *Phaeocystis* is rather small between 60-90° S (3-4% throughout the year, yellow areas in Fig. 5a & c), implying that their differences in $\alpha_{\mathrm{PI}}$ (43% higher for *Phaeocystis*, see Table 1) are balanced by differences in photoacclimation in ROMS-BEC in this area (see Eq. B9 and Geider et al., 1998, note that $\theta_{\mathrm{chl:N,\,max}}^{\mathrm{D}} > \theta_{\mathrm{chl:N,\,max}}^{\mathrm{PA}}$, see Table 1). In contrast, in the Ross Sea, differences in light limitation between diatoms and *Phaeocystis* are large, especially in spring (the growth of diatoms is 32% more light limited; Fig. 5b & d). Therefore, the difference in light limitation predominantly controls the seasonality of the relative growth ratio (Fig. 5b) and promotes the dominance of *Phaeocystis* over diatoms early in the growing season in this area in our model (Fig. 5j), which is not simulated when averaging over 60-90° S (Fig. 5i). Nevertheless, acknowledging the sensitivity of the simulated *Phaeocystis* and diatom biomass levels to all chosen model parameters describing the growth of the respective PFT (the annual mean biomass changes by >17% and >14% for *Phaeocystis* and diatoms, respectively, in the experiments TEMPERATURE, ALPHA$_{\mathrm{PI}}$, and IRON, Fig. A2 & Fig. S10), the sensitivity simulations support the importance of light in controlling the annual mean high-latitude phytoplankton community structure for both subareas, as the elimination of the differences in $\alpha_{\mathrm{PI}}$ between the PFTs results in the largest biomass changes both between 60-90° S (-76% and +52% for *Phaeocystis* and diatoms, respectively) and in the Ross Sea (-87% and +86%, Fig. A2). Altogether, in ROMS-BEC, differences in growth between diatoms and *Phaeocystis* are mostly controlled by seasonal differences in iron/temperature (60-90° S) and iron/light conditions (Ross Sea), respectively. Still, given the simulated growth advantage of *Phaeocystis* throughout much of the growing season in both subareas, bottom-up factors alone cannot explain why *Phaeocystis* only dominates over diatoms temporarily (Fig. 5i & j), implying that top-down factors need to be considered to explain their biomass evolution in our model.

In both subareas, the simulated relative total loss ratio is positive throughout spring and summer, implying that the specific total loss rate of *Phaeocystis* is higher than that of diatoms ($\gamma_{\mathrm{total}}^{\mathrm{PA}} > \gamma_{\mathrm{total}}^{\mathrm{D}}$, see Eq. 6), which favors the accumulation of diatom biomass relative to that of *Phaeocystis* (Fig. 5e-h). In fact, the total loss rate of *Phaeocystis* is on average 17%/38% (60-90° S) and 18%/40% (Ross Sea) higher than that of diatoms in spring/summer (Fig. 5g & h), despite the higher prescribed maximum grazing rate on *Phaeocystis* in ROMS-BEC (Table 1). In the model, the relative total loss ratio is only negative in early fall in both subareas ($\gamma_{\mathrm{total}}^{\mathrm{D}} > \gamma_{\mathrm{total}}^{\mathrm{PA}}$, Fig. 5e & f), but the difference between diatoms and *Phaeocystis* in their specific total loss rates is rather small in this season (9% and 3% between 60-90° S and in the Ross Sea, respectively, Fig. 5g & h). In all top-down sensitivity experiments, the simulated change in *Phaeocystis* biomass levels is larger than for the bottom-up experiments (>20% for experiments GRAZING, AGGREGATION, and MORTALITY, see Fig. A2), and the dominance of *Phaeocystis* over diatoms increases in magnitude and duration both between 60-90° S and in the Ross Sea if disadvantages of *Phaeocystis* in the loss processes are eliminated (Fig. S10). The simulated seasonality of the total loss ratio is the result of the interplay between losses through grazing, aggregation, and non-grazing mortality of each phytoplankton type in ROMS-BEC (Eq. 6,

colors in Fig. 5e-h). Of all three loss pathways, differences in aggregation losses in the *Baseline* simulation are largest between
*Phaeocystis* and diatoms both between 60-90° S (up to 200% higher aggregation losses for *Phaeocystis* in summer, yellow in
Fig. 5e & g) and in the Ross Sea (up to 250% higher in summer, Fig. 5f & h). In comparison, differences between *Phaeocystis*
and diatoms in grazing (up to 16% and 14% between 60-90° S and in the Ross Sea, respectively) and mortality losses (50%
everywhere) are considerably smaller (see blue and red areas in Fig. 5e-h, respectively), suggesting that aggregation losses
predominantly contribute to the simulated differences in the total loss rates between *Phaeocystis* and diatoms.

In summary, between 60-90° S, the simulated growth advantage of *Phaeocystis* early in the season (facilitated by advantages
in the temperature limitation of their growth) are not large enough to outweigh the disadvantages in iron limitation of their
growth and in the biomass losses they experience. As a result, in spring and summer, *Phaeocystis* do not accumulate substantial
biomass relative to (or even dominate over) diatoms in this subarea in ROMS-BEC. In the Ross Sea, however, the simulated
growth advantages of *Phaeocystis* (resulting from advantages in the light and temperature limitation of their growth) are large
enough to outweigh the disadvantages in iron limitation and specific biomass loss rates, allowing them to dominate over
diatoms early in the growing season in our model and explaining the simulated succession from *Phaeocystis* to diatoms close
to the Antarctic continent (see also section 3.2). Ultimately, this simulated spatio-temporal variability in the relative importance
of *Phaeocystis* and diatoms has implications for SO carbon cycling, which we will assess in the following.

### 3.4 Quantifying the importance of *Phaeocystis* for the SO carbon cycle

*Phaeocystis* is an important member of the SO phytoplankton community in our model, particularly south of 60°S, where
it contributes 46±21% and 40±20% to total annual NPP and POC formation, respectively (Table 3 & Fig. 6). Even when
considering the entire region south of 30°S, the contribution of *Phaeocystis* to NPP (15±24%) and POC production (16±22%)
is sizeable. The simulated spatial differences in phytoplankton community structure have direct implications for the fate of
organic carbon upon biomass loss, and Fig. 6 illustrates the annually integrated importance of different pathways of POC
formation related to each PFT in ROMS-BEC. Overall, in our model, the p ratio, i.e., the fraction of NPP that is transformed to
sinking POC (Laufkötter et al., 2016), is higher at high latitudes south of 60° S (45%) than the domain average (37%, Fig. 6).
This is a direct result of the higher fraction of large phytoplankton types, i.e., *Phaeocystis* and diatoms, in the ecosystem south
of 60° S (67% of total carbon biomass) than between 30-90° S (47%; Fig. 6, but see also Fig. 2), facilitating more carbon
export relative to NPP in the model. In fact, our model results suggest that these two large phytoplankton types contribute more
to POC formation than to total biomass (76% and 89% of total POC formation between 30-90° S and 60-90° S, respectively;
compare yellow and green boxes in Fig. 6). Integrated annually, diatoms contribute most of all PFTs to POC formation in
our model (60% and 49% between 30-90° S and 60-90° S, respectively, Fig. 6). For both diatoms and *Phaeocystis*, grazing
by zooplankton (i.e., the formation of fecal pellets) is the most important pathway of POC production in ROMS-BEC (black
arrows in Fig. 6, 9%/52% and 20%/37% of total POC production for *Phaeocystis*/diatoms between 30-90° S and 60-90° S,
respectively). Yet, at high latitudes (60-90° S), aggregation of *Phaeocystis* biomass contributes significantly to POC formation
(20% of total POC production, 9% of NPP, grey arrows in Fig. 6b). Given that the loss of biomass via a given pathway is
a function of the local biomass concentrations of each PFT at any given point in time (see section 2.1 and appendix B), the

relative importance of any PFT or biomass loss pathway for total POC formation and hence the total POC produced vary throughout the year.

The seasonal variability in total POC formation is governed by the variability in total chlorophyll concentrations both between 30-90° S and 60-90° S, and peak POC formation rates of 35 mmol m$^{-2}$ d$^{-1}$ (30-90° S) and 65 mmol m$^{-2}$ d$^{-1}$ (60-90° S) are simulated for December in ROMS-BEC (Fig. 7a & b; compare to Fig. 3a). Similarly, the contribution of *Phaeocystis* and diatoms to total POC formation closely follows their contribution to total biomass over the year, with the contribution of *Phaeocystis* peaking in January (23%) and February (63%) for 30-90° S and 60-90° S, respectively (Fig. 7a & b; compare timing to

Fig. 2b & c). As a result of the close link between POC formation and chlorophyll concentrations in ROMS-BEC, the majority of the annual POC formation occurs between November and February in our model (64% and 88% south of 30° S and 60° S, respectively, Fig. 7c & d). During these months, the simulated pathways of POC formation differ from the annually integrated perspective in Fig. 6, especially for *Phaeocystis*. While grazing is the most important pathway throughout the year for diatoms in both subareas in our model (red bars in Fig. 7e & f), aggregation of *Phaeocystis* is as important as grazing in December and

January between 30-90° S (blue bars in Fig. 7e) and even dominantly contributes to POC formation between November and January at high SO latitudes (up to 65%, blue bars in Fig. 7f). Altogether, this implies that both spatial and temporal variations in SO phytoplankton community structure critically impact the fate of carbon beyond the upper ocean.

## 4    Discussion

### 4.1    Drivers of phytoplankton biogeography and the competition between *Phaeocystis* and diatoms

In ROMS-BEC, the interplay of iron availability with temperature (60-90° S) and light levels (Ross Sea), respectively, largely controls the competitive fitness of *Phaeocystis* relative to diatoms in the high-latitude SO. Yet, differences in the simulated biomass loss rates between the two PFTs (in particular via aggregation) need to be considered in order to explain why peak *Phaeocystis* biomass levels precede those of diatoms only close to the Antarctic continent in the model. In the literature, the spatial distribution of *Phaeocystis* and diatoms and the temporal succession from *Phaeocystis* to diatoms is almost exclusively

discussed in terms of light and iron availability (see e.g. Arrigo et al., 1999; Smith et al., 2014). In this context, regions/times of low light and/or high mixed layer depth are typically associated with high *Phaeocystis* abundance (Alvain et al., 2008; Smith et al., 2014), explaining their bloom in spring, whereas iron availability has been suggested to largely control the magnitude of the summer diatom bloom (Peloquin and Smith, 2007; Smith et al., 2011). This is in agreement with the simulated dynamics and parameters chosen in ROMS-BEC, in which the difference in light limitation between growth of *Phaeocystis* and diatoms

facilitates early *Phaeocystis* blooms in the Ross Sea. Yet, it has to be noted that advantages in temperature limitation contribute to the growth advantage of *Phaeocystis* in the high-latitude SO in ROMS-BEC as well and without it, *Phaeocystis* would contribute substantially less to high-latitude phytoplankton biomass (Fig. A2). Currently, this growth advantage of *Phaeocystis* at temperatures <4° C is possibly underestimated in the model, as diatom growth at low temperatures is currently overestimated when comparing to available laboratory measurements (Fig. A1a). Nevertheless, in agreement with Peloquin and Smith (2007)

and Smith et al. (2011), when diatoms reach peak chlorophyll levels in summer in our model, the simulated difference in iron

limitation between the two PFTs is largest across the high-latitude SO (Fig. 5a & b), suggesting that any change in summer iron availability will indeed strongly impact peak diatom and hence total chlorophyll levels in ROMS-BEC.

An important limitation in the assessment of the role of iron in controlling the relative importance of *Phaeocystis* in the high-latitude phytoplankton community is the assumption of a constant $k_{Fe}$ of *Phaeocystis* in the model (0.2 $\mu$mol m$^{-3}$, Table 1). In laboratory experiments, the affinity of *Phaeocystis* for iron has been shown to be sensitive to light (Garcia et al., 2009), which is not accounted for in the *Baseline* simulation of ROMS-BEC. In order to assess the possible effect of a varying $k_{Fe}$ on the competition between *Phaeocystis* and diatoms, we fit a polynomial function to describe the $k_{Fe}$ of *Phaeocystis* as a function of the light level (VARYING_kFE simulation in Table 2, Fig. A1b, Garcia et al., 2009). Acknowledging the uncertainty in the fit, our model simulates $k_{Fe}$ <0.2 $\mu$mol m$^{-3}$ only at highest light intensities in summer and mostly close to the surface, and 0.2 $\mu$mol m$^{-3}$ < $k_{Fe}$ ≤ 0.26 $\mu$mol m$^{-3}$ elsewhere as a result of low light levels (Fig. S12a & b). While the contribution of *Phaeocystis* to total NPP is only affected to a lesser extent as a consequence (37% and 13% south of 60° S and 30° S, respectively, instead of 46% and 15% in the *Baseline* simulation), the simulated phytoplankton seasonality is impacted substantially. The maximum chlorophyll levels of diatoms occur earlier than those of *Phaeocystis* in many more places of the SO compared to the *Baseline* simulation, both in coastal areas and in the open ocean (Fig. S12c & d). Thus, in order to include light-iron interactions in future modeling efforts with *Phaeocystis* and to assess their impact on the competition of *Phaeocystis* with diatoms throughout the SO, additional measurements are needed for how $k_{Fe}$, but also e.g. $\alpha_{PI}$ and the Fe:C uptake ratio of phytoplankton vary as a function of the surrounding light level. Taken together, given the likely underestimation of the growth advantage of *Phaeocystis* in temperature and at least occasionally in iron in ROMS-BEC, we probably currently underestimate the competitive advantage in growth of *Phaeocystis* relative to diatoms in the model. However, such a potential underestimation in growth advantage does not automatically mean that the contribution of *Phaeocystis* to the phytoplankton community is underestimated as well. This is because of the important role of biomass loss processes to explain why *Phaeocystis* do not outcompete diatoms everywhere in the high latitudes in ROMS-BEC (Fig. 5). Furthermore, the simulated spatio-temporal variability of the high-latitude phytoplankton community structure is in agreement with that suggested by available pigment data (Fig. 2).

Loss processes, such as aggregation and grazing, clearly matter for the competitive advantage of one PFT over another, but these loss processes are generally not well quantified and often not studied with sufficient detail. For example, while the modeling study by Le Quéré et al. (2016) demonstrates the importance of such top-down control for total SO phytoplankton biomass concentrations, an analysis of the impact on phytoplankton community structure is yet to be done. In fact, in the literature, only few studies discuss the role of top-down factors for the relative importance of *Phaeocystis* and diatoms in the high-latitude SO (Granéli et al., 1993; van Hilst and Smith, 2002). Consequently, very little quantitative information exists to constrain model parameters (see section 2.1) or to validate the simulated non-grazing mortality, grazing, or aggregation loss rates of *Phaeocystis* and diatoms over time. In agreement with our results, aggregation has been suggested to be an important process facilitating high POC export when *Phaeocystis* biomass is high (Asper and Smith, 1999; Ducklow et al., 2015; Asper and Smith, 2019), but to what extent this process significantly contributes to the observed relative importance of *Phaeocystis* and diatoms throughout the year in the high-latitude SO remains largely unknown. Certainly, the simulated aggregation rates

in the model and their impact on spatio-temporal distributions of PFT biomass concentrations and rates of NPP are associated with substantial uncertainty due to the immediate conversion of biomass to sinking detritus in the model, the equal treatment of POC originating from all PFTs, the neglect of disaggregation, and due to the calculation of aggregation rates based on the biomass concentrations of individual PFTs rather than all PFTs or even particles combined (see e.g. Turner, 2015). Given that the simulated biomass distributions in ROMS-BEC are most sensitive to differences in parameters describing non-grazing mortality (e.g. viral lysis) and aggregation (Fig. A2 & S11), any changes in these loss processes will significantly impact the relative abundance of *Phaeocystis* and diatoms in the SO. Additionally, as discussed in Nissen et al. (2018), the lack of multiple zooplankton groups in the SO model (Le Quéré et al., 2016) and the parametrization of the single zooplankton grazer using fixed prey preferences and separate grazing on each prey using a Holling Type II function (Holling, 1959), which thus precludes a saturation of feeding at high total phytoplankton biomass, are major limitations of ROMS-BEC. To what extent accounting implicitly for grazing by higher trophic levels in the non-grazing mortality term makes up for not including more zooplankton PFTs remains unclear. Nevertheless, by changing the overall coupling between phytoplankton and zooplankton and through the distinct grazing preferences of the different zooplankton types, the addition of larger zooplankton grazers would likely change the simulated temporal evolution of *Phaeocystis* and diatom biomass in the model (Le Quéré et al., 2016). Therefore, the above mentioned uncertainties should be addressed by future in situ or laboratory measurements in order to better constrain the simulated biomass loss processes, as our findings suggest these to be necessary to explain the seasonal evolution of the relative importance of *Phaeocystis* and diatoms in the high-latitude SO.

## 4.2 Biogeochemical implications of high-latitude SO *Phaeocystis* biogeography

Based on our model results, *Phaeocystis* is a substantial contributor to global NPP and POC export. Comparing the integrated NPP and POC export between 30-90° S in ROMS-BEC with data-based estimates of global NPP and POC export suggests that SO *Phaeocystis* alone contribute about 5% to globally integrated NPP ($58\pm7$ Pg C yr$^{-1}$, Buitenhuis et al., 2013), and about the same percentage to global POC export ($9.1\pm0.2$ Pg C yr$^{-1}$, DeVries and Weber, 2017). Thereby, our simulated contribution of *Phaeocystis* to global NPP is higher than that found in the previous modeling study by Wang and Moore (2011), particularly at higher latitudes, where Wang and Moore (2011) diagnosed a contribution of 23% to NPP south of 60° S ($46\pm21\%$ in ROMS-BEC). We interpret the difference to stem primarily from differences in parameter choices of the PFTs between the two models. For example, the lower ratio of the half-saturation constants of iron of *Phaeocystis* and diatoms in our model (25%; Table 1) as compared to the one in Wang and Moore (2011, 125%) leads to a larger growth advantage of *Phaeocystis* over diatoms in our model. In fact, differences in model parameters between *Phaeocystis* and diatoms in ROMS-BEC can alter the simulated contribution of *Phaeocystis* to total NPP from 5-32% and 17-63% between 30-90° S and 60-90° S, respectively (see section 2.2 and also section A1). This illustrates how single model parameters sensitively impact the competitive success of *Phaeocystis* in the SO. Still, the simulated community structure in the *Baseline* simulation with ROMS-BEC is supported by available observations (see section 3.1), giving us confidence in our estimates.

The simulated contribution of *Phaeocystis* to POC export in ROMS-BEC (16% and 40% south of 30° S and 60° S) is in broad agreement with the previous estimate from Wang and Moore (2011, 19% and 30% south of 40° S and 60° S, respec-

625 tively). This is despite the differences in high-latitude phytoplankton community structure between the two models (see above) and demonstrates our on-going limited quantitative understanding of the fate of biomass losses (see also Laufkötter et al., 2016). Across the parameter sensitivity runs in ROMS-BEC (section 2.2), the contribution of *Phaeocystis* to POC production and export varies from 4-23% and 13-59% south of 30° S and 60° S, respectively. In addition to this uncertainty resulting only from the growth and loss parameters of *Phaeocystis* in the model, further uncertainty arises from parameters describing
the partitioning of biomass losses amongst dissolved and particulate carbon species, which we did not assess in this study. Acknowledging that the exact numbers are highly sensitive to parameter choices in the model, our analysis reveals how the pathways of POC production, in particular the relative importance of fecal pellets from zooplankton and aggregated phytoplankton cells, are impacted by the simulated spatio-temporal variability in phytoplankton community structure throughout the year (Fig. 7). In this regard, the simulated strong temporal coupling between POC fluxes and biomass distributions in ROMS-
BEC is a direct result of the model formulations describing particle sinking (particles sink implicitly, i.e., they are not laterally advected, Lima et al., 2014). This coupling is supported by observations, e.g., from the Ross Sea, where the POC flux from the upper ocean has been found to be closely linked to biomass levels in the overlying surface layer (with aggregates being an important vector for POC export when *Phaeocystis* dominated the community, Asper and Smith, 1999). Yet, the coupling in our model is potentially too strong in other areas, where reprocessing of POC by zooplankton in the upper ocean or lateral
advection of POC could decouple the seasonal evolution of phytoplankton biomass and POC export (e.g. Lam and Bishop, 2007; Stange et al., 2017), the effect of which we can currently not assess. Given the possibly large importance of different POC production pathways for carbon and nutrient cycling through their impact on the remineralization depth of organic matter, these processes should be better constrained in the future, in order to further quantify the imprint of spatio-temporal variations in the relative importance of *Phaeocystis* for the high-latitude cycling of carbon.

Besides its impact on the carbon cycle, *Phaeocystis* is the major contributor to the marine sulphur cycle in the SO through its production of DMSP (Keller et al., 1989; Liss et al., 1994; Stefels et al., 2007). Though not explicitly including the biogeochemical cycling of sulphur, we can nevertheless use model output from ROMS-BEC to obtain an estimate of DMS production by *Phaeocystis* through a simple back-of-the-envelope calculation. Integrating the modeled *Phaeocystis* biomass loss rates via zooplankton grazing and non-grazing mortality over the top 10 m, assuming a molar DMSP:C ratio for *Phaeocystis* of 0.011
(Stefels et al., 2007), and a DMSP-to-DMS conversion efficiency between 0.2-0.7 (the DMS yield depends on the local sulphur demand of bacteria, Stefels et al., 2007; Wang et al., 2015), our estimated annual DMS production by *Phaeocystis* in ROMS-BEC amounts to 3.3-11.5 Tg S and 1.8-6.4 Tg S south of 30° S and 60° S, respectively. Consequently, assuming that all of this DMS production quickly escapes to the atmosphere, our estimates correspond to 11.6-40.1% (30-90° S) and 6.5-22.7% (60-90° S) of the global flux of DMS to the atmosphere previously estimated by Lana et al. (2011, 28.1 Tg S yr$^{-1}$). Our estimate
is an upper bound, however, as not all DMS produced in seawater is readily released to the atmosphere. In fact, a fraction is likely broken down by bacteria, by photolysis, or is mixed down in the water column (see e.g. Simó and Pedrós-Alló, 1999; Stefels et al., 2007). Still, given that other phytoplankton types also produce DMS(P) (Keller et al., 1989; Stefels et al., 2007), the ROMS-BEC-based contribution of SO *Phaeocystis* alone (3.3-11.5 Tg S yr$^{-1}$) to the global flux of DMS to the atmosphere

is in agreement with the flux suggested in Lana et al. (2011, 8.1 Tg S yr$^{-1}$ south of 30° S, i.e., 29% of their global estimate),
and the substantial contribution of SO *Phaeocystis* underpins its major role for the global cycling of sulphur.

### 4.3 Limitations & Caveats

Our results may be affected by several shortcomings regarding the parameterization of *Phaeocystis*, in particular the representation of its life cycle, the fate of its biomass losses, the temperature and light limitation of its growth, and its nutrient uptake stoichiometry. We considered here only colonial *Phaeocystis*, thereby implicitly assuming that a seed population of solitary cells is always available for colony formation. Not including an explicit parameterization for single cells and hence life cycle transitions might substantially impact both the seasonal *Phaeocystis* biomass evolution and the competition with diatoms, as solitary cells have been proposed to require less iron (Veldhuis et al., 1991) and are possibly subject to higher loss rates due to e.g. zooplankton grazing compared to colonies (Smith et al., 2003; Nejstgaard et al., 2007). The transition from solitary to colonial cells is a function of the seed population and light and nutrient levels (Verity, 2000; Bender et al., 2018), and transition models have been applied in SO marine ecosystem models (e.g. Popova et al., 2007; Kaufman et al., 2017; Losa et al., 2019). For example, in their higher complexity, self-organizing ecosystem model (Follows et al., 2007), Losa et al. (2019) include both life stages of *Phaeocystis* and two types of diatoms to simulate phytoplankton competition at high SO latitudes. While our model results suggest that this is not required to reproduce the observed SO biogeography of *Phaeocystis* and diatoms in ROMS-BEC, it nevertheless highlights the need for further research on the impact of the chosen marine ecosystem complexity on the modeled biogeochemical fluxes (Ward et al., 2013). To date, the implementation of morphotype transitions of *Phaeocystis* into a basin-wide SO model such as ROMS-BEC is severely hindered by data availability. At the moment, 390 *Phaeocystis* biomass observations are included in the MAREDAT data base south of 30° S, and the distinction between solitary and colonial cells is often difficult (Vogt et al., 2012), impeding the basin-wide model evaluation of both *Phaeocystis* life stages. In addition, colonies of *Phaeocystis* are surrounded by a gelatinous matrix, which contains nutrients and carbon (Schoemann et al., 2005), leading to an underestimation of modeled *Phaeocystis* carbon biomass estimates if not accounting for this mucus (Vogt et al., 2012). In ROMS-BEC, this underestimation is likely small, as <20% of the total *Phaeocystis* biomass is reportedly encorporated into the mucus in the SO (Fig. 9 in Vogt et al., 2012). Nevertheless, through its function as a nutrient storage, the mucus promotes the accumulation of *Phaeocystis* biomass relative to other phytoplankton types when the latter become limited by low nutrient availability. While the gelatinous matrix is additionally thought to prevent grazing, the literature on grazing losses of *Phaeocystis* colonies is non-conclusive (Schoemann et al., 2005). This is possibly a result of the large range of sizes of both *Phaeocystis* and the respective grazers, with smaller zooplankton typically grazing less on *Phaeocystis* colonies than larger zooplankton (see reviews by Schoemann et al., 2005; Nejstgaard et al., 2007). As discussed above, the fate of biomass losses of *Phaeocystis* is still poorly constrained (this applies to all model PFTs, see also Laufkötter et al., 2016). Currently, ROMS-BEC treats POC from all formation pathways equal, i.e., once produced, there is no differentiation between POC originating from diatoms or *Phaeocystis* or from grazing or aggregation. In reality, *Phaeocystis* aggregates might be recycled more readily than those from diatoms. This could reconcile our model results, i.e., the substantial simulated contribution of *Phaeocystis* to POC export at 100 m, with observations which suggest that the contribution of *Phaeocystis* to the POC flux across 200 m

is small (<5%, Gowing et al., 2001; Accornero et al., 2003; Reigstad and Wassmann, 2007). Furthermore, other functional relationships than those used in ROMS-BEC exist to describe the light and temperature dependent growth of *Phaeocystis* (e.g. Moisan and Mitchell, 2018). In comparison to the equations used in ROMS-BEC (see appendix B), the ones suggested by Moisan and Mitchell (2018) lead to generally lower *Phaeocystis* growth rates, especially at PAR$<$50 W m$^{-2}$, suggesting that our biomass estimates at high latitudes and early/late in the season are associated with substantial uncertainty. As iron-light interactions are key for the simulated Fe:C and Chl:C ratios of SO phytoplankton (Buitenhuis and Geider, 2010) and in light of more recent advances regarding our understanding of the adaptation of SO phytoplankton to persisting low levels of light, iron, and temperature (Strzepek et al., 2019), a reassessment of model parametrizations describing phytoplankton growth and photoacclimation is advisable in future work. Ultimately, the C:P and N:P nutrient uptake ratios by *Phaeocystis* and diatoms are higher (147$\pm$26.7 and 19.2$\pm$0.61) and lower (94.3$\pm$20.1 and 9.67$\pm$0.33), respectively (Arrigo et al., 1999, 2000), than those originally suggested by Redfield and currently used in ROMS-BEC (117:16:1 for C:N:P uptake by *Phaeocystis* and diatoms, Anderson and Sarmiento, 1994). Consequently, this suggests that not accounting for the non-Redfield ratios in nutrient uptake by these PFTs leads to an over(under)estimation of carbon fixation per unit of P and hence POC export where/when *Phaeocystis* (diatoms) dominate the phytoplankton community.

## 5 Conclusions

In this modeling study, we present a thorough assessment of the factors controlling the relative importance of SO *Phaeocystis* and diatoms throughout the year and quantify the implications of the spatio-temporal variability in phytoplankton community structure for POC export. In ROMS-BEC, *Phaeocystis* colonies are an important member of the SO phytoplankton community, contributing 15% (16%) to total annual NPP (POC export) south of 30° S. Moreover, their contribution is threefold higher south of 60° S in our model. Given that our results imply a contribution of approximately 5% of SO *Phaeocystis* colonies to total global NPP and POC export, respectively, we recommend the inclusion of an explicit representation of *Phaeocystis* in ecosystem models of the SO. This will allow for a more realistic representation of the SO phytoplankton community structure, in particular the relative importance of silicifying diatoms and non-silicifying phytoplankton, which we here find to significantly impact the simulated high-latitude carbon fluxes and nutrient distributions. Follow-up studies with both regional SO and global marine ecosystem models should more closely assess what the impact of this simulated change in the relative concentrations of silicic acid and nitrate in the high-latitude SO is on subantarctic and low latitude phytoplankton dynamics.

On a basin-scale, we find that the competition of *Phaeocystis* and diatoms is controlled by seasonal differences in temperature and iron availability, but that variations in light levels are critical on a local scale. Yet, our model suggests that the relative importance of *Phaeocystis* and diatoms over a complete annual cycle is ultimately determined by differences in their biomass loss rates (such as zooplankton grazing and aggregation, Le Quéré et al., 2016), which in turn impacts the formation of sinking particles and hence carbon transfer to depth. Despite knowing of the importance of top-down factors for global phytoplankton biomass distributions (Behrenfeld, 2014) and for the formation of sinking particles (e.g. Steinberg and Landry, 2017), model

parameters describing the fate of carbon after its fixation during photosynthesis are still surprisingly uncertain (Laufkötter et al., 2016), complicating the assessment of the role of biomass loss processes in regulating global biogeochemical cycles.

Environmental conditions in the SO have changed considerably in the last million years (see e.g. Martínez-García et al., 2014), as well as during the past decades (Constable et al., 2014), and are projected to change further during this century (IPCC, 2014). These changes will impact the competitive fitness of *Phaeocystis* and diatoms (see e.g. Hancock et al., 2018; Boyd, 2019) and hence affect the entire phytoplankton community with likely repercussions for the entire food web (Smetacek et al., 2004). Consequently, based on our results, future laboratory and modeling studies should assess how uncertainties in marine ecosystem models surrounding e.g. the parameterization of the life cycle of *Phaeocystis* and the fate of biomass losses impact the simulated relative importance of this phytoplankton type and carbon transfer to depth at high SO latitudes. Thereby, such studies will allow us to better constrain how potential future changes in the high-latitude phytoplankton community structure impact global biogeochemical cycles.

*Data availability.* Model data are available upon email request to the first author (cara.nissen@usys.ethz.ch) or in the ETH library archive (available at https://www.research-collection.ethz.ch/handle/20.500.11850/409193, last access: 2 December 2020; Nissen and Vogt, 2020).

## Appendix A:  Evaluating the simulated phytoplankton dynamics in ROMS-BEC

### A1    Sensitivity of *Phaeocystis* biogeography to chosen parameter differences between *Phaeocystis* and diatoms

We assess the sensitivity of the simulated annual mean *Phaeocystis* biogeography to parameter choices by performing a set of sensitivity experiments (competition experiments, runs 1-8 in Table 2). Overall, the simulated surface *Phaeocystis* biomass concentrations change by $\gtrless \pm 50\%$ for each of the experiments in the high-latitude SO (Fig. A2). Between 60-90° S and in the Ross Sea, the largest increases in *Phaeocystis* biomass concentrations are simulated for THETA_N_MAX (+332% and +217%, respectively, Fig. A2b & c) and AGGREGATION (+112% and +96%, respectively, Fig. A2b & c), whereas the strongest decline is simulated for ALPHA$_{PI}$ (-76% and -87%, respectively, Fig. A2b & c). As a response to changes in *Phaeocystis* parameters, diatom biomass changes overall more than that of SP on a basin scale, suggesting *Phaeocystis* is indeed mostly competing with diatoms for resources in the high-latitude SO. Between 60-90° S, the magnitude of change is similar for the experiments TEMPERATURE (-73%), ALPHA$_{PI}$ (-76%), and IRON (+70%), while in the Ross Sea, the response in IRON is substantially smaller (+17%) than that for the other two experiments (-82% and -87% for TEMPERATURE and ALPHA$_{PI}$, respectively; Fig. A2b & c). This supports our findings from section 3.3, namely that the difference in light sensitivity between *Phaeocystis* and diatoms is more important in coastal areas than on a basin scale in controlling the relative importance of *Phaeocystis* for total phytoplankton biomass.

## Appendix B: BEC equations: Phytoplankton growth & loss

Any change in phytoplankton biomass $P$ [mmol C m$^{-3}$] of phytoplankton $i$ ($i \in \{PA, D, C, SP, N\}$) over time is determined by the balance of growth and loss terms:

$$\frac{dP^i}{dt} = \text{Growth} - \text{Loss} \tag{B1}$$

$$= \mu^i \cdot P^i - \gamma^i(P^i) \cdot P^i \tag{B2}$$

$$= \mu^i \cdot P^i - \gamma_g^i(P^i) \cdot P^i - \gamma_m^i \cdot P^i - \gamma_a^i(P^i) \cdot P^i \tag{B3}$$

In the above equation, $\gamma_g$ denotes the loss by zooplankton grazing, $\gamma_m$ the loss by non-grazing mortality, and $\gamma_a$ the loss by aggregation.

### B1 Phytoplankton growth

The specific growth rate $\mu^i$ [day$^{-1}$] of phytoplankton $i$ ($i \in \{D, C, SP, N\}$, i.e., all but *Phaeocystis*) is determined by the maximum growth rate $\mu_{max}^i$ (Table 1) and modifications due to temperature (T), nutrients (N) and irradiance (I), following:

$$\mu^i = \mu_{max}^i \cdot f^i(T) \cdot g^i(N) \cdot h^i(I) \tag{B4}$$

The temperature function $f(T)$ is an exponential function, which is modified by the constant $Q_{10}$ specific to every phytoplankton $i$ (Table 1):

$$f^i(T) = Q_{10}^i \cdot \exp\left(\frac{T - T_{ref}}{10°C}\right) \tag{B5}$$

Note that for *Phaeocystis* in ROMS-BEC, an optimum temperature function is used (Eq. 1), as this PFT is parametrized to only represent *Phaeocystis antarctica* in the SO application of this study (see section 2.1).

First, the limitation of growth of phytoplankton $i$ ($i \in \{PA, D, C, SP, N\}$) by the surrounding nutrient $L^i(N)$ is calculated individually for each nutrient (nitrogen, phosphorus, iron for all phytoplankton, silicate for diatoms only) following a Michaelis-Menten function (see Table 1 for half-saturation constants $k_N^i$). Accordingly, the limitation factor is calculated as follows for iron (Fe) and silicate (SiO3):

$$L^i(N) = \frac{N}{N + k_N^i} \tag{B6}$$

For nitrogen and phosphorus, the combined limitation by nutrient $N$ and $M$ (nitrate (NO3) and ammonium (NH4) for nitrogen, phosphate (PO4) and dissolved organic phosphorus (DOP) for phosphorus) is accounted for following:

$$L^i(N, M) = \frac{N}{k_N^i + N + M \cdot (k_N^i/k_M^i)} + \frac{M}{k_M^i + M + N \cdot (k_M^i/k_N^i)} \tag{B7}$$

In the model, the phytoplankton growth rate is then only limited by the most limiting nutrient:

$$g^i(N) = \min(L^i(NO3, NH4), L^i(PO4, DOP), L^i(Fe), L^i(SiO3)) \tag{B8}$$

The light limitation function $h^i(I)$ includes the effects of photoacclimation by including the chlorophyll-to-carbon ratio $\theta_{\text{chl:C}}^i$ and the growth of the respective phytoplankton $i$ ($i \in \{PA, D, C, SP, N\}$) limited by nutrients and temperature:

$$h^i(I) = 1 - \exp(-1 \cdot \frac{\alpha_{PI}^i \cdot \theta_{\text{chl:C}}^i \cdot I}{\mu_{\max}^i \cdot g^i(N) \cdot f^i(T)}) \tag{B9}$$

Here, same as in Nissen et al. (2018), growth by coccolithophores is set to zero at PAR levels $<1$ W m$^{-2}$ (Zondervan, 2007) and is linearly reduced at temperatures $<6°$C following:

$$\mu^C = \mu^C \cdot \frac{\max(T + 2°C), 0)}{8°C} \tag{B10}$$

Coccolithophore calcification amounts to 20% of their photosynthetic growth at any location and point in time in ROMS-BEC.

Diazotroph growth is zero at temperatures $<14°$C.

In BEC, the Fe:C ratio $\theta_{\text{Fe:C}}^i$ [$\mu$mol mol$^{-3}$] of growth by phytoplankton $i$ varies between the maximum Fe:C ratio $\theta_{\text{Fe:C,max}}^i$ at high seawater Fe concentrations and the minimum Fe:C ratio $\theta_{\text{Fe:C,min}}^i$ at very low Fe concentrations. Below a critical surrounding Fe concentration, which depends on each PFT's half-saturation constant of iron $k_{\text{Fe}}^i$ (see Table 1), the ratio is reduced from the maximum Fe:C ratio following:

$$\theta_{\text{Fe:C}}^i \quad = \theta_{\text{Fe:C,max}}^i \tag{B11}$$

$$\theta_{\text{Fe:C}}^i \quad = \max(\theta_{\text{Fe:C}}^i \cdot \frac{[\text{Fe}]}{9 \cdot k_{\text{Fe}}^i}, \theta_{\text{Fe:C,min}}^i) \qquad \text{where } [\text{Fe}] < 9 \cdot k_{\text{Fe}}^i \tag{B12}$$

For this study, $\theta_{\text{Fe:C,max}}^i$ is 60 for diazotrophs and 20 for all other PFTs, and $\theta_{\text{Fe:C,min}}^i$ is 12 for diazotrophs and 3 for all other PFTs.

## B2   Phytoplankton loss

In ROMS-BEC, the corrected phytoplankton biomass $P'^i$ is used to compute loss rates of phytoplankton biomass, to prevent phytoplankton biomass loss at very low biomass levels:

$$P'^i = \max(P^i - c_{\text{loss}}^i, 0) \tag{B13}$$

In this equation, $c_{\text{loss}}^i$ is the threshold of phytoplankton biomass $P^i$ below which no losses occur ($c_{\text{loss}}^N$=0.022 mmol C m$^{-3}$ and $c_{\text{loss}}^{PA,D,C,SP}$=0.04 mmol C m$^{-3}$).

The single zooplankton grazer $Z$ [mmol C m$^{-3}$] feeds on the respective phytoplankton $P'^i$ [mmol C m$^{-3}$] at a grazing rate $\gamma_g^i$ [mmol C m$^{-3}$ day$^{-1}$] that is given by:

$$\gamma_g^i = \gamma_{\max}^i \cdot f^Z(T) \cdot Z \cdot \frac{P'^i}{z_{\text{grz}}^i + P'^i} \tag{B14}$$

with

$$f^Z(T) = 1.5 \cdot \exp(\frac{T - T_{\text{ref}}}{10°C}) \tag{B15}$$

The non-grazing mortality rate $\gamma_m^i$ [mmol C m$^{-3}$ day$^{-1}$] of phytoplankton $i$ [mmol C m$^{-3}$] is the product of a maximum mortality rate $m_0^i$ [day$^{-1}$] scaled by the temperature function $f^i(T)$ with the modified phytoplankton biomass $P'^i$:

$$\gamma_m^i = m_0^i \cdot f^i(T) \cdot P'^i \tag{B16}$$

with $m_0^i$ being 0.15 day$^{-1}$ for diazotrophs and 0.12 day$^{-1}$ for all other phytoplankton.

Phytoplankton $P'^i$ [mmol C m$^{-3}$] aggregate at an aggregation rate $\gamma_a^i$ [mmol C m$^{-3}$ day$^{-1}$] which is computed with the quadratic mortality rate constants $\gamma_{a,0}^i$ ([m$^3$ (mmol C)$^{-1}$ d$^{-1}$], Table 1) and :

$$\gamma_a^i \quad = \min(\gamma_{a,max}^i \cdot P'^i, \gamma_{a,0}^i \cdot P'^i \cdot P'^i) \tag{B17}$$

$$\gamma_a^i \quad = \max(\gamma_{a,min}^i \cdot P'^i, \gamma_a^i) \tag{B18}$$

In ROMS-BEC, $\gamma_{a,min}^i$ is 0.01 day$^{-1}$ for small phytoplankton and coccolithophores and 0.02 day$^{-1}$ for *Phaeocystis* and diatoms, and with $\gamma_{a,max}^i$ being 0.9 day$^{-1}$ for *Phaeocystis*, diatoms, coccolithophores, and small phytoplankton. Note that phytoplankton immediately stop photosynthesizing upon aggregation and that aggregation losses do not occur for diazotrophs 800 in ROMS-BEC.

*Author contributions.* MV and CN conceived the study. CN set up the model simulations, performed the analysis, and wrote the paper. MV contributed to the interpretation of the results and the writing of the paper.

*Competing interests.* The authors declare that they have no conflict of interest.

*Acknowledgements.* We acknowledge all the scientists who contributed phytoplankton and zooplankton cell count data to the MAREDAT 805 initiative and William Balch, Helen Smith, Mariem Saavedra-Pellitero, Gustaaf Hallegraeff, José-Abel Flores, and Alex Poulton for providing additional cell count data. Furthermore, GlobColour data (http://globcolour.info) used in this study has been developed, validated, and distributed by ACRI-ST, France. We would like to thank Nicolas Gruber, Matthias Münnich, and Domitille Louchard for valuable discussions and Damian Loher for technical support. Additionally, we would like to thank Gianna Ferrari for the analysis of early ROMS-BEC simulations with *Phaeocystis*. Ultimately, we thank four reviewers for their valuable reviews and comments, which have improved the quality of the 810 manuscript. This research was financially supported by the Swiss Federal Institute of Technology Zürich (ETH Zürich) and the Swiss National Science Foundation (project SOGate, grant no. 200021_153452). The simulations were performed at the HPC cluster of ETH Zürich, Euler, which is located in the Swiss Supercomputing Center (CSCS) in Lugano and operated by ETH ITS Scientific IT Services in Zürich. Model output is available upon request to the corresponding author, Cara Nissen (cara.nissen@usys.ethz.ch).

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

1190

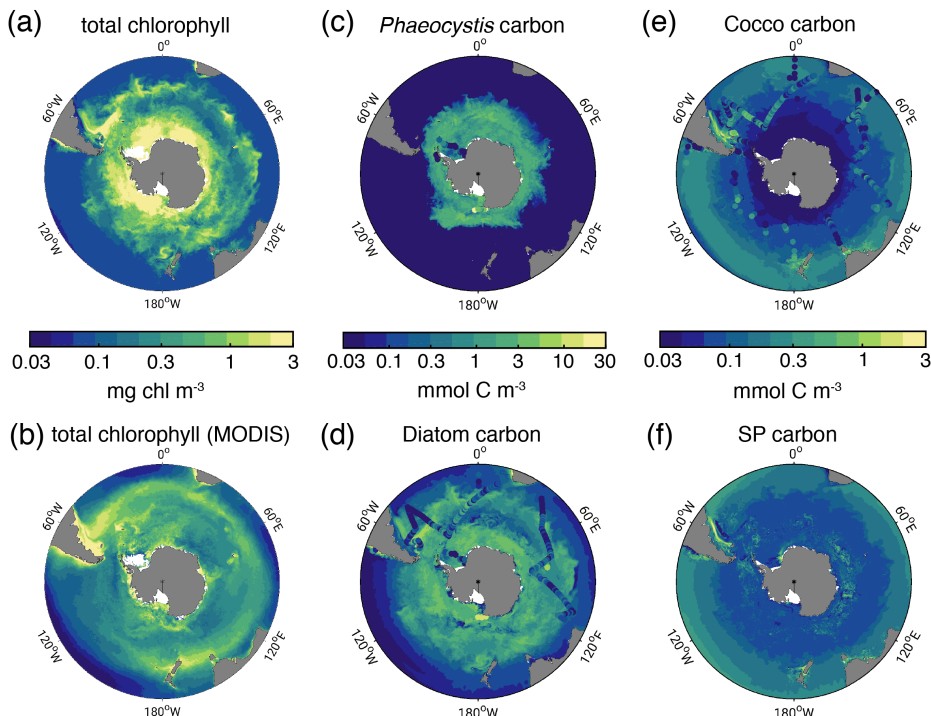

**Figure 1.** Biomass distributions for December-March (DJFM). Total surface chlorophyll [mg chl m$^{-3}$] in a) ROMS-BEC and b) MODIS-Aqua climatology (NASA-OBPG, 2014a), using the chlorophyll algorithm by Johnson et al. (2013). c)-f) Mean top 50 m c) *Phaeocystis*, d) diatom, e) coccolithophore, and f) small phytoplankton carbon biomass concentrations [mmol C m$^{-3}$] in ROMS-BEC. *Phaeocystis*, diatom, and coccolithophore biomass observations from the top 50 m are indicated by colored dots in c), d), and e), respectively (Balch et al., 2016; Saavedra-Pellitero et al., 2014; O'Brien et al., 2013; Vogt et al., 2012; Leblanc et al., 2012; Tyrrell and Charalampopoulou, 2009; Gravalosa et al., 2008; Cubillos et al., 2007). For more details on the biomass evaluation, see Nissen et al. (2018).

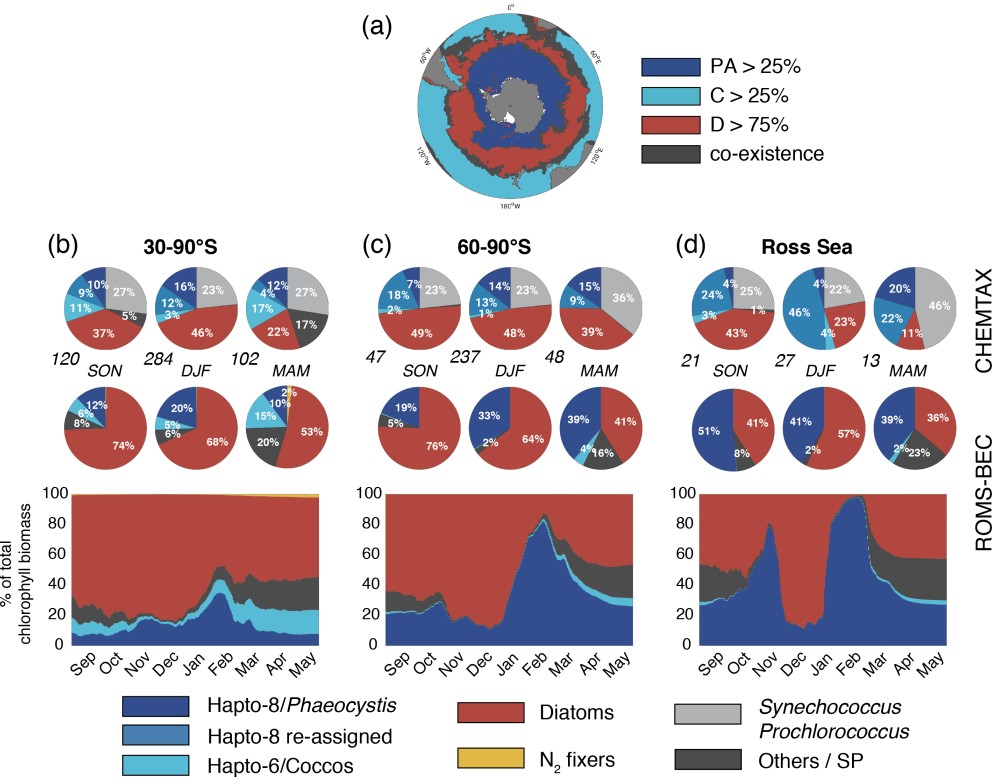

**Figure 2.** Spatio-temporal distribution of phytoplankton communities in the SO. a) Diatom-dominated phytoplankton community vs. mixed communities with substantial contributions of *Phaeocystis*, coccolithophores and small phytoplankton in ROMS-BEC. Communities in which neither *Phaeocystis* (PA, dark blue) or coccolithophores (C, light blue) contribute >25 % nor diatoms (D, red) contribute >75 % to total annual NPP are classified as co-existence communities (grey). b)-d) Relative contribution of the five phytoplankton PFTs to total chlorophyll biomass [mg chl m$^{-3}$] for b) 30-90° S, c) 60-90° S, and d) the Ross Sea. The top pie charts denote the climatological mixed layer average community composition suggested by CHEMTAX analysis of HPLC pigments for spring, summer, and fall, respectively (the total number of available observations for a given region and season is given at the lower left side, Swan et al., 2016), and the lower pie charts denote the corresponding community structure in the top 50 m in ROMS-BEC. Note that the categories in the CHEMTAX analysis are not 100% equivalent to the model PFTs. Here, "others" in the CHEMTAX fractions corresponds to dinoflagellates, cryptophytes, and chlorophytes, and "Hapto-8 reassigned" corresponds to the contribution of Hapto-6 where the temperature is <2° C (see also section 2.3.1). The panels at the bottom denote the daily contribution of each PFT in ROMS-BEC to total surface chlorophyll biomass.

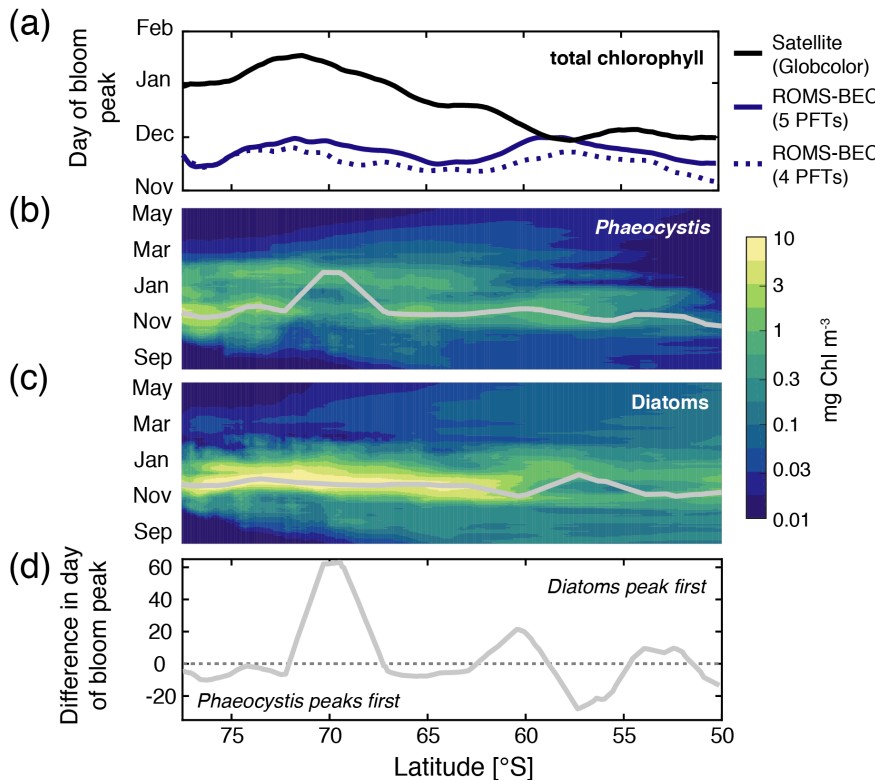

**Figure 3.** Hovmoller plots south of 50° S of a) the day of maximum total chlorophyll concentrations in a satellite product (black line, Globcolor climatology from 1998-2018 based on the daily 25 km chlorophyll product, see Fanton d'Andon et al., 2009; Maritorena et al., 2010), the *Baseline* simulation of this study (solid blue line), and the *Baseline* simulation of Nissen et al. (2018, dashed blue line; without *Phaeocystis*), and daily surface b) diatom and c) *Phaeocystis* chlorophyll biomass concentrations [mg chl m$^{-3}$]. Overlain are the average day of the peak concentrations for each latitude (see also section 2.3.1). Panel d) denotes the difference in days in the timing of the bloom peak of diatoms and *Phaeocystis* for each latitude, with negative values denoting a succession from *Phaeocystis* to diatoms throughout the season.

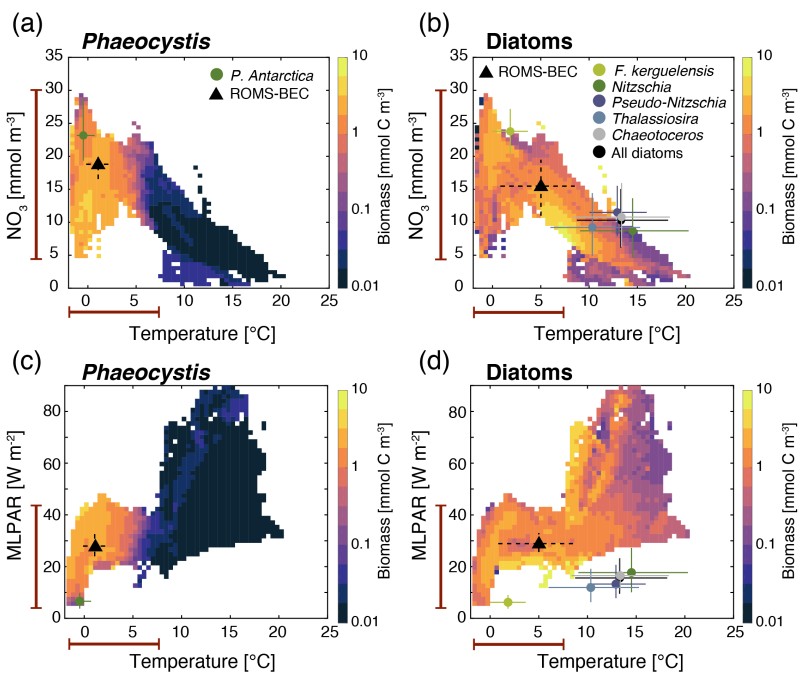

**Figure 4.** Simulated DJFM average top 50 m average a) & c) *Phaeocystis* and b) & d) diatom carbon biomass concentrations (mmol C m$^{-3}$) south of 40° S as a function of the simulated temperature (° C) and a)-b) nitrate concentrations (mmol N m$^{-3}$) and c)-d) mixed layer PAR levels (W m$^{-2}$). Overlain are the observed ecological niche centers (median) and breadths (inter quartile ranges) for example taxa of the two functional types from Brun et al. (2015, circles and solid lines) and as simulated in ROMS-BEC (triangles and dashed lines; area and biomass weighted). The red bars on the axes indicate the simulated range of the respective environmental condition in ROMS-BEC between 60-90° S and averaged over DJFM and the top 50 m.

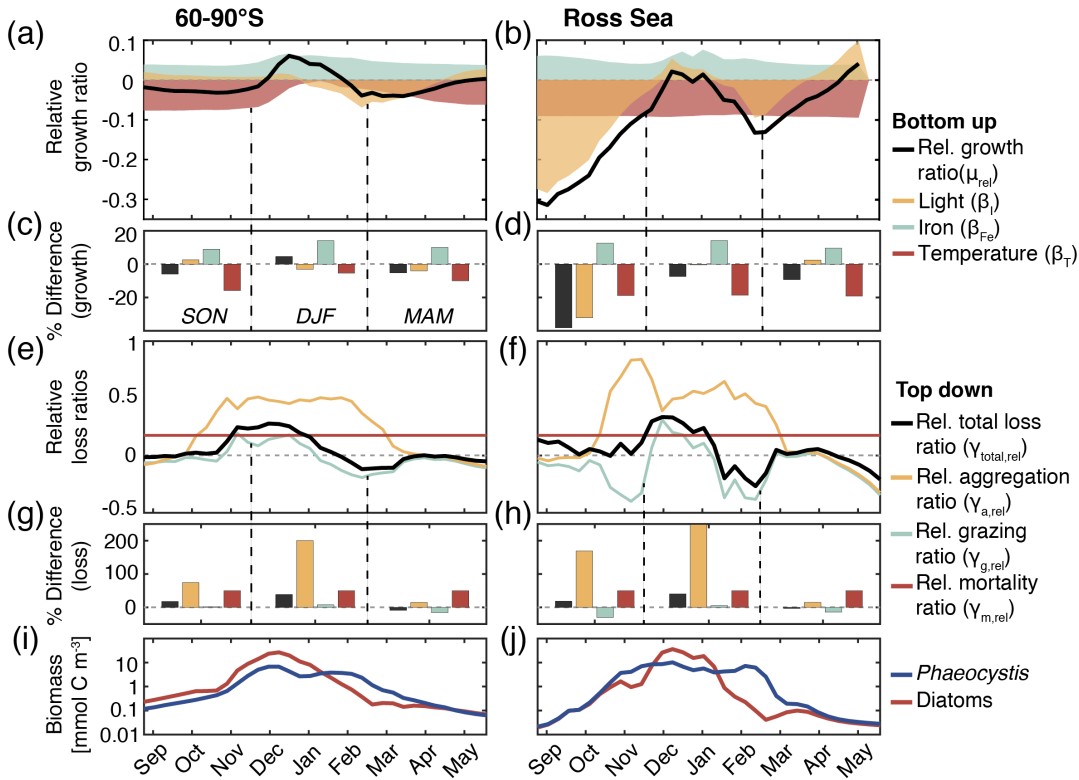

**Figure 5.** a) & b) Relative growth ratio (black) of diatoms vs. *Phaeocystis*. The colored areas are the contributions of the limitation of growth by light (yellow, $\beta_I$), iron (blue, $\beta_{Fe}$), and temperature (red, $\beta_T$, see Eq. 2). c) & d) Seasonally averaged percent difference between diatoms and *Phaeocystis* in the specific growth rate (black), light limitation (yellow), iron limitation (blue), and temperature limitation (red). Calculated from non-log-transformed ratios, i.e., e.g. black bar corresponds to $10^{\mu_{rel}^{DPA}}$ (see Eq. 2). e) & f) Relative total loss ratio (black) of diatoms vs. *Phaeocystis*, with contributions of the relative grazing ratio (blue), relative non-grazing loss ratio (red), and relative aggregation ratio (yellow, see Eq. 3-6). g) & h) Seasonally averaged percent difference between diatoms and *Phaeocystis* in the total specific loss rate (black), specific aggregation rate (yellow), specific grazing rate (blue), and specific mortality rate (red), calculated from non-log-transformed ratios. i) & j) *Phaeocystis* (blue) and diatom (red) surface carbon biomass concentrations [mmol C m$^{-3}$]. For all metrics, the left panels are surface averages over 60-90° S and those on the right for the Ross Sea.

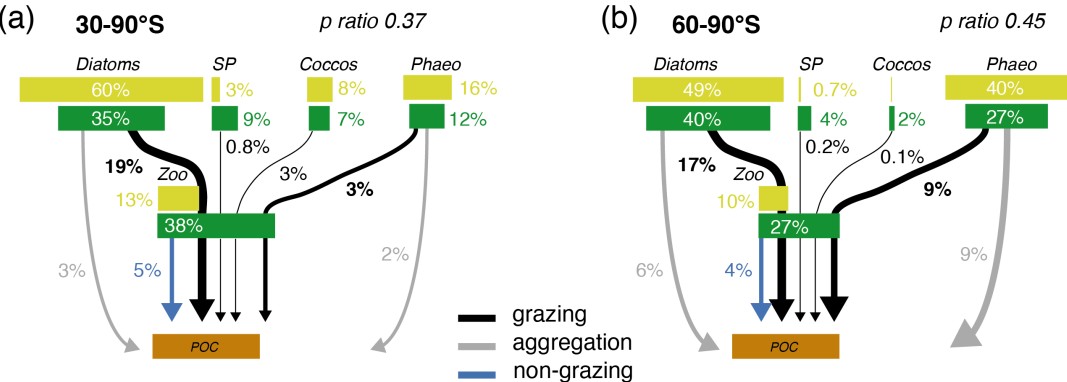

**Figure 6.** Pathways of particulate organic carbon (POC) formation in the *Baseline* simulation of ROMS-BEC averaged annually over a) 30-90° S and b) 60-90° S. The results for the Ross Sea are comparable to those between 60-90° S (see Fig. S11). The green and yellow boxes show the relative contribution (%) of *Phaeocystis*, diatoms, coccolithophores, small phytoplankton (SP), and zooplankton (Zoo) to the combined phytoplankton and zooplankton biomass (green) and total POC production (yellow) in the top 100 m, respectively. The arrows denote the relative contribution of the different POC production pathways associated with each PFT (black = grazing by zooplankton, grey = aggregation, blue = non-grazing mortality), given as % of total NPP in the top 100 m. Numbers are printed if ≥0.1% and rounded to the nearest integer if >1%. The sum of all arrows gives the POC production efficiency, i.e., the fraction of NPP which is converted into sinking POC upon biomass loss (p ratio). Note that diazotrophs are not included in this figure due to their minor contribution to NPP in the model domain.

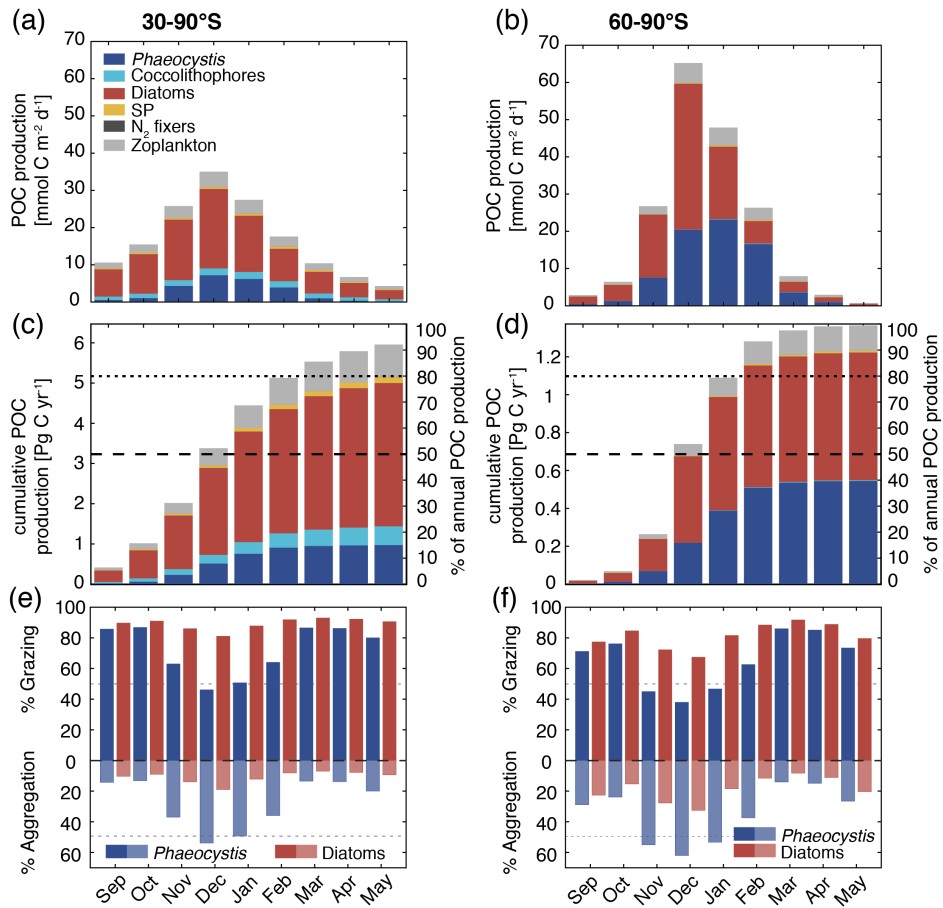

**Figure 7.** Simulated vertically integrated production of particulate organic carbon (POC) a) & b) as a function of time [mmol C m$^{-2}$ d$^{-1}$], c) & d) cumulative over time (absolute production in Pg C yr$^{-1}$ on the left axis and relative to annually integrated production on the right axis), and e) & f) as a function of time via grazing and aggregation, respectively. The colors correspond to the different PFTs in ROMS-BEC, and the panels correspond to averages or integrals over 30-90° S (left) and 60-90° S (right), respectively. The results for the Ross Sea are comparable for those between 60-90° S (see Fig. S11).

**Table 1.** BEC parameters controlling phytoplankton growth and loss for the five phytoplankton PFTs diatoms (D), *Phaeocystis* (PA), coc-colithophores (C), small phytoplankton (SP), and diazotrophs (DZ). Z=zooplankton, P=phytoplankton, PI=photosynthesis-irradiance. If not given in section 2.1, the model equations describing phytoplankton growth and loss rates are given in Nissen et al. (2018).

| Parameter | Unit | Description | D | PA | C | SP | DZ[†] |
|---|---|---|---|---|---|---|---|
| $\mu_{max}$ | $d^{-1}$ | max. growth rate at 30° C | 4.6 | ‡ | 3.8 | 3.6 | 0.9 |
| $Q_{10}$ | | temperature sensitivity | 1.55 | ‡ | 1.45 | 1.5 | 1.5 |
| $k_{NO3}$ | mmol m$^{-3}$ | half-saturation constant for $NO_3$ | 0.5 | 0.5 | 0.3 | 0.1 | 1.0 |
| $k_{NH4}$ | mmol m$^{-3}$ | half-saturation constant for $NH_4$ | 0.05 | 0.05 | 0.03 | 0.01 | 0.15 |
| $k_{PO4}$ | mmol m$^{-3}$ | half-saturation constant for $PO_4$ | 0.05 | 0.05 | 0.03 | 0.01 | 0.02 |
| $k_{DOP}$ | mmol m$^{-3}$ | half-saturation constant for DOP | 0.9 | 0.9 | 0.3 | 0.26 | 0.09 |
| $k_{Fe}$ | $\mu$mol m$^{-3}$ | half-saturation constant for Fe | 0.15 | 0.2 | 0.125 | 0.1 | 0.5 |
| $k_{SiO3}$ | mmol m$^{-3}$ | half-saturation constant for $SiO_3$ | 1.0 | - | - | - | - |
| $\alpha_{PI}$ | $\frac{\text{mmol C m}^2}{\text{mg Chl W s}}$ | initial slope of PI-curve | 0.44 | 0.63 | 0.4 | 0.44 | 0.38 |
| $\theta_{chl:N, max}$ | $\frac{\text{mg chl}}{(\text{mmol N})^{-1}}$ | max. Chl:N ratio | 4.0 | 2.5 | 2.5 | 2.5 | 2.5 |
| $\gamma_{g,max}$ | $d^{-1}$ | max. growth rate of Z grazing on P | 3.8 | 3.6 | 4.4 | 4.4 | 3.0 |
| $z_{grz}$ | mmol m$^{-3}$ | half-saturation constant for ingestion | 1.0 | 1.0 | 1.05 | 1.05 | 1.2 |
| $\gamma_{m,0}$ | $d^{-1}$ | linear non-grazing mortality | 0.12 | 0.18 | 0.12 | 0.12 | 0.15 |
| $\gamma_{a,0}$ | $\frac{\text{m}^3}{\text{mmol C d}}$ | quadratic loss rate in aggregation | 0.001 | 0.005 | 0.001 | 0.001 | - |
| $r_g$ | - | fraction of grazing routed to POC | 0.42 | 0.3 | 0.2 | 0.05 | 0.05 |

[†] Compared to Nissen et al. (2018), the $k_{Fe}$ of diazotrophs in ROMS-BEC is higher than for all other PFTs, consistent with literature reporting high Fe requirements of *Trichodesmium* (Berman-Frank et al., 2001). Furthermore, the maximum grazing rate on diazotrophs is lowest in the model (Capone, 1997). Still, diazotrophs continue to be a minor player in the SO phytoplankton community, contributing <1% to domain-integrated NPP in ROMS-BEC.

[‡] The temperature-limited growth rate of *Phaeocystis* is calculated based on an optimum function according to Eq. 1 (see also Fig. A1a).

**Table 2.** Overview of sensitivity experiments aiming to 1) assess the sensitivity of the simulated *Phaeocystis*-diatom competition to chosen parameter values and parameterizations of *Phaeocystis* (competition experiments, runs 1-8) and 2) assess the sensitivity of the simulated biomass distributions to chosen *Phaeocystis* parameter values (parameter sensitivity experiments, runs 9-22). The results of the parameter sensitivity experiments are discussed in the supplementary material. See Table 1 and section 2.1 for parameter values and parameterizations of *Phaeocystis* in the reference simulation. PA=*Phaeocystis*, D=diatoms.

| Competition | Run Name | Description |
|---|---|---|
| 1 | TEMPERATURE | Use $\mu_{max}^D$, $Q_{10}^D$, and $\mu_T^{PA} = \mu_{max}^D \cdot Q_{10}^{D}{}^{\frac{T-T_{ref}}{10^\circ C}}$ to compute the temperature-limited growth rate of *Phaeocystis* instead of Eq. 1 |
| 2 | ALPHA$_{PI}$ | Set $\alpha_{PI}^{PA}$ to $\alpha_{PI}^D$ |
| 3 | IRON | Set $k_{Fe}^{PA}$ to $k_{Fe}^D$ |
| 4 | GRAZING | Set $\gamma_{g,max}^{PA}$ to $\gamma_{max}^D$ |
| 5 | AGGREGATION | Set $\gamma_{a,0}^{PA}$ to $\gamma_{a,0}^D$ |
| 6 | MORTALITY | Set $\gamma_{m,0}^{PA}$ to $\gamma_{m,0}^D$ |
| 7 | THETA_N_MAX | Set $\theta_{chl:N,\,max}^{PA}$ to $\theta_{chl:N,\,max}^D$ |
| 8 | VARYING_kFE | Use $k_{Fe}^{PA}(I) = 2.776 \cdot 10^{-5} \cdot (I+20)^2$ - $0.00683 \cdot (I+20) + 0.46$ (with the irradiance $I$ in W m$^{-2}$) instead of a constant $k_{Fe}^{PA}$ |

| Parameter sensitivity | Run Name | Description |
|---|---|---|
| 9 | Topt150 | Increase $T_{opt}^{PA}$ by 50% |
| 10 | Topt50 | Decrease $T_{opt}^{PA}$ by 50% |
| 11 | kFe150 | Increase $k_{Fe}^{PA}$ by 50% |
| 12 | kFe50 | Decrease $k_{Fe}^{PA}$ by 50% |
| 13 | alphaPI150 | Increase $\alpha_{PI}^{PA}$ by 50% |
| 14 | alphaPI50 | Decrease $\alpha_{PI}^{PA}$ by 50% |
| 15 | mortality150 | Increase $\gamma_{m,0}^{PA}$ by 50% |
| 16 | mortality50 | Decrease $\gamma_{m,0}^{PA}$ by 50% |
| 17 | aggregation150 | Increase $\gamma_{a,0}^{PA}$ by 50% |
| 18 | aggregation50 | Decrease $\gamma_{a,0}^{PA}$ by 50% |
| 19 | grazing150 | Increase $\gamma_{g,max}^{PA}$ by 50% |
| 20 | grazing50 | Decrease $\gamma_{g,max}^{PA}$ by 50% |
| 21 | thetaNmax150 | Increase $\theta_{chl:N,\,max}^{PA}$ by 50% |
| 22 | thetaNmax50 | Decrease $\theta_{chl:N,\,max}^{PA}$ by 50% |

**Table 3.** Comparison of ROMS-BEC based phytoplankton biomass, production, and export estimates with available observations (given in parentheses). Data sources are given below the Table. The reported uncertainty of the contribution of the PFTs to the simulated integrated NPP corresponds to the area-weighted spatial variability of each PFT's contribution to annual NPP ($1\sigma$ in space).

| | | ROMS-BEC (Data) | |
| | | 30-90° S | 60-90° S |
| --- | --- | --- | --- |
| Surface chlorophyll biomass | total, annual mean [Gg chl] | 40.8 (34.5[a]) | 17.1 (9.5[a]) |
| Diatom carbon biomass | 0-200m, annual mean [Pg C] | 0.059 (global[b]: 0.10-0.94) | 0.015 |
| *Phaeocystis* carbon biomass | 0-200m, annual mean [Pg C] | 0.019 (global[b]: 0.11-0.71) | 0.010 |
| Coccolithophore carbon biomass | 0-200m, annual mean [Pg C] | 0.012 (global[b]: 0.001-0.03) | 0.001 |
| NPP | Pg C yr$^{-1}$ | 17.2 (12.1-12.5[c]) | 3.0 (0.68-1.7[c]) |
| | Diatoms [%] | 52.0 ($\pm$26.2) | 49.1 ($\pm$19.9) |
| | *Phaeocystis* [%] | 15.3 ($\pm$24.5) | 45.8 ($\pm$20.7) |
| | Coccolithophores [%] | 14.6 ($\pm$15.3) | 0.7 ($\pm$1.0) |
| | SP [%] | 17.2 ($\pm$16.1) | 4.5 ($\pm$1.9) |
| POC export at 100m | Pg C yr$^{-1}$ | 3.1 (2.3-2.96[d]) | 0.62 (0.21-0.24[d]) |

[a] Monthly climatology from MODIS Aqua (2002-2016, NASA-OBPG, 2014a), SO algorithm (Johnson et al., 2013)

[b] The reported estimates from the MAREDAT data base in Buitenhuis et al. (2013) are global estimates of phytoplankton biomass.

[c] Monthly climatology from MODIS Aqua VGPM (2002-2016, Behrenfeld and Falkowski, 1997; O'Malley, last access: 16 May 2016), NPP climatology from Buitenhuis et al. (2013, 2002-2016)

[d] Monthly output from a biogeochemical inverse model (Schlitzer, 2004) and a data-assimilated model (DeVries and Weber, 2017).

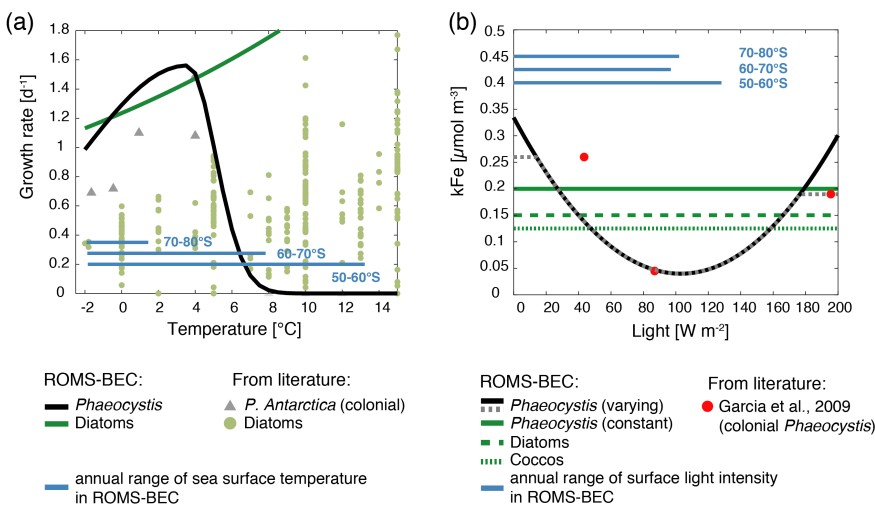

**Figure A1.** a) Growth rates of *Phaeocystis antarctica* colonies as a function of temperature (conditions of nutrients and light are non-limiting) in laboratory data (grey triangles, see compilation by Schoemann et al., 2005) and as used in ROMS-BEC (black line, see Eq. 1). Green circles and the green line show the temperature-limited growth rate of diatoms in laboratory data (see compilation by Le Quéré et al., 2016) and as used in ROMS-BEC, respectively (see also Table 1). b) Half-saturation constant of Fe ($k_{Fe}$) of *Phaeocystis* as a function of light intensity $I$ (W m$^{-2}$) in laboratory data (red circles) and the polynomial fit ($k_{Fe}^{PA}(I) = 2.776 \cdot 10^{-5} \cdot (I + 20)^2 - 0.00683 \cdot (I + 20) + 0.46$) without (black) and with (dashed grey, as used in ROMS-BEC in simulation VARYING_kFe, see Table 2) the correction at low and high light intensities to restrict $k_{Fe}$ to the range measured in the laboratory experiments by Garcia et al. (2009). The green lines correspond to the half-saturation constants used for *Phaeocystis* (solid), diatoms (dashed), and coccolithophores (dotted) in the *Baseline* simulation in this study (see Table 1). In both panels, the blue lines correspond to the simulated annual range in a) sea surface temperature [° C] and b) light intensity [W m$^{-2}$] between 50-60° S, 60-70° S, and 70-80° S, respectively.

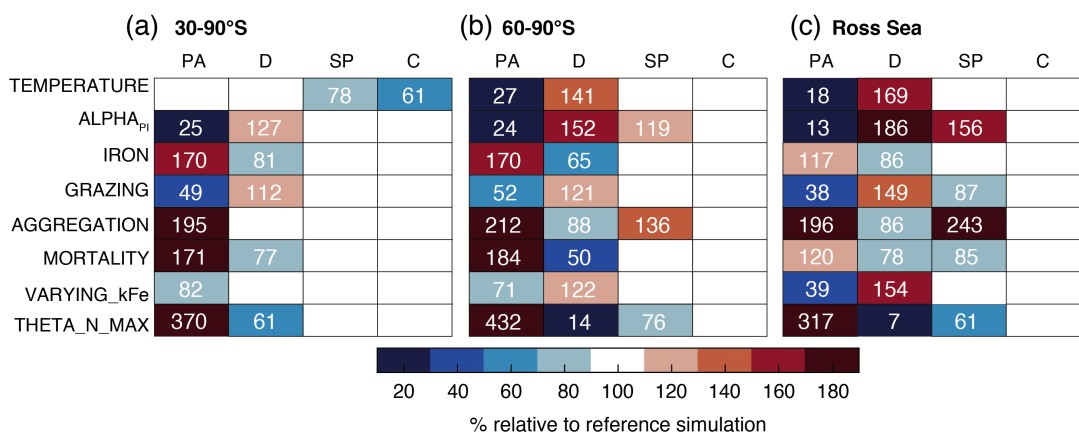

**Figure A2.** Annual mean surface chlorophyll concentrations of *Phaeocystis* (PA), diatoms (D), small phytoplankton (SP), and coccolitho-phores (C) in the competition sensitivity simulations (see section 2.2 and runs 1-8 in Table 2) relative to the *Baseline* simulation. The model output is averaged over a) 30-90° S, b) 60-90° S, and c) the Ross Sea. Numbers are only printed if the relative change exceeds ±10%