# Peer review of "Factors controlling the competition between *Phaeocystis* and diatoms in the Southern Ocean and implications for carbon export fluxes"

_Biogeosciences, 2019_

## Referee Comment (RC1) · Anonymous Referee #1 · 26 Feb 2020

**General Comments**

Nissen and Vogt present a model study on the relative importance of the colonial form of *Phaeocystis* for ecosystem processes and biogeochemical fluxes; they evaluate their results with observations from different data sources. A comparable study (Nissen et al 2018) had been performed with a focus on coccolithophores instead of *Phaeocystis* with similar analyses. In that respect this work is not overly innovative nor are original ideas presented. More critical is, however, that there is no thread in this manuscript; a clear goal is missing. A number of topics (e.g. phenology, competition, carbon and DMS-fluxes) are touched but not thoroughly permeated. It is unclear whether the au-

thors would like to study the success of *Phaeocystis* compared to other phytoplankton functional groups or the importance of *Phaeocystis* for carbon export fluxes. Either way, no comprehensible motivation for either of these broad themes is provided. Some aspects of the methodology also need to be revised with consequences for the model analyses. Last but not least, recent work on this topic has been ignored. Overall this manuscript is premature and the authors must clarify their focus before publication. To sharpen the focus maybe it helps to look at the unpublished, recent modelling work on Southern Ocean *Phaeocystis* and PFTs (Losa et al. 2019) that has been put up for discussion in *Biogeosciences Discussion*.

**Specific Comments**

- title: the title only partly reflects the content of this study

- abstract and entire manuscript: it is unclear which research gap the authors want to fill. What is currently unclear - which open question in this research field are attempted to be answered with ROMS-BEC?

- the manuscript should stand alone. Currently important parts of the model description are missing. The prognostic equation for *Phaeocystis* with all source and sink terms as well as all functional dependencies of rates to environmental drivers need to be provided.

- the newly introduced formulation of the temperature dependent growth for the PFT *Phaeocystis* is fundamentally different from the description of the PFTs of the original BEC model. The former is a "Gauss-like" temperature dependent growth function with a temperature optimum. Any deviation from the optimum is a limitation, varying between 0...1. In contrast, the Q10-approach with different

Q10 values that is applied to the other PFTs denotes the "sensitivity" in the exponential growth towards temperature - in these cases the higher the temperature, the higher the growth. Even if a relatively high reference temperature of 30 degrees Celsius is given (which is likely not reached in the Southern Ocean), there is no such thing as an optimum in the Q10 approach. Thus, the "limitation" values used in the analyses cannot easily be compared.

[Generally the question arises whether the Q10 approach should be applied to PFTs at all. Introduced by Eppley it is valid and a good description for bulk phytoplankton but as soon as the bulk is divided into groups, "Gauss-like functions" with a clear optimum seem to be more adequate.]

- temperature-dependent growth functions of any organism group usually have a negatively skewed thermal reaction norm. This is also true for *Phaeocystis antarctica*. Since there already exists a mathematical description for the temperature-& light-dependent growth function of *Phaeocystis antarctica* (Moisan and Mitchell 2018), I wonder why the authors have not used it. In fact there are more recent observation-based publications on *Phaeocystis antarctica* that may be of interest for this study.

- please specify which atmospheric forcing fields have been used.

- model results: there is a mixture of model results, model evaluation, model comparison with results from previous experiments which makes it difficult to read and to follow the arguments; the entire results section needs to be revised.

  – the sections about the ecological niches, bottom-up and top-down effects are tedious to read and questionable with respect to temperature (see my comments above).

  – the section about carbon cycling arises out of sudden.

- figures: some of the selected figures are not convincing. Why focus sometimes on *Phaeocystis* and diatoms, sometimes on *Phaeocystis*, diatoms and coccolithophores and sometimes on all PFTs?

    – Fig. 2 presents a rather artificial classification of the phytoplankton community. Why is the 25% used for *Phaeocystis* and coccolithophores but 75% for diatoms (Fig 2a)? Is "Mixed" (Fig. 2a) the same as "Others" (Fig. 2b-d)?
    – how does the annual or climatological "relative contribution of the five PFTs" looks like (and not the seasonal contribution as in Fig. 2b-c)? If such a figure were shown the statements in the paragraph l. 348–354 might be more comprehensible.
    – Fig. 4 - why is silicate not chosen as an important factor for diatoms? At least in the northern part of the SO (south of ∼40°S) diatoms are limited by silicate.

- the discussion and conclusion sections suffer from what I commented above. The authors must make clear what the paper is about in the first place. I am confident that also the discussion and conclusion section will then be easier to write.

**References**

Losa, S. N., Dutkiewicz, S., Losch, M., Oelker, J., Soppa, M. A., Trimborn, S., ... & Bracher, A. (2019). On modeling the Southern Ocean Phytoplankton Functional Types. Biogeosciences Discussions, 1-37.
Moisan, T. A., & Mitchell, B. G. (2018). Modeling Net Growth of *Phaeocystis antarctica* Based on Physiological and Optical Responses to Light and Temperature Co-limitation. Frontiers in Marine Science, 4, 437.

Nissen, C., Vogt, M., Münnich, M., Gruber, N., & Haumann, F. A. (2018). Factors controlling coccolithophore biogeography in the Southern Ocean. Biogeosciences, 15(22), 6997-7024.

---

## Referee Comment (RC2) · Anonymous Referee #2 · 2 Mar 2020

This is a very nicely written paper about a comprehensive and thorough model study that addresses the contribution of Phaeocystis to NPP and POC production in the Southern Ocean. The authors undertake a great effort to parameterize, test and constrain their model. Also, the discussion and conclusions take into account the uncertainties associated with this Phaeocystis, which I really appreciate. I enjoyed reading the paper and have just a few moderate comments and suggestions.

One of the outcomes of this study is that the relative and absolute importance of Phaeocystis for biogeochemical fluxes in this region is determined by its loss processes, in particular zooplankton grazing, and the so-called aggregation. Here, a few additional

sentences might help that discuss:

(a) The choice of food preferences and feeding parameterization of zooplankton. What I could find in preceding papers of the BEC model is that zooplankton is parameterized via fixed feeding preferences. However, other biogeochemical models have applied zooplankton grazing formulations that saturate with the total amount of food, or even employ a switching behaviour of zooplankton (see, e.g., Appendix A of the classic paper by Fasham et al., 1990, J. Mar. Res., 591-639). A few notes on that could complement the discussion; also, given that this process seems to be of importance, it might be helpful for the reader to have a brief explanation of the grazing formulation (and the preferences) in the methods description (so that the reader does not have to look up earlier papers).

(b) Aggregation: To my opinion, this term is somehow loosely defined in the present paper. Sometimes it is referred to as "mortality" (Table 1), sometimes as aggregation. Do phytoplankton become detritus after aggregation? But why? Theoretically, this process only describes that the cells or colonies collide and stick together - will they instantaneously stop being "green", i.e. cease photosynthesis and growth and become detritus? I assume that this is the case in the model, possibly with the argument that in this case they sink out of the euphotic quickly. However, given that in many cases aggregates ("marine snow") sink rather slowly, or not at all, this does not have to be the case. As for (a), given the large importance of this loss term for the simulated biogeo-chemistry, I would recommend some more in depth model description and discussion of this assumption,

Some few smaller comments:

Table 1 and line 175: The unit of quadratic mortality (aggregation) is given as 1/d. Shouldn't it be 1/((mmol N/m3)*d), given that it will be multiplied with the squared concentration?

Line 184: "we us monthly climatological fields for all tracers" - For all nutrients? Dissolved inorganic tracers? Please specify.

Lines 197-214, spinup procedure of the coupled model: here a simple diagram of the spinup procedure could help a lot! E.g. (if I understood correctly),

..30y physics.....10yBEC...10yBaseline..10ySensi

[Figure]

.............................................|5yAn|.....|5yAn|

Line 275: "phytoplankton biomass ... is the balance" - I suggest to rephrase this as "phytoplankton biomass ... is determined by the balance"

Line 320 and elsewhere: "In ROMS-BEC" - I assume what is referred to here is the baseline experiment? If so, I'd suggest to use "Baseline", to not confuse this simulation with the earlier non-Phaeocystis model and simulation.

Figure 4: The upper and lower panels would be easier to compare if in the lower panels the x- and y-axis were swapped (i.e., to have always temperature on the x-axis.

Figure 5: The caption could also note over what depth these terms were calculated.

Figure 6: If I add up the different contributions to POC formation in the right panel (60-90S) I end up with (6+17+4(bluearrow)+0.2+0.1+13+9=49.3% but the p-ratio is given as 45%. Does the blue arrow not contribute to the total flux? If so, then in the left panel the p-ratio should be 3+19+0.8+3+5+2=32.8% (and not 37%). Please clarify.

―――――――――――――――――――

---

## Author Comment (AC1) · 2 May 2020

**Answer to referee #1:**

We thank referee #1 for reviewing our manuscript. His/Her valuable comments and suggestions have significantly improved the quality of our manuscript.

Below, we include our detailed answers to all comments and questions.

**Answers to general comments (GC):**

**General Comment #1:**

*Nissen and Vogt present a model study on the relative importance of the colonial form of Phaeocystis for ecosystem processes and biogeochemical fluxes; they evaluate their results with observations from different data sources. A comparable study (Nissen et al 2018) had been performed with a focus on coccolithophores instead of Phaeocystis with similar analyses. In that respect this work is not overly innovative nor are original ideas presented. More critical is, however, that there is no thread in this manuscript; a clear goal is missing. A number of topics (e.g. phenology, competition, carbon and DMS-fluxes) are touched but not thoroughly permeated. It is unclear whether the authors would like to study the success of Phaeocystis compared to other phytoplankton functional groups or the importance of Phaeocystis for carbon export fluxes. Either way, no comprehensible motivation for either of these broad themes is provided. Some aspects of the methodology also need to be revised with consequences for the model analyses. Last but not least, recent work on this topic has been ignored. Overall this manuscript is premature and the authors must clarify their focus before publication. To sharpen the focus maybe it helps to look at the unpublished, recent modelling work on Southern Ocean Phaeocystis and PFTs (Losa et al. 2019) that has been put up for discussion in Biogeosciences Discussion.*

**Answer to GC1:**

We thank reviewer 1 for his/her constructive criticism on our work, regarding the focus, the motivation, the novelty, the methodology, and the presentation of our study. We address the concerns of the reviewer 1 with regard to these aspects in the revised manuscript through the following changes:

1) We have changed the title of the manuscript to "Factors controlling the competition between *Phaeocystis* and diatoms in the Southern Ocean and implications for carbon export fluxes" so that it better reflects the focus of the study, namely the links between the variability in phytoplankton community structure and downward carbon fluxes in the high-latitude Southern Ocean throughout the year.
2) We have entirely revised the introduction which now clarifies the focus and novelty of the study and includes additional recent literature.
3) We have restructured the result section and adjusted the relative weighting of the individual sections to have a more balanced representation of the different aspects of the study, especially regarding the drivers of the competition between *Phaeocystis* and diatoms and its biogeochemical implications.
4) Ultimately, within the discussion section of the revised version of the manuscript, we have adopted the same structure of subsections as in the result section, making it easier for the reader to follow. Furthermore, we have adjusted the lengths of the discussion of the individual aspects, in order to better represent the main focus of the study.

For the comment regarding the methodology (i.e., the temperature sensitivity of phytoplankton growth), we refer the reviewer to our detailed answer to SC4 and SC5 below.

In our study, we set out for a comprehensive assessment of the link between plankton biogeography and biogeochemical cycling in the Southern Ocean over the course of the year. Since we consider the comprehensiveness as a key strength and key aspect of novelty of the current paper as compared to previous work, the emphasis of our revision has been to (1) clarify the aims of the study in the revised version of the introduction, (2) highlight the current gaps in our understanding with regard to the

missing link between plankton biogeography and ecosystem function in terms of global biogeochemical cycling, and (3) improve upon the presentation of our study in the manuscript. Previous studies have often only presented snap shots of the factors controlling the relative importance of *Phaeocystis* and diatoms at high SO latitudes and its implications for downward carbon fluxes at a specific location and/or point in time (e.g. **Arrigo et al., 1998, Garcia et al., 2009, Wang et al., 2011**, but see the introduction of the manuscript for a comprehensive overview), meaning that the biogeochemical implications of the seasonally varying phytoplankton community remain under-explored, especially on larger spatial scales. We clarify these issues in the revised version of the manuscript, as detailed in the sections below.

In the following, we will address the individual concerns raised by the reviewer in more detail and summarize how we have addressed them in the revised version of the manuscript.

**Focus/Novelty/Motivation**

In this paper we set out to assess the link between the spatio-temporal variability in high-latitude Southern Ocean phytoplankton community structure and the variability in downward carbon fluxes. To that aim, we extended the work by **Nissen et al., 2018** to develop a model which would include all major biogeochemical actors of this region, a prerequisite to address this research question. Hence, with this tool, we were able to provide a first comprehensive assessment of the spatio-temporal variability of pathways leading to downward fluxes of carbon, which are inherently linked to the overlying phytoplankton community structure.

To clarify the focus of the study, we have changed the title of the manuscript to "Factors controlling the competition between *Phaeocystis* and diatoms in the Southern Ocean and implications for carbon export fluxes", so that it sets up the reader for the link between phytoplankton community structure and the implications for the carbon cycle.

Furthermore, we have substantially rewritten the introduction, to better highlight the focus, the novelty, and the motivation of our study. In this context, we apologize for the omission of certain recent papers in our initial submission. In response to the reviewer's comment, we have performed an additional extensive literature research and included the identified novel work in the revised version of our manuscript. We identified the following additional 7 papers that are of relevance for the current paper, and that were not included in the reference list of the initial submission:

**Papers describing the succession from *Phaeocystis* to diatoms throughout the season in the Ross Sea (Ryan-Keogh et al., 2017) and off the Western Antarctic Peninsula (Arrigo et al. 2017):**

Ryan-Keogh, T. J., DeLizo, L. M., Smith, W. O., Sedwick, P. N., McGillicuddy, D. J., Moore, C. M., & Bibby, T. S. (2017). Temporal progression of photosynthetic-strategy in phytoplankton in the Ross Sea, Antarctica. *Journal of Marine Systems*, *166*, 87–96. https://doi.org/10.1016/j.jmarsys.2016.08.014

Arrigo, K. R., van Dijken, G. L., Alderkamp, A., Erickson, Z. K., Lewis, K. M., Lowry, K. E., … van de Poll, W. (2017). Early Spring Phytoplankton Dynamics in the Western Antarctic Peninsula. *Journal of Geophysical Research: Oceans*, *122*(12), 9350–9369. https://doi.org/10.1002/2017JC013281

**Paper describing the impact of Fe concentrations on colony formation by *Phaeocystis Antarctica*:**

Bender, S. J., Moran, D. M., McIlvin, M. R., Zheng, H., McCrow, J. P., Badger, J., … Saito, M. A. (2018). Colony formation in Phaeocystis antarctica: connecting molecular mechanisms with iron biogeochemistry. *Biogeosciences*, *15*(16), 4923–4942. https://doi.org/10.5194/bg-15-4923-2018

**Papers on recent modeling of *Phaeocystis Antarctica*, focusing either on interactions between light and temperature on growth rates (Moisan & Mitchel, 2018) or functional type modeling in the Southern Ocean (Losa et al., 2019):**

Moisan, T. A., & Mitchell, B. G. (2018). Modeling Net Growth of Phaeocystis antarctica Based on Physiological and Optical Responses to Light and Temperature Co-limitation. *Frontiers in Marine Science*, *4*(February), 1–15. https://doi.org/10.3389/fmars.2017.00437

Losa, S. N., Dutkiewicz, S., Losch, M., Oelker, J., Soppa, M. A., Trimborn, S., Xi, H., and Bracher, A.: On modeling the Southern Ocean Phytoplankton Functional Types, Biogeosciences Discuss., https://doi.org/10.5194/bg-2019-289, 2019.

**Papers discussing the role of aggregates (especially those from *Phaeoycstis Antarcitca*) as a vector for carbon transfer to depth in the Southern Ocean (relative to that of e.g. fecal pellets):**

Asper, V. L., & Smith, W. O. (2019). Variations in the abundance and distribution of aggregates in the Ross Sea, Antarctica. *Elem Sci Anth*, *7*(1), 23. https://doi.org/10.1525/elementa.355

Ducklow, H. W., Wilson, S. E., Post, A. F., Stammerjohn, S. E., Erickson, M., Lee, S., … Yager, P. L. (2015). Particle flux on the continental shelf in the Amundsen Sea Polynya and Western Antarctic Peninsula. *Elementa: Science of the Anthropocene*, *3*, 000046. https://doi.org/10.12952/journal.elementa.000046

The analysis of this body of work reveals that these more recent findings are complementary to our results, and their inclusion into the introduction and discussion sections of our paper increases the quality of the discussion in the revised version of the manuscript.

In particular, we have included the references on the role of aggregates for POC export in the discussion section 4.2. of the revised manuscript (section on biogeochemical implications) and have added the study by **Losa et al. (2019)** in the discussion section 4.3 (Limitations & Caveats), discussing the complexity in marine ecosystem models:

[revised manuscript text omitted]

**Structure**

In response to the reviewer's comments, we have revised the results and the discussion section of the manuscript to make the order and relative weighting of individual sections clearer to the reader, and to better align the presentation of results with the core questions this paper aims to address.
In particular, we have merged the result sections 3.3 & 3.4 of the original version of the manuscript into a single section in the revised manuscript, which is entitled "Drivers of SO phytoplankton biogeography, phenology, and succession patterns". This section was shortened in the merging process, with the aim to make it more readable and to better balance the amount of text spent on the description of a) simulated patterns of biogeography, phenology, and succession, b) the drivers of the competition, and c) its biogeochemical implications. Please see our answer to SC8 for the new result section 3.3.

Furthermore, in order to make it easier for the reader to follow, we have adjusted the order of subsections within the discussion section to reflect their order in the result section, i.e., swapped discussion section 4.1 & 4.2 of the original manuscript. In the revised manuscript, the discussion of the

drivers of the competition of *Phaeocystis* and diatoms (section 4.1) is now followed by the discussion of its biogeochemical implications (section 4.2). In addition, in the latter, we have modified the paragraph on the implications of Southern Ocean *Phaeocystis* biogeography for DMS fluxes. In particular, we have shortened the paragraph on DMS from the method section 2.3.1 and moved it to section 4.2 in the revised manuscript, so that the manuscript is more clearly focused on carbon fluxes up until this point. Please see our answer to SC14 for the new paragraph on DMS.

**Answers to specific comments (SC):**

**SC1**: *title: the title only partly reflects the content of this study*

We thank the reviewer for this important comment, as it made us aware of imbalances in terms of content in the original version of the manuscript. As the analysis regarding the implications of the variability in phytoplankton community structure on high-latitude carbon fluxes is an important, novel aspect of the study, which has been highlighted even more in the revised version of the manuscript (see also answer to GC 1 above), the revised version of the manuscript will be entitled "Factors controlling the competition between *Phaeocystis* and diatoms in the Southern Ocean and implications for carbon export fluxes". Thereby, the content of the manuscript is better reflected by its title, helping the reader to follow.

**SC2**: *abstract and entire manuscript: it is unclear which research gap the authors want to fill. What is currently unclear - which open question in this research field are attempted to be answered with ROMS-BEC?*

In response to the reviewer's comments, we have substantially reworked the manuscript, in order to more clearly highlight the knowledge gap filled with this study. After having added *Phaeocystis* as a functional type to ROMS-BEC, we were able to provide a first comprehensive assessment of the spatio-temporal variability of pathways leading to downward fluxes of carbon at high Southern Ocean latitudes, which are inherently linked to the overlying phytoplankton community structure, especially the competition between *Phaeocystis* and diatoms. We kindly refer the reviewer to our response to GC1 above for more details.

**SC3**: *the manuscript should stand alone. Currently important parts of the model description are missing. The prognostic equation for Phaeocystis with all source and sink terms as well as all functional dependencies of rates to environmental drivers need to be provided.*

We fully agree with the reviewer on this point and apologize for not having included a full description of growth and loss terms for phytoplankton biomass in the original version of the manuscript. In the revised version, we have included a full description of the relevant model equations of BEC in Appendix B and added corresponding references to this section in the method section 2.1 and throughout the text:

**Appendix B: BEC equations: Phytoplankton growth & loss**

[revised manuscript text omitted]

Phytoplankton $P'^i$ [mmol C m$^{-3}$] aggregate at an aggregation rate $\gamma^i_a$ [mmol C m$^{-3}$ day$^{-1}$] which is computed with the quadratic mortality rate constants $\gamma^i_{a,0}$ ([m$^3$ (mmol C)$^{-1}$ d$^{-1}$], Table 1) and :

$$\gamma^i_a = \min(\gamma^i_{a,max} \cdot P'^i, \gamma^i_{a,0} \cdot P'^i \cdot P'^i) \qquad (B15)$$
$$\gamma^i_a = \max(\gamma^i_{a,min} \cdot P'^i, \gamma^i_a) \qquad (B16)$$

In ROMS-BEC, $\gamma^i_{a,min}$ is 0.01 day$^{-1}$ for small phytoplankton and coccolithophores and 0.02 day$^{-1}$ for *Phaeocystis* and diatoms, and with $\gamma^i_{a,max}$ being 0.9 day$^{-1}$ for *Phaeocystis*, diatoms, coccolithophores, and small phytoplankton. Note that phytoplankton immediately stop photosynthesizing upon aggregation and that aggregation losses do not occur for diazotrophs in ROMS-BEC.

**SC4**: *the newly introduced formulation of the temperature dependent growth for the PFT Phaeocystis is fundamentally different from the description of the PFTs of the original BEC model. The former is a "Gauss-like" temperature dependent growth function with a temperature optimum. Any deviation from the optimum is a limitation, varying between 0...1. In contrast, the Q10-approach with different Q10 values that is applied to the other PFTs denotes the "sensitivity" in the exponential growth towards temperature - in these cases the higher the temperature, the higher the growth. Even if a relatively high reference temperature of 30 degrees Celsius is given (which is likely not reached in the Southern Ocean), there is no such thing as an optimum in the Q10 approach. Thus, the "limitation" values used in the analyses cannot easily be compared.*
*[Generally the question arises whether the Q10 approach should be applied to PFTs at all. Introduced by Eppley it is valid and a good description for bulk phytoplankton but as soon as the bulk is divided into groups, "Gauss-like functions" with a clear optimum seem to be more adequate.]*

We thank the reviewer for raising this important point. First of all, we completely agree with the reviewer in that the two approaches ("optimum" vs "Q10") to model the temperature-limited growth rates of phytoplankton are fundamentally different. However, we think that a comparison of the temperature-limited growth rates of *Phaeocystis* ("optimum") to that of diatoms (Q10) is still valid in our model, for reasons outlined in the following.

In lab experiments, individual phytoplankton species typically show an optimum temperature for growth, above and below which its growth is slowed down (see Fig. 1 below). Yet, in models, the Q10-approach describes the temperature-limited growth as an exponential function without a temperature optimum (see black lines in Fig. 2 below or Fig. A1 in our manuscript). Since models typically represent the whole phytoplankton community by a set of plankton functional types (PFTs,

**Le Quéré et al., 2005**), thereby combining multiple species into a single PFT, this Q10-function can hence be interpreted as the overlap of numerous optimum curves of numerous individual species.

In the 5-PFT setup of ROMS-BEC presented here, the PFT "*Phaeocystis*" only represents the single species of *Phaeocystis* present in the SO, namely *Phaeocystis antarctica* (**Schoemann et al., 2005**). This species has been shown to stop growing above temperatures of ~8°C (**Buma et al., 1991**), thus an optimum curve applies. At the same time, within the model PFT "diatoms", we do not model a specific species of diatoms, but the whole diatom community (typical PFT approach; **Le Quéré et al., 2005**). This means that with increasing temperatures towards lower latitudes, diatom growth will be less and less temperature-limited (relative to the prescribed maximum growth rate at 30°C), as we assume that there is always a species that can cope with these higher temperatures (see also blue dots in Fig. 2 below). Yet, this is not the case for *Phaeocystis antarctica*, which is not observed northwards of approximately 60°S (**Schoemann et al., 2005**). At latitudes north of 60°S, other bloom-forming species of *Phaeocystis* are typically found (**Schoemann et al., 2005** and Fig. 3 below). While these are *not* included in our study, there is no reason not to include these other species in global models, thus suggesting that the applicability of a temperature optimum curve to describe the growth of *Phaeocystis* in global models may be limited (see also black line in the lower panel of Fig. 2 below). Yet, the literature review of available growth rates of all *Phaeocystis* species presented in **Schoemann et al. (2005)** is best fit by using a temperature optimum curve despite multiple species being included in the analysis (see Fig. 3 below; compare to the fit Fig. 2), suggesting that the Q10 approach may be unsuitable – at least for the bloom-forming species of this phytoplankton type.

[Figure]

Fig. 1: Growth rates as a function of temperature for example high-latitude SO species of diatoms and *Phaeocystis* (**Boyd 2019**).

[Figure]

Fig. 2: Global compilation of diatom (top) and *Phaeocystis* (bottom) growth rates as a function of temperature by **Le Quéré et al. (2016)**. Black lines are Q10-functions fit to the data with Q10=1.93 and Q10=1.66 for diatoms and *Phaeoycstis*, respectively., as used in the PlankTOM10 model.

[Figure]

Fig. 3: Global compilation of *Phaeocystis* growth rates as a function of temperature by **Schoemann et al. (2005)**. Triangles represent *Phaeocystis Antarctica*, filled triangles its colonial stage.

To account for the different formulations to describe the temperature-limited growth rates of *Phaeocystis* and diatoms in ROMS-BEC in our analysis of their competition over time (section 3.4 of the manuscript), we directly compare the temperature-limited growth rates (in d[-1]) rather than the growth limitation by temperature of these two phytoplankton types (see Eq. 2 of the manuscript).

**SC5**: *temperature-dependent growth functions of any organism group usually have a negatively skewed thermal reaction norm. This is also true for Phaeocystis antarctica. Since there already exists a mathematical description for the temperature-& light-dependent growth function of Phaeocystis antarctica (Moisan and Mitchell 2018), I wonder why the authors have not used it. In fact there are more recent observation-based publications on Phaeocystis antarctica that may be of interest for this study.*

We thank the reviewer for pointing us to the manuscript by **Moisan and Mitchell (2018)**, which we had not been aware of. In comparison to the formulation used in ROMS-BEC (**Geider et al., 1998**), the equations presented in **Moisan & Mitchell (2018)** include the possibility for photoinhibition at high light intensities (expressed by beta; **Platt et al. 1980**) and a temperature dependent initial slope of the photosynthesis-irradiance-curve (alpha), but do not explicitly account for all effects of photoacclimation in their equations that are included in ROMS-BEC (e.g., the local chlorophyll:carbon ratio of phytoplankton and the nutrient limitation of its growth, see Eq. 3a-3d in

**Moisan & Mitchell, 2018** and Eq. B9 of the revised manuscript for ROMS-BEC). As a result, the set of equations provided by **Moisan & Mitchell (2018)** and the ones currently used in ROMS-BEC predict different temperature-light-limited net growth rates of *Phaeocystis antarctica* for any given temperature and PAR level (see Fig. 4 below). Furthermore, the ratio of the growth rate predicted by ROMS-BEC and that obtained with **Moisan & Mitchell (2018)** varies substantially across temperatures and light levels (see Fig. 4d).

Overall, as a result of the differences between the formulation in **Moisan & Mitchell (2018)** and that in **Geider et al. (1998)**, the light limitation of growth by *Phaeocystis* is generally lower in ROMS-BEC than that predicted with the equations by **Moisan & Mitchell (2018)**, leading to substantially higher net growth rates in the current model than would be predicted if we were to apply the parameterization in **Moisan & Mitchell (2018)** to describe temperature and light-limited growth of *Phaeocystis* in ROMS-BEC (especially at low PAR levels, see Fig. 4d below). Due to the impact of nutrient limitation and chlorophyll:carbon ratios on the simulated net growth rates in ROMS-BEC, implementing the formulation by **Moisan & Mitchell 2018** would lead to substantially lower *Phaeocystis* biomass south of 60°S and would require a major retuning in the model to facilitate any substantial biomass accumulation of *Phaeocystis antarctica* colonies relative to diatoms in the high-latitudes, where these two phytoplankton types have been shown to locally and temporarily reach equally high biomass concentrations (**Vogt et al., 2012; Leblanc et al., 2012**).

A further issue with the parameterization that we encounter is its applicability within the temperature regime that constitutes the ecological niche of *Phaeocystis* in ROMS-BEC. We note that the parametrization by **Moisan & Mitchell (2018)**, being derived from laboratory experiments conducted at temperatures between -1.5-4°C, is currently only defined for temperatures below 6.8°C, above which the predicted growth rate becomes ecologically meaningless due to a negative alpha value (whereas this value should be >0, as it describes the sensitivity of photosynthetic rates of phytoplankton to increases of irradiance levels at low light). Altogether, given that the equations by **Moisan & Mitchel (2018)** do not account for all effects of photoacclimation which are accounted for in ROMS-BEC for all phytoplankton types and given that the alphaPI currently used in ROMS-BEC is backed up by the literature review in **Schoemann et al. (2005)**, we refrain from implementing the formulation by **Moisan & Mitchell (2018)** at this stage.

Nevertheless, taken together, this highlights the uncertainty still associated with model formulations describing the growth of phytoplankton functional types in general and *Phaeocystis* in particular. In response to the reviewer, we have modified section 4.3 (Limitations & Caveats) and added the following statement in the revised version of the manuscript:

"Furthermore, other functional relationships than those used in ROMS-BEC exist to describe the light and temperature dependent growth of *Phaeocystis* (e.g. Moisan and Mitchell, 2018). In comparison to the equations used in ROMS-BEC (see appendix B), the ones suggested by Moisan and Mitchell (2018; based on laboratory cultures of *Phaeocystis antarctica* grown under continuous blue light and at 4 different temperatures between -1.5°C and 4°C) lead to generally lower *Phaeocystis* growth rates, especially at PAR<50 W m$^{-2}$, suggesting that our biomass estimates at high latitudes and early/late in the season are associated with substantial uncertainty."

[Figure]

Fig. 4: a) Net growth rate of *Phaeocystis antarctica* as a function of temperature and light levels based on the equations in **Moisan & Mitchell (2018)**, assuming no photoinhibition (same as in ROMS-BEC), i.e., beta=0. b) & c) Same plot as a) obtained with the equations used in ROMS-BEC (see appendix B of revised manuscript and answer to SC3 above). Panel b) and c) show the resulting net growth rates for different nutrient conditions, with "severe nutrient limitation" in panel b) using g(N)=0.1 in Eq. B9 of the revised manuscript and "no nutrient limitation" in panel c) using g(N)=1. For both cases, we have here taken the surface annual average chlorophyll:carbon ratio of *Phaeocystis* in the *Baseline* simulation of the model (0.1434 mg chl / mmol C). Note that the formulation by **Moisan & Mitchell (2018)** does not account for the nutrient conditions or the chlorophyll:carbon ratio. Panel d) shows the ratio of panel a) and b), with the black contour denoting a 10-times higher growth rate in panel a) as compared to panel b).

**SC6**: *please specify which atmospheric forcing fields have been used.*
We refer the reviewer to L. 185/186 of the original version of the manuscript, where we state
"At the ocean surface, the model is forced with a 2003-normal year forcing for momentum, heat, and
freshwater fluxes (**Dee et al., 2011**)."

**SC7**: *model results: there is a mixture of model results, model evaluation, model comparison with
results from previous experiments which makes it difficult to read and to follow the arguments; the
entire results section needs to be revised.*

We thank the reviewer for this helpful comment, which made us reassess the chosen structure in the
result section of the original version of manuscript, leading to changes in the revised version as
outlined in the following. As we consider the addition of a new phytoplankton functional type a major
change in the complexity of ROMS-BEC, we have decided to first present a thorough model
evaluation of this new model setup by comparing to available observational data sets (sections 3.1 &
3.2). For the purpose of this study, a realistic representation of the high-latitude phytoplankton
community structure in both space and time is essential to address the competition of *Phaeocystis* and
diatoms throughout the year on the one hand and the implications for downward carbon fluxes on the
other. This part of the result section therefore also had the purpose to highlight model improvements
compared to the earlier version of the model without *Phaeocystis*, in order to stress why the 5-PFT
setup was essential for the questions at hand. Thereafter, we first present a detailed analysis on the
drivers of the competition between *Phaeocystis* and diatoms (sections 3.3 & 3.4 of the original
manuscript) and secondly on the implications for downward carbon fluxes (section 3.5 of the original
manuscript).

To increase the clarity of the result section and to overall better reflect the focus of the manuscript,
sections 3.3 & 3.4 of the original manuscript are merged into a single section called "Drivers of SO
phytoplankton biogeography, phenology, and succession patterns" in the revised version of the
manuscript. This new section 3.3 was shortened in the merging process (see also SC8 & SC10), in
order to better balance the two aspects of the study, namely the drivers of the competition between
*Phaeocystis* and diatoms and the implications for high-latitude carbon cycling. Furthermore, the title
of section 3.2 was changed in the revision process (now: "Patterns of phytoplankton phenology and
seasonal succession"), so that the reader is more clearly guided throughout the result section, starting
with a description of the simulated biogeography (section 3.1) and succession patterns (section 3.2)
and ending with a description of the drivers of these spatial and temporal patterns (section 3.3) and
their implications for carbon cycling (section 3.4). Please see also our answer to SC8-SC10 for more
details.

**SC8**: *the sections about the ecological niches, bottom-up and top-down effects are tedious to read and
questionable with respect to temperature (see my comments above).*

In the revised version of the manuscript, we tried to improve upon the readability of sections 3.3 and
3.4. In particular, we have moved the part on coccolithophores from section 3.3 of the original
manuscript to the supplement, in order to focus more clearly on the main topic of this study, namely
the competition between *Phaeocystis* and diatoms (see also our response to the reviewer's comment
SC10). Furthermore, we have merged the sections 3.3 & 3.4 of the original manuscript into a single
section in the revised version of the manuscript and revised its content in the process, in order to
improve the readability (see also SC7). The revised section 3.3 of the manuscript reads:

[revised manuscript text omitted]

Regarding the importance of temperature, the reviewer is kindly referred to our answer to SC4.

**SC9**: *the section about carbon cycling arises out of sudden.*

We thank the reviewer for this important comment. We fully agree with the reviewer in that the parts on the cycling of carbon were not motivated thoroughly enough in the original version of the manuscript. In response, we have added this aspect to the title of the revised manuscript, so that it now better reflects the content of the study (see also SC1). Further, we have substantially rewritten the introduction, so that it now better reflects and motivates the aspects covered in the result section and discussed thereafter, in particular the implications of variability in phytoplankton community structure for downward fluxes of carbon at high SO latitudes. The reviewer is referred to our answer to GC1 for more details.

**SC10**: *figures: some of the selected figures are not convincing. Why focus sometimes on Phaeocystis and diatoms, sometimes on Phaeocystis, diatoms and coccolithophores and sometimes on all PFTs?*

In general, we decided to show all PFTs in the model validation (Fig. 1 & 2). Furthermore, we chose to show the whole phytoplankton community whenever showing averages/integrals over 30-90°S (Fig. 6 & 7), where coccolithophores and small phytoplankton are non-negligible members of the community. In the manuscript, Fig. 3 & 5 directly concern the competition of diatoms and *Phaeocystis* at high latitudes. In these areas, these two phytoplankton types contribute >90% of the simulated NPP, which is why no other PFT is included in these figures (see Table 3 and Fig. 2 of the manuscript).

The only exception to the above reasoning in the original manuscript is Figure 4, where we had decided to show coccolithophores in addition to diatoms and *Phaeocystis*, but not the small phytoplankton PFT. The choice "pro coccolithophores" and "contra small phytoplankton" was motivated by the fact that coccolithophores do occupy a niche that is distinct from that of diatoms and *Phaeocystis*, whereas small phytoplankton do less so and are therefore not shown. Yet, we thank the reviewer for pointing out that this choice might be confusing for the reader. In order to make the focus of the paper clearer, we changed Fig. 4 so that the new version of this figure shows diatoms & *Phaeocystis only* in the revised version of the manuscript, thus moving the niche plots for coccolithophores to the supplement (new Fig. S8, see Figure below). This way, Fig. 3-5 of the revised manuscript include only diatoms and *Phaeocystis*. Together with the substantial revisions of result sections 3.3 & 3.4 of the original manuscript (see SC7 & SC8), the result section of the revised version of the manuscript is thereby now more clearly divided into a descriptive part of the simulated patterns in space and time (partly including coccolithophores and small phytoplankton, sections 3.1 & 3.2), a section describing the drivers of the competition of *Phaeocystis* and diatoms at high latitudes (section 3.3) and its implications for carbon cycling (section 3.4).

In the method section 2.3.1 of the revised manuscript, we have added the following statement: "In section 3.3 of this manuscript, only the results for *Phaeocystis* and diatoms will be shown, the corresponding figures for coccolithophores can be found in the supplementary material (Fig. S8 & S9)."

[Figure]

Fig. 5: Fig. S8 in the revised version of the manuscript

**SC11**: *Fig. 2 presents a rather artificial classification of the phytoplankton community. Why is the 25% used for Phaeocystis and coccolithophores but 75% for diatoms (Fig 2a)? Is "Mixed" (Fig. 2a) the same as "Others" (Fig. 2b-d)?*

Admittedly, the chosen thresholds are rather arbitrary and were chosen with the sole goal to indicate broad patterns of phytoplankton biogeography across the SO. The different thresholds for diatoms on the one hand and *Phaeocystis* and coccolithophores on the other hand were motivated by their different relative importance in their main region of occurrence. E.g., coccolithophores never dominate over diatoms, but still, we can define a clear SO coccolithophore biogeography – simply based on where they contribute most to NPP across the SO. If the 75% threshold was used for all PFTs, it would only be "diatoms" or "mixed". In this context, "mixed" denotes areas where diatoms do not contribute >75%, but neither coccolithophores nor *Phaeocystis* contribute >25%, e.g. if diatoms contribute 60% and coccolithophores and *Phaeocystis* 20%, respectively.

Consequently, "mixed" in panel a is not the same as "other" in panels b-d. As indicated in the method section 2.3.1 (L 224-226 of the original manuscript): "*The CHEMTAX analysis splits the*

*phytoplankton community into diatoms, nitrogen fixers (such as Trichodesmium), pico-phytoplankton (such as Synechococcus and Prochlorococcus), dinoflagellates, cryptophytes, chlorophytes (all three combined into the single group "Others" here), and haptophytes (such as coccolithophores and Phaeocystis)."*

In order to clarify this, areas, that were labeled "mixed" in the original version of Fig. 2a, are now labeled "co-existence" and we changed the figure caption accordingly. Furthermore, we added a statement in the figure caption in the revised version of the manuscript defining "others" in the panels including CHEMTAX information: "[…] "others" in the CHEMTAX fractions corresponds to dinoflagellates, cryptophytes, and chlorophytes […]"

**SC12**: *how does the annual or climatological "relative contribution of the five PFTs" looks like (and not the seasonal contribution as in Fig. 2b-c)? If such a figure were shown the statements in the paragraph l. 348–354 might be more comprehensible.*

We decided to only give the annual mean/integral numbers for NPP (see Table 3) and focus on the seasonal evolution for chlorophyll in Figure 2, which we can directly compare to HPLC-based estimates. The annual mean contribution to mixed layer chlorophyll levels of *Phaeocystis*/diatoms/coccolithophores amounts to 12.2/64.5/9.8 (30-90°S) and 31.1/54.8/2.4 (60-90°S) in our model, in rather close agreement with the estimates for NPP (15.3/53/14.6 between 30-90°S and 45.8/49.1/0.7 between 60-90°S, see Table 3).
Furthermore, we want to highlight the data scarcity in general and especially in all seasons besides summer in this context (see numbers printed below upper pie charts in Fig. 2), preventing a meaningful comparison of *annual mean* community structure in the model with the CHEMTAX data, which is why no annual mean figure is shown for the CHEMTAX data.

**SC13**: *Fig. 4 - why is silicate not chosen as an important factor for diatoms? At least in the northern part of the SO (south of ~40°S) diatoms are limited by silicate.*

As shown in Fig. S1, the reviewer is correct in pointing to a growth limitation of diatoms by silicic acid close to 40°S. Yet, as the focus of this paper is the competition between diatoms and *Phaeocystis*, which mainly takes place south of 60°S in ROMS-BEC, we chose not to show silicic acid as one of the environmental variables here, as the availability of silicic acid does not limit diatom growth in the focus area of this study. In fact, across Si levels, diatom biomass varies substantially south of 40°S (see Figure below), indicating that it is not a major control on diatom biomass levels in the area. For completeness, we add the figure below to the supplementary material (Fig. S8 in original manuscript, S9 in revised version) in the revised version of the manuscript.

[Figure]

Fig. 6: Same as the ecological niche plots in Fig. 4 of the manuscript, but showing phytoplankton biomass as a function of silicic acid concentrations [mmol m$^{-3}$]. This figure will be added as Fig. S9 to the revised manuscript.

**SC14**: *the discussion and conclusion sections suffer from what I commented above. The authors must make clear what the paper is about in the first place. I am confident that also the discussion and conclusion section will then be easier to write.*

We thank the reviewer for this comment, in direct response to which we have made several modifications to the manuscript. Besides small modifications to the text of all discussion sections and the conclusion section to improve upon the clarity of the text and to better reflect the focus of the study, we have changed the order of discussion sections 4.1 & 4.2 in the revised version of the manuscript, so that it reflects the order in which these aspects are described in the result sections (first drivers, then biogeochemical implications).

Furthermore, we have moved the part about DMS from the method section 2.3.1 in the original version of the manuscript to the new discussion section 4.2, to more clearly focus the method section on aspects regarding carbon cycling, which is the main focus of the paper.

The part about DMS was shortened in the process, and the new paragraph reads:

Besides its impact on the carbon cycle, *Phaeocystis* is the major contributor to the marine sulphur cycle in the SO through its production of DMSP (Keller et al., 1989; Liss et al., 1994; Stefels et al., 2007). Though not explicitly including the biogeochemical cycling of sulphur, we can nevertheless use model output from ROMS-BEC to obtain an estimate of DMS production by *Phaeocystis* through a simple back-of-the-envelope calculation. Integrating the modeled *Phaeocystis* biomass loss rates via zooplankton grazing and non-grazing mortality over the top 10 m, assuming a molar DMSP:C ratio for *Phaeocystis* of 0.011 (Stefels et al., 2007), and a DMSP-to-DMS conversion efficiency between 0.2-0.7 (the DMS yield depends on the local sulphur demand of bacteria, Stefels et al., 2007; Wang et al., 2015), our estimated annual DMS production by *Phaeocystis* in ROMS-BEC amounts to 3.3-11.5 Tg S and 1.8-6.4 Tg S south of 30° S and 60° S, respectively. Consequently, assuming that all of this DMS production quickly escapes to the atmosphere, our estimates correspond to 11.6-40.1% (30-90° S) and 6.5-22.7% (60-90° S) of the global flux of DMS to the atmosphere previously estimated by Lana et al. (2011, 28.1 Tg S yr$^{-1}$). Our estimate is an upper bound, however, as not all DMS produced in seawater is readily released to the atmosphere. In fact, a fraction is likely broken down by bacteria, by photolysis, or is mixed down in the water column (see e.g. Simó and Pedrós-Alló, 1999;

Stefels et al., 2007). Still, given that other phytoplankton types also produce DMS(P) (Keller et al., 1989; Stefels et al., 2007), the ROMS-BEC-based contribution of SO *Phaeocystis* alone (3.3-11.5 Tg S yr$^{-1}$) to the global flux of DMS to the atmosphere is in agreement with the flux suggested in Lana et al. (2011, 8.1 Tg S yr$^{-1}$ south of 30° S, i.e., 29% of their global estimate), and the substantial contribution of SO *Phaeocystis* underpins its major role for the global cycling of sulphur.

**Cited literature**

Arrigo, K. R., Schnell, A., & Lizotte, M. P. (1998). Primary production in Southern Ocean waters. *Journal of Geophysical Research*, *103*(C8), 15587–15600. https://doi.org/10.1029/98JC00930

Boyd, P. W. (2019). Physiology and iron modulate diverse responses of diatoms to a warming Southern Ocean (supplement). *Nature Climate Change*, *9*(2), 148–152. https://doi.org/10.1038/s41558-018-0389-1

Brun, P., Vogt, M., Payne, M. R., Gruber, N., O'Brien, C. J., Buitenhuis, E. T., … Luo, Y.-W. (2015). Ecological niches of open ocean phytoplankton taxa. *Limnology and Oceanography*, *60*(3), 1020–1038. https://doi.org/10.1002/lno.10074

Buma, A. G. J., Bano, N., Veldhuis, M. J. W., & Kraay, G. W. (1991). Comparison of the pigmentation of two strains of the prymnesiophyte Phaeocystis sp. *Netherlands Journal of Sea Research*, *27*(2), 173–182. https://doi.org/10.1016/0077-7579(91)90010-X

Dee, D. P., Uppala, S. M., Simmons, A. J., Berrisford, P., Poli, P., Kobayashi, S., … Vitart, F. (2011). The ERA-Interim reanalysis: configuration and performance of the data assimilation system.

*Quarterly Journal of the Royal Meteorological Society*, *137*(656), 553–597. https://doi.org/10.1002/qj.828

Garcia, N., Sedwick, P., & DiTullio, G. (2009). Influence of irradiance and iron on the growth of colonial Phaeocystis antarctica: implications for seasonal bloom dynamics in the Ross Sea, Antarctica. *Aquatic Microbial Ecology*, *57*(2), 203–220. https://doi.org/10.3354/ame01334

Geider, R. J., MacIntyre, H. L., & Kana, T. M. (1998). A dynamic regulatory model of phytoplanktonic acclimation to light, nutrients, and temperature. *Limnology and Oceanography*, *43*(4), 679–694. https://doi.org/10.4319/lo.1998.43.4.0679

Leblanc, K., Arístegui, J., Armand, L., Assmy, P., Beker, B., Bode, A., … Yallop, M. (2012). A global diatom database – abundance, biovolume and biomass in the world ocean. *Earth System Science Data*, *4*(1), 149–165. https://doi.org/10.5194/essd-4-149-2012

Le Quéré, C., Harrison, S. P., Colin Prentice, I., Buitenhuis, E. T., Aumont, O., Bopp, L., … Wolf-Gladrow, D. (2005). Ecosystem dynamics based on plankton functional types for global ocean biogeochemistry models. *Global Change Biology*, *11*, 2016–2040. https://doi.org/10.1111/j.1365-2486.2005.1004.x

Le Quéré, C., Buitenhuis, E. T., Moriarty, R., Alvain, S., Aumont, O., Bopp, L., … Vallina, S. M. (2016). Role of zooplankton dynamics for Southern Ocean phytoplankton biomass and global biogeochemical cycles. *Biogeosciences*, *13*(14), 4111–4133. https://doi.org/10.5194/bg-13-4111-2016

Nissen, C., Vogt, M., Münnich, M., Gruber, N., & Haumann, F. A. (2018). Factors controlling coccolithophore biogeography in the Southern Ocean. *Biogeosciences*, *15*(22), 6997–7024. https://doi.org/10.5194/bg-15-6997-2018

Platt, T., Gallegos, C. L., & Harrison, W. G. (1980). Photoinhibition of Photosynthesis in Natural Assemblages of Marine Phytoplankton. *Journal of Marine Research*.

Schoemann, V., Becquevort, S., Stefels, J., Rousseau, V., & Lancelot, C. (2005). Phaeocystis blooms in the global ocean and their controlling mechanisms: a review. *Journal of Sea Research*, *53*(1–2), 43–66. https://doi.org/10.1016/j.seares.2004.01.008

Vogt, M., O'Brien, C., Peloquin, J., Schoemann, V., Breton, E., Estrada, M., … Peperzak, L. (2012). Global marine plankton functional type biomass distributions: *Phaeocystis* spp. *Earth System Science Data*, *4*(1), 107–120. https://doi.org/10.5194/essd-4-107-2012

Wang, S., & Moore, J. K. (2011). Incorporating Phaeocystis into a Southern Ocean ecosystem model. *Journal of Geophysical Research*, *116*(C1), C01019. https://doi.org/10.1029/2009JC005817

---

## Author Comment (AC2) · 2 May 2020

**Answer to referee #2:**

We thank referee #2 for taking the time to provide valuable comments and suggestions that have helped to improve our manuscript.

Below, we include our detailed answers to all comments and questions.

**Answers to general comments (GC):**

" […] a few additional sentences might help that discuss

*GC1: The choice of food preferences and feeding parameterization of zooplankton. What I could find in preceding papers of the BEC model is that zooplankton is parameterized via fixed feeding preferences. However, other biogeochemical models have applied zooplankton grazing formulations that saturate with the total amount of food, or even employ a switching behaviour of zooplankton (see, e.g., Appendix A of the classic paper by Fasham et al., 1990, J. Mar. Res., 591-639). A few notes on that could complement the discussion; also, given that this process seems to be of importance, it might be helpful for the reader to have a brief explanation of the grazing formulation (and the preferences) in the methods description (so that the reader does not have to look up earlier papers).*

We thank the reviewer for raising this point. The reviewer is correct in that BEC currently assumes fixed feeding preferences, which are set by differences in the maximum grazing rate $\gamma_{g,max}$ across the PFTs. Here, based on size assumptions, we assume a preferential feeding of the single zooplankton grazer in ROMS-BEC on smaller phytoplankton (higher $\gamma_{g,max}$ for small phytoplankton and coccolithophores than larger ones like diatoms and *Phaeocystis*, see table 1 in manuscript). Similarly, we assume preferential feeding on diatoms relative to *Phaeocystis* colonies (see section 2.1 of the original version of the manuscript).

Admittedly, by only including a single grazer that includes characteristics of both micro- and macrozooplankton (see **Moore et al., 2002**, but especially **Sailley et al., 2013**), the grazing formulation in ROMS-BEC is likely overly simplistic (e.g. **Le Quéré et al. 2016**). Furthermore, not accounting for adaptive feeding preferences or for total biomass to saturate zooplankton feeding at high total biomass levels are major shortcomings of the current parametrization (**Vallina et al. 2014**; **Vallina and Le Quéré, 2011**). These can be expected to significantly alter the interactions of the zooplankton with each PFT over the course of the growing season by e.g. temporarily alleviating the grazing pressure on all or single phytoplankton PFTs. The inclusion of multiple zooplankton functional types in ROMS-BEC is planned in current and ongoing work in our lab, but goes beyond the scope of this paper. Rather, the action of the zooplankton FT upon its prey should be viewed as a closure term, with phyto- and zooplankton biomass tightly coupled in space and time.

To clarify for the reader what parametrizations are currently used in ROMS-BEC, we have included a full description of the model equations describing growth and loss rates of phytoplankton biomass, including the equation for grazing, in the appendix of the revised version of the manuscript (see also our answer to reviewer #1):

The single zooplankton grazer $Z$ [mmol C m$^{-3}$] feeds on the respective phytoplankton $P'^i$ [mmol C m$^{-3}$] at a grazing rate $\gamma_g^i$ [mmol C m$^{-3}$ day$^{-1}$] that is given by:

$$\gamma_g^i = \gamma_{max}^i \cdot f^Z(T) \cdot Z \cdot \frac{P'^i}{z_{grz}^i + P'^i} \tag{B12}$$

with

$$f^Z(T) = 1.5 \cdot \exp(\frac{T - T_{ref}}{10°C}) \tag{B13}$$

In the discussion section 4.1, we have modified the text to mention the shortcomings of the grazing parametrization in ROMS-BEC more explicitly:

"Additionally, as discussed in Nissen et al. (2018), the lack of multiple zooplankton groups in the SO model (Le Quéré et al. 2016), and the parametrization of the single zooplankton grazer using fixed prey preferences and separate grazing on each prey using a Holling Type II function (Holling et al., 1959), which thus precluding a saturation of feeding at high total phytoplankton biomass, are major limitations of ROMS-BEC."

*GC2: Aggregation: To my opinion, this term is somehow loosely defined in the present paper. Sometimes it is referred to as "mortality" (Table 1), sometimes as aggregation. Do phytoplankton become detritus after aggregation? But why? Theoretically, this process only describes that the cells or colonies collide and stick together - will they instantaneously stop being "green", i.e. cease photosynthesis and growth and become detritus? I assume that this is the case in the model, possibly with the argument that in this case they sink out of the euphotic quickly. However, given that in many cases aggregates ("marine snow") sink rather slowly, or not at all, this does not have to be the case. As for (a), given the large importance of this loss term for the simulated biogeochemistry, I would recommend some more in depth model description and discussion of this assumption"*

We thank the reviewer for this point and apologize for any confusion. Yes, phytoplankton biomass in ROMS-BEC immediately becomes detritus after aggregation, thus immediately stops being "green". We agree with the reviewer in that this is likely not what happens for small aggregates in the real ocean, which do not sink out of the euphotic zone rapidly, suggesting that current model formulations in ROMS-BEC and other models are overly simplistic (see e.g. **Laufkötter et al., 2016**). Assuming that aggregation is less effective in quickly removing the smaller phytoplankton cells from the upper ocean, aggregation is formulated to be more effective for larger phytoplankton in ROMS-BEC (in our case diatoms and *Phaeocystis* colonies). Still, once formed, no differentiation is made in the model in how quickly the particles are transferred to depth between POC originating from aggregated small phytoplankton cells and those from larger phytoplankton types. We note, however, that this differentiation is prevented by the currently used single POC class in the model (see also section 4.3 in the originally submitted manuscript, L. 657ff). Furthermore, ideally, aggregation losses of each PFT should be calculated based on total biomass rather than based on the biomass of each PFT separately and should additionally consider larger detritus particles (POC) of different size classes. Since the ROMS-BEC set-up we use currently uses an implicit sinking formulation in which POC is directly redistributed and remineralized across the water column upon its formation, this precludes a tracking of aggregates and their fate in space and time (**Lima et al., 2014**).

Overall, we fully agree with the reviewer that our model (and other models, see discussion in **Laufkötter et al., 2016**) would benefit from an increased complexity regarding the fate of biomass losses and the resulting particles, and quantitative relationships should be established as more observations become available to guide model parametrizations (see e.g. **Guidi et al., 2015**).

In direct response to the reviewer's comment, we have revised the text in the manuscript to make a clearer distinction between non-grazing mortality and aggregation. In particular, we have revised the respective part of method section 2.1, which now reads:

"Furthermore, based on the assumption that for a given biomass concentration, larger cells are more likely than smaller cells to form aggregates and to subsequently stop photosynthesizing and sink as POC, we use a higher quadratic loss rate for *Phaeocystis* (0.005 $d^{-1}$) than for diatoms (0.001 $d^{-1}$) in the model (see $\gamma_{a,0}$ in Table 1)."

In Table 1 of the revised manuscript, we refer to the constant $\gamma_{a,0}$ as "quadratic loss rate in aggregation" in the revised manuscript:

**Table 1.** BEC parameters controlling phytoplankton growth and loss for the five phytoplankton PFTs diatoms (D), *Phaeocystis* (PA), coccolithophores (C), small phytoplankton (SP), and diazotrophs (N). Z=zooplankton, P=phytoplankton, PI=photosynthesis-irradiance. If not given in section 2.1, the model equations describing phytoplankton growth and loss rates are given in Nissen et al. (2018).

| Parameter | Unit | Description | D | PA | C | SP | N[†] |
|---|---|---|---|---|---|---|---|
| $\mu_{max}$ | $d^{-1}$ | max. growth rate at 30° C | 4.6 | ‡ | 3.8 | 3.6 | 0.9 |
| $Q_{10}$ | | temperature sensitivity | 1.55 | ‡ | 1.45 | 1.5 | 1.5 |
| $k_{NO3}$ | $mmol\ m^{-3}$ | half-saturation constant for $NO_3$ | 0.5 | 0.5 | 0.3 | 0.1 | 1.0 |
| $k_{NH4}$ | $mmol\ m^{-3}$ | half-saturation constant for $NH_4$ | 0.05 | 0.05 | 0.03 | 0.01 | 0.15 |
| $k_{PO4}$ | $mmol\ m^{-3}$ | half-saturation constant for $PO_4$ | 0.05 | 0.05 | 0.03 | 0.01 | 0.02 |
| $k_{DOP}$ | $mmol\ m^{-3}$ | half-saturation constant for DOP | 0.9 | 0.9 | 0.3 | 0.26 | 0.09 |
| $k_{Fe}$ | $\mu mol\ m^{-3}$ | half-saturation constant for Fe | 0.15 | 0.2 | 0.125 | 0.1 | 0.5 |
| $k_{SiO3}$ | $mmol\ m^{-3}$ | half-saturation constant for $SiO_3$ | 1.0 | - | - | - | - |
| $\alpha_{PI}$ | $\frac{mmol\ C\ m^2}{mg\ Chl\ W\ s}$ | initial slope of PI-curve | 0.44 | 0.63 | 0.4 | 0.44 | 0.38 |
| $\gamma_{g,max}$ | $d^{-1}$ | max. growth rate of Z grazing on P | 3.8 | 3.6 | 4.4 | 4.4 | 3.0 |
| $z_{grz}$ | $mmol\ m^{-3}$ | half-saturation constant for ingestion | 1.0 | 1.0 | 1.05 | 1.05 | 1.2 |
| $\gamma_{m,0}$ | $d^{-1}$ | linear non-grazing mortality | 0.12 | 0.18 | 0.12 | 0.12 | 0.15 |
| $\gamma_{a,0}$ | $\frac{m^3}{mmol\ C\ d}$ | quadratic loss rate in aggregation | 0.001 | 0.005 | 0.001 | 0.001 | - |
| $r_g$ | - | fraction of grazing routed to POC | 0.3 | 0.42 | 0.2 | 0.05 | 0.05 |

[†] Compared to Nissen et al. (2018), the $k_{Fe}$ of diazotrophs in ROMS-BEC is higher than for all other PFTs, consistent with literature reporting high Fe requirements of *Trichodesmium* (Berman-Frank et al., 2001). Furthermore, the maximum grazing rate on diazotrophs is lowest in the model (Capone, 1997). Still, diazotrophs continue to be a minor player in the SO phytoplankton community, contributing <1% to domain-integrated NPP in ROMS-BEC.

‡ The temperature-limited growth rate of *Phaeocystis* is calculated based on an optimum function according to Eq. 1 (see also Fig. A1a).

Furthermore, in order to make the differences between all the loss terms in the model more apparent, we have added a full description of the model equations as an appendix in the revised version of the manuscript (see also our response to reviewer #1). There, we have also included a sentence stating that phytoplankton in the model stop photosynthesizing upon aggregation:

Phytoplankton $P'^i$ [mmol C $m^{-3}$] aggregate at an aggregation rate $\gamma_a^i$ [mmol C $m^{-3}$ $day^{-1}$] which is computed with the quadratic mortality rate constants $\gamma_{a,0}^i$ ([$m^3$ (mmol C)$^{-1}$ $d^{-1}$], Table 1) and :

$$\gamma_a^i = \min(\gamma_{a,max}^i \cdot P'^i, \gamma_{a,0}^i \cdot P'^i \cdot P'^i) \tag{B15}$$

$$\gamma_a^i = \max(\gamma_{a,min}^i \cdot P'^i, \gamma_a^i) \tag{B16}$$

In ROMS-BEC, $\gamma_{a,min}^i$ is 0.01 $day^{-1}$ for small phytoplankton and coccolithophores and 0.02 $day^{-1}$ for *Phaeocystis* and diatoms, and with $\gamma_{a,max}^i$ being 0.9 $day^{-1}$ for *Phaeocystis*, diatoms, coccolithophores, and small phytoplankton. Note that phytoplankton immediately stop photosynthesizing upon aggregation and that aggregation losses do not occur for diazotrophs in ROMS-BEC.

As an important caveat of this study, we have added the following sentences regarding the current formulation of aggregation in ROMS-BEC in section 4.1 of the revised manuscript:

«Here, our findings suggest an important role for biomass loss processes in controlling the relative importance of *Phaeocystis* and diatoms in ROMS-BEC, but very little quantitative information exists to constrain model parameters (see section 2.1) or to validate the simulated non-grazing mortality, grazing, or aggregation loss rates of *Phaeocystis* and diatoms over time. **Certainly, the simulated aggregation rates in the model and their impact on spatio-temporal distributions of PFT biomass concentrations and rates of NPP are associated with substantial uncertainty due to the immediate conversion of biomass to sinking detritus in the model, the equal treatment of POC originating from all PFTs, the neglect of disaggregation, and due to the calculation of aggregation rates based on the biomass concentrations of individual PFTs rather than all PFTs or even particles combined (see e.g. Turner, 2015).**»

**Answers to specific comments (SC):**

**SC1**: *Table 1 and line 175: The unit of quadratic mortality (aggregation) is given as 1/d. Shouldn't it be 1/((mmol N/m3)*d), given that it will be multiplied with the squared concentration?*
We thank the reviewer for this comment. The unit of the constant $\gamma_{a,0}$ given in Table 1 should indeed be 1/(mmol C m-3 d-1) and we have corrected this in the revised version of the manuscript (see also the revised Table 1 on the previous page). Furthermore, in response to a comment by reviewer #1, we have provided a full description of the model equations describing phytoplankton growth and loss in the appendix of the revised version of the manuscript.

**SC2**: *Line 184: "we use monthly climatological fields for all tracers" - For all nutrients? Dissolved inorganic tracers? Please specify.*
Yes, we use monthly climatological fields for all nutrient tracers. We used climatological data from World Ocean Atlas 2013 for all macronutrients (**Garcia et al., 2013**), data from GLODAP for DIC and alkalinity (**Lauveset et al., 2016**), and climatological output fields from a global simulation with CESM-BEC for ammonium, dissolved inorganic Fe, and all dissolved organic phases of the nutrients (DOC, DOP, DOPr, DON, DONr, DOFe, **Yang et al., 2017**).

**SC3**: *Lines 197-214, spin up procedure of the coupled model: here a simple diagram of the spinup procedure could help a lot! E.g. (if I understood correctly), ...30y physics.....10yBEC...10yBaseline (5 yr analysis)..10ySensitivity (5 yr analysis)*
Indeed, the reviewer has understood our procedure of the model simulations correctly. Given that the results presented in this study are not qualitatively dependent on the exact years analyzed (due to the climatological forcing applied in the simulations) and in light of the length of the manuscript, we refrain from adding another figure after careful consideration of the issue. However, we have slightly modified the description of the setup of the sensitivity experiments to make things even clearer:

"All sensitivity experiments use the same physical and biogeochemical spin-up as the *Baseline* simulation and start from the end of year 10 of the coupled ROMS-BEC spin-up."

**SC4**: *Line 275: "phytoplankton biomass ... is the balance" - I suggest to rephrase this as "phytoplankton biomass ... is determined by the balance"*
We have rephrased as suggested.

**SC5**: *Line 320 and elsewhere: "In ROMS-BEC" - I assume what is referred to here is the baseline experiment? If so, I'd suggest to use "Baseline", to not confuse this simulation with the earlier non-Phaeocystis model and simulation.*

We have modified the indicated sentence to start by "In the 5-PFT *Baseline* simulation of ROMS-BEC, […]". Furthermore, for the revised version of the manuscript, we have double-checked the whole text and clarified wherever we thought confusion was possible.

**SC6**: *Figure 4: The upper and lower panels would be easier to compare if in the lower panels the x-and y-axis were swapped (i.e., to have always temperature on the x-axis.*

We thank the reviewer for this excellent suggestion regarding Fig. 4. We have adopted this in the revised version of the manuscript (see Figure below). Furthermore, in response to a comment by reviewer #1, we have additionally moved the panels showing the ecological niches of coccolithophores to the supplementary material, in order to focus the manuscript earlier on the competition between *Phaeocystis* and diatoms.

[Figure]

Fig. 1: Revised version of Fig. 4 in the manuscript.

**SC7**: *Figure 5: The caption could also note over what depth these terms were calculated.*

We have modified the figure caption to state that Fig. 5 only shows the quantities at the surface:

"For all metrics, the left panels are surface averages over 60-90° S and those on the right for the Ross Sea."

We note that this choice is mainly motivated by the higher available temporal frequency in the necessary output variables. Overall, the dynamics of the seasonal competition between diatoms and *Phaeocystis* also broadly hold (at least qualitatively) for averages over the mixed layer over the growing season (not shown).

**SC8**: *Figure 6: If I add up the different contributions to POC formation in the right panel (60- 90S) I end up with (6+17+4(bluearrow)+0.2+0.1+13+9=49.3% but the p-ratio is given as 45%. Does the blue arrow not contribute to the total flux? If so, then in the left panel the p-ratio should be 3+19+0.8+3+5+2=32.8% (and not 37%). Please clarify.*

We thank the reviewer for spotting this inconsistency of the numbers, as there was indeed a mistake in the figure in the submitted manuscript regarding the individual pathways leading to POC production (i.e., the indicated p ratio was correct). As a result of correcting the respective factor applied in the post-processing of the model output, the fraction of grazing on *Phaeocystis* leading to POC production

are now corrected down to 3.4% (5% before) and 9.2% (13% before) for 30-90°S, respectively (see corrected Fig. 6 below).

[Figure]

Fig. 2: revised Fig. 6 of the manuscript.

While this does not affect the general conclusion from this analysis, we note that this affects the discussion in the text (see below). While grazing remains the main POC production pathway for *Phaeocystis*, the difference to aggregation is now minor at high latitudes (9.2% for grazing, 8.9% for aggregation).

Accordingly, we reformulate the corresponding part of the manuscript, which now reads:

"For both diatoms and *Phaeocystis*, grazing by zooplankton (i.e., the formation of fecal pellets) is the most important pathway of POC production in ROMS-BEC (black arrows in Fig. 6, 9%/52% and 20%/37% of total POC production for *Phaeocystis*/diatoms between 30-90° S and 60-90° S, respectively). Yet, at high latitudes (60-90° S), aggregation of *Phaeocystis* biomass contributes *equally* to POC formation."

Furthermore, we corrected a minor mistake in the caption of Fig. 6, where we falsely stated that the numbers describing the importance of the respective POC production pathway relative to total NPP were rounded to the nearest integer if they were >0.5%. Instead, this is only the case if the contribution of a respective pathway is >1%.

**Cited literature**

Garcia, H. E., Locarnini, R. A., Boyer, T. P., Antonov, J. I., Baranova, O. K., Zweng, M. M., … Johnson, D. R. (2013). *World Ocean Atlas 2013, Volume 4 : Dissolved Inorganic Nutrients (phosphate, nitrate, silicate)*. (S. (Ed. . Levitus & A. (Technical E. . Mishonov, Eds.) (Vol. 4). NOAA Atlas NESDIS 76.

Guidi, L., Legendre, L., Reygondeau, G., Uitz, J., Stemmann, L., & Henson, S. A. (2015). A new look at ocean carbon remineralization for estimating deepwater sequestration. *Global Biogeochemical Cycles*, *29*(7), 1044–1059. https://doi.org/10.1002/2014GB005063

Laufkötter, C., Vogt, M., Gruber, N., Aumont, O., Bopp, L., Doney, S. C., … Völker, C. (2016). Projected decreases in future marine export production: the role of the carbon flux through the upper ocean ecosystem. *Biogeosciences*, *13*(13), 4023–4047. https://doi.org/10.5194/bg-13-4023-2016

Lauvset, S. K., Key, R. M., Olsen, A., Van Heuven, S., Velo, A., Lin, X., … Watelet, S. (2016). A new global interior ocean mapped climatology: The 1° × 1° GLODAP version 2. *Earth System Science Data*, *8*(2), 325–340. https://doi.org/10.5194/essd-8-325-2016

Le Quéré, C., Buitenhuis, E. T., Moriarty, R., Alvain, S., Aumont, O., Bopp, L., … Vallina, S. M. (2016). Role of zooplankton dynamics for Southern Ocean phytoplankton biomass and global biogeochemical cycles. *Biogeosciences*, *13*(14), 4111–4133. https://doi.org/10.5194/bg-13-4111-2016

Lima, I. D., Lam, P. J., & Doney, S. C. (2014). Dynamics of particulate organic carbon flux in a global ocean model. *Biogeosciences*, *11*(4), 1177–1198. https://doi.org/10.5194/bg-11-1177-2014

Moore, J. K., Doney, S. C., Kleypas, J. A., Glover, D. M., & Fung, I. Y. (2002). An intermediate complexity marine ecosystem model for the global domain. *Deep Sea Research Part II: Topical Studies in Oceanography*, *49*(1–3), 403–462. https://doi.org/10.1016/S0967-0645(01)00108-4

Sailley, S. F., Vogt, M., Doney, S. C., Aita, M. N., Bopp, L., Buitenhuis, E. T., … Yamanaka, Y. (2013). Comparing food web structures and dynamics across a suite of global marine ecosystem models. *Ecological Modelling*, *261–262*, 43–57. https://doi.org/10.1016/j.ecolmodel.2013.04.006

Vallina, S. M., Ward, B. A., Dutkiewicz, S., & Follows, M. J. (2014). Maximal feeding with active prey-switching: A kill-the-winner functional response and its effect on global diversity and biogeography. *Progress in Oceanography*, *120*, 93–109. https://doi.org/10.1016/j.pocean.2013.08.001

Vallina, S. M., & Le Quéré, C. (2011). Stability of complex food webs: Resilience, resistance and the average interaction strength. *Journal of Theoretical Biology*, *272*(1), 160–173. https://doi.org/10.1016/j.jtbi.2010.11.043

Yang, S., Gruber, N., Long, M. C., & Vogt, M. (2017). ENSO-Driven Variability of Denitrification and Suboxia in the Eastern Tropical Pacific Ocean. *Global Biogeochemical Cycles*, *31*(10), 1470–1487. https://doi.org/10.1002/2016GB005596

---

## Referee Report (RR1)

Review of "Factors controlling the competition between Phaeocystis and diatoms in the Southern Ocean and implications for carbon export fluxes" (bg-2019-488) by C. Nissen and M. Vogt.

**Introduction**

In this manuscript, the authors extend previous work reported by Nissen et al. (2018) for the simulation of phytoplankton in the Southern Ocean. Its primary novelty is the separation from other modeled phytoplankton functional types of *Phaeocystis* colonies into a new model component.

In brief, this work involved the use of an existing model, creating a new phytoplankton group, and running the model for the Southern Ocean over 10 years. Analyses with the simulated 5-year daily climatology consist of scrutinizing the relationships between variables in the simulation and comparing the model output to data. Conclusions are drawn about mechanisms driving spatial and temporal patterns and carbon export. Tangentially, seven sensitivity experiments are run to highlight which aspects of the new phytoplankton group have had the greatest affect in distinguishing it from the original no-*Phaeocystis* model.

This work does not include much experimentation with the model. Instead, the simulation outputs of a *baseline* run are most thoroughly examined, starting with general biogeographical patterns. However, the attempt to include or address so many questions in this manuscript (how important is *Phaeocystis* to carbon export; what are the spatial and temporal patterns of *Phaeocystis* and diatom biomass; what are the drivers of *Phaeocystis* and diatoms' spatio-temporal patterns) makes it feel unfocused. The structure of the paper and section headings only partly help. I am left wondering, which comparisons do the authors feel are most important, most revealing, or most surprising? Although main conclusions are stated in the conclusion section and abstract, the attention paid to each analysis step and their findings in the body does not seem to match these main points.

This work provides a step towards more thorough and comprehensive modeling of Southern Ocean phytoplankton. Therefore, I do think this should be published following textual revisions by which the aims and scope of the research are more clearly represented.

**General comments**

The section headings do not seem consistent with the scope of what is being assessed, particularly which PFTs are addressed. Section 3.1 includes "phytoplankton" in the heading and assessed all PFTs. In contrast, sections 3.2 and 3.3 include "phytoplankton" in the heading but only addresses *Phaeocystis* and diatoms. Likewise, section 4.2 includes "phytoplankton" in the heading but seems to only discuss

*Phaeocystis.* I recommend the authors revise the headings or make the content more consistent with the headings.

As I read, I wonder: why is the Ross Sea singled out for evaluation, aside from other coastal areas? Also, in some manuscript sections the Ross Sea is included in the comparisons (e.g. Figure 2, section 3.3 about drivers) and other sections do not include it (e.g. Table 3, section 3.4 about carbon cycle). Why is it only considered for some of the analyses? Without an explanation, these choices make the analysis seem arbitrary. The authors should explain why the Ross Sea is being used as a special study area and why/when it is or is not being included in analyses.

The differences in carbon to chlorophyll ratio may have a substantial impact on some of the conclusions, and yet it seems to have been given little consideration. I refer the authors to several additional papers discussing C:Chl ratios for *Phaeocystis* and diatoms in the Ross Sea: DiTullio and Smith (1996), Smith et al. (1998), Mathot et al. (2000), Kaufman et al. (2018).

I appreciate that calibration is difficult with such a large model, however, this seems to be an important limitation not discussed. I suggest the authors consider addressing it. Moreover, If the authors did train some of the model parameters before picking the 'best' values for their *baseline* run, it should be made clear whether or not model evaluation was done using the same or different data than was used for parameter training/tuning/calibration.

**Specific comments**

The last paragraph of the introduction does not accurately reflect the organization of the paper. This seems like a great place for the authors to more coherently state the purpose of the analyses.

Line 95: Perhaps it is just me, but I am confused by this sentence. Also, the implication that the model provides "a **correct** representation of SO phytoplankton biogeography" (emphasis added) seems very presumptuous.

In section 2.1, the authors refer to a "*baseline*" simulation before it is described. It would be helpful for the authors to refrain from referencing *baseline* before it is defined.

Sect. 2.3: I wonder what the authors mean by "analysis framework" in the section title? To me, growth rate ratios are not an analysis framework, but rather simply a diagnostic variable.

Sect. 2.3.2: The authors should define Betas in the text.

Line 279, and elsewhere: I think "N" is being used to represent both diazotrophs and nutrients. The authors should restrict its meaning to only one or the other.

Lines 318-323: I think the bias could also be due to poor calibration, especially of the newly introduced *Phaeocystis* group.

Line 331: "compared to the 4-PFT"

Line 361-362: "Our model suggests that *Phaeocystis* is an important member of the high-latitude phytoplankton community." -- I question whether the authors claim that their model suggests something new here is actually in regard to something already known. Furthermore, this is already evidenced by the fact that the authors saw *Phaeocystis* as important enough to include in the model and write a manuscript about.

Figure 6: I believe the "p ratio" should be defined in the caption or removed.

**References:**

DiTullio, G.R., Smith, W.O., 1996. Spatial patterns in phytoplankton biomass and pigment distributions in the Ross Sea. J. Geophys. Res. Ocean. 101, 18467–18477. https://doi.org/10.1029/96JC00034

Kaufman, D.E., Friedrichs, M.A.M., Hemmings, J.C.P., Smith Jr., W.O., 2018. Assimilating bio-optical glider data during a phytoplankton bloom in the southern Ross Sea. Biogeosciences 15, 73–90. https://doi.org/10.5194/bg-15-73-2018

Mathot, S., Smith, W.O., Carlson, C.A., Garrison, D.L., Gowing, M.M., Vickers, C.L., 2000. Carbon partitioning within Phaeocystis Antarctica (Prymnesiophyceae) colonies in the Ross Sea, Antarctica. J. Phycol. 36, 1049–1056. https://doi.org/10.1046/j.1529-8817.2000.99078.x

Smith Jr., W.O., Carlson, C.A., Ducklow, H.W., Hansell, D.A., 1998. Growth dynamics of Phaeocystis antarctica- dominated plankton assemblages from the Ross Sea. Mar. Ecol. Prog. Ser. 168, 229–244. https://doi.org/10.3354/meps168229

---

## Author Response (AR3)

**Answer to referee #3:**

We thank referee #3 for reviewing our manuscript. His/Her valuable comments and suggestions have significantly improved the quality of our manuscript.

Below, we include our detailed answers to all comments and questions.

**Answers to general comments (GC):**

**General Comment #1:**
*[…] However, the attempt to include or address so many questions in this manuscript (how important is Phaeocystis to carbon export; what are the spatial and temporal patterns of Phaeocystis and diatom biomass; what are the drivers of Phaeocystis and diatoms' spatio-temporal patterns) makes it feel unfocused. The structure of the paper and section headings only partly help. I am left wondering, which comparisons do the authors feel are most important, most revealing, or most surprising? Although main conclusions are stated in the conclusion section and abstract, the attention paid to each analysis step and their findings in the body does not seem to match these main points.*
*[…]*
*The section headings do not seem consistent with the scope of what is being assessed, particularly which PFTs are addressed. Section 3.1 includes "phytoplankton" in the heading and assessed all PFTs. In contrast, sections 3.2 and 3.3 include "phytoplankton" in the heading but only addresses Phaeocystis and diatoms. Likewise, section 4.2 includes "phytoplankton" in the heading but seems to only discuss Phaeocystis. I recommend the authors revise the headings or make the content more consistent with the headings.*

**Answer to GC1:**
We thank the reviewer for this important comment, highlighting the need for clarification regarding the main focus of our manuscript and a better guidance of the reader throughout. In the revised manuscript, we have adapted the section titles in the result and discussion sections to better reflect each subsection's content. The sections are now named as follows:

**3.1 Phytoplankton biogeography and community composition in the SO**
**3.2 Phytoplankton phenology and the seasonal succession of *Phaeocystis* and diatoms**
**3.3 Drivers of the high-latitude biogeography and seasonal succession of *Phaeocystis* and diatoms**
**3.4 Quantifying the importance of *Phaeocystis* for the SO carbon cycle**
**4.1 Drivers of phytoplankton biogeography and the competition between *Phaeocystis* and diatoms**
**4.2 Biogeochemical implications of high-latitude SO *Phaeocystis* biogeography**

Further, we have modified the last paragraph of the introduction to better reflect the structure of the manuscript. The paragraph now reads:

"In this study, we investigate the competition between *Phaeocystis* and diatoms and its implications for carbon cycling using a regional coupled physical-biogeochemical-ecological model configured at eddy-permitting resolution for the SO (ROMS-BEC, Nissen et al., 2018). To address the missing link between SO phytoplankton biogeography, ecosystem function, and the SO carbon cycle, we have added *Phaeocystis* colonies as an additional PFT to the model, so that it includes all major known biogeochemically relevant phytoplankton types of the SO (see e.g. Buesseler et al., 1998; DiTullio et al., 2000). Using available observations, such as satellite-derived chlorophyll concentrations, carbon biomass and pigment data, we first validate the simulated phytoplankton distributions and community structure across the SO and then particularly focus on the temporal variability of diatoms and *Phaeocystis* in the high-latitude SO. After assessing the relative importance of bottom-up and topdown factors in controlling the contribution of *Phaeocystis* colonies and diatoms to total phytoplankton biomass over a complete annual cycle in the high-latitude SO, we show that the spatially and temporarily varying phytoplankton community composition leaves a distinct, PFT-specific imprint on upper ocean carbon cycling and POC export across the SO ."

**General Comment #2:**
*As I read, I wonder: why is the Ross Sea singled out for evaluation, aside from other coastal areas? Also, in some manuscript sections the Ross Sea is included in the comparisons (e.g. Figure 2, section 3.3 about drivers) and other sections do not include it (e.g. Table 3, section 3.4 about carbon cycle). Why is it only considered for some of the analyses? Without an explanation, these choices make the analysis seem arbitrary. The authors should explain why the Ross Sea is being used as a special study area and why/when it is or is not being included in analyses.*

**Answer to GC2:**
We acknowledge that some readers might be interested in the simulated dynamics in coastal regions other than the Ross Sea, but we decided on the latter as one of the focus areas in this study for several reasons: First, there is a tremendous body of literature available for the Ross Sea phytoplankton community, i.e. work discussing the competition between diatoms and *Phaeocystis*, thus making it a key focus area in the context of model evaluation. Further, we note that the simulated dynamics in ROMS-BEC in other coastal areas, e.g. the Amundsen Sea, are rather similar to those simulated in the Ross Sea (see Fig. 1 below).

[Figure]

**Fig. 1:** Same as Fig. 2 & 5 in the manuscript, but averaged over the Amundsen Sea.

Keeping the length and readability of our manuscript in mind and given the chosen circumpolar model domain covering the whole Southern Ocean up to 24°S, we decided to contrast the phytoplankton dynamics within one coastal area to that for the basin-wide assessment, rather than presenting results for different coastal areas. Admittedly, only showing results for the larger regions 30-90°S and 60-90°S in section 3.4 (carbon cycling) and Table 3 and not for the Ross Sea might be confusing for the reader. For completeness, we have therefore added Fig. 2 below to the supplementary material of the revised version of the manuscript, which shows the cycling of carbon in the Ross Sea in ROMS-BEC. Yet, in this context, the region 60-90°S is in many aspects representative for the Ross Sea (compare Fig. 6 & 7 of the manuscript to Fig. 2 below). Any difference in total POC production and the

partitioning amongst the PFTs and pathways is small and reflects the difference in phytoplankton community structure between 60-90°S and the Ross Sea, i.e., relatively more *Phaeocystis* (especially early in the season) and hence relatively more aggregation than grazing. In the revised version of the manuscript, we have therefore added the following statement in the captions of Fig. 6 & 7 in the main text:

"Results for the Ross Sea are comparable to those between 60-90°S (see Fig. S11)."

[Figure]

**Fig. 2:** Carbon cycling in the Ross Sea: The upper panel corresponds to Fig. 6 in the main text, the lower panels shows the quantities from Fig. 7 for the Ross Sea.

**General Comment #3:**
*The differences in carbon to chlorophyll ratio may have a substantial impact on some of the conclusions, and yet it seems to have been given little consideration. I refer the authors to several additional papers discussing C:Chl ratios for Phaeocystis and diatoms in the Ross Sea: DiTullio and Smith (1996), Smith et al. (1998), Mathot et al. (2000), Kaufman et al. (2018). I appreciate that calibration is difficult with such a large model, however, this seems to be an important limitation not discussed. I suggest the authors consider addressing it. Moreover, If the authors did train some of the model parameters before picking the 'best' values for their baseline run, it should be made clear whether or not model evaluation was done using the same or different data than was used for parameter training/tuning/calibration.*

**Answer to GC3:**
We thank the reviewer for raising this important point. In the *Baseline* setup of ROMS-BEC, the simulated monthly mean surface C:Chl ratios range from 23-103 mg C (mg chl)$^{-1}$ between 60-90°S, with peak ratios in summer. While this range is within the range suggested by observations (14-200 mg C (mg chl)$^{-1}$; DiTullio and Smith, 1996, Kaufmann et al., 2018, Mathot et al., 2000), we note that on average, in contrast to observations, ROMS-BEC currently simulates higher C:Chl ratios for *Phaeocystis* (38-103 mg C (mg chl)$^{-1}$) than for diatoms (23-66 mg C (mg chl)$^{-1}$). This model behavior is a direct result of the chosen model formulations and has implications for the presented results as outlined in the following.

In BEC, the base unit for calculations of phytoplankton biomass accumulation is carbon, and photoacclimation of phytoplankton, i.e., the amount of chlorophyll produced per unit of carbon, is then calculated following Geider et al. (1998). In this formulation, photoacclimation is calculated based on the PAR level and each PFT's specific growth rate, nitrogen uptake rate, maximum Chl:N ratio (thetaN_max, 4.0 mg chl (mmol N)$^{-1}$ for diatoms, 2.5 mg chl (mmol N)$^{-1}$ for *Phaeocystis*), sensitivity to changes in light intensity at low light levels (alphaPI, see Table 1 in the manuscript, alphaPI$^{phaeo}$ > alphaPI$^{diat}$ in the *Baseline* setup) and local Chl:C biomass ratio. As a result, the PFT with the higher alphaPI and lower thetaN_max (*Phaeocystis* in the *Baseline* setup) has lower rates of photoacclimation than the PFT with the lower alphaPI and higher thetaN_max (diatoms) for the same specific growth rate, nitrogen uptake rate, and PAR (see Fig. 3 below), resulting in lower Chl:C and thus higher C:Chl biomass ratios, explaining the higher average C:Chl ratio of *Phaeocystis* compared to that of diatoms in the *Baseline* setup of ROMS-BEC.

[Figure]

**Fig. 3**: Rates of chlorophyll assimilation as a function of PAR following the formulation of Geider et al. (1998). For the visualization of the functional relationship, a constant local chl:C ratio of 0.2 mg chl (mmol C)$^{-1}$, a nitrogen uptake rate of 1 mmol N m$^{-3}$ s$^{-1}$, and a specific growth rate of 1 d$^{-1}$ were used. The solid lines denote the photoacclimation rates with thetaNmax and alphaPI of diatoms (blue) and *Phaeocystis* (red) from the *Baseline* setup (see Table 1 of the manuscript). For the dashed lines, thetaNmax (purple) and alphaPI (light blue) of diatoms was replaced by the respective value of *Phaeocystis*.

As expected, varying the parameters alphaPI and thetaNmax of *Phaeocystis* will directly impact its simulated C:Chl ratios. As a response to a comment by reviewer 4, we have performed additional model experiments for the revised manuscript, in which we have, amongst others, varied these key parameters for photoacclimation by +/-50% (see Table 1 below, which was added to the revised supplementary material as Table S1). In these experiments, the simulated minimum C:Chl ratio between 60-90°S remains largely unchanged at 23-24 mg C (mg chl)$^{-1}$, but the maximum monthly mean ratio changes to 124 mg C (mg chl)$^{-1}$ (alphaPI150), 89 mg C (mg chl)$^{-1}$ (alphaPI50), 78 mg C (mg chl)$^{-1}$ (thetaNmax150), and 183 mg C (mg chl)$^{-1}$ (thetaNmax50), demonstrating the rather large sensitivity of the simulated C:Chl ratios to chosen model parameters. Yet, we note that in all these experiments, the average C:Chl biomass ratio of *Phaeocystis* remains larger than that of diatoms and that *absolute* biomass concentrations of the two PFTs change substantially as well (by more than 90% compared to the *Baseline* setup between 60-90°S, see Fig. 4 below), demonstrating the difficulty to

tune both the C:Chl ratios of Southern Ocean phytoplankton and each PFT's absolute biomass concentrations with this model formulation and a complex model like ROMS-BEC. In fact, Buitenhuis & Geider (2010) point out that for iron-limited regions such as the Southern Ocean, the parameterizations by Geider et al (1998) should be modified to include the effects of iron (rather than nitrogen as in Geider et al., 1998) on phytoplankton Fe:C and Chl:C ratios, and the effect of this modification for our model results presented here should be assessed in future work.

As biomass concentrations are the basis for the analysis presented here, our results are impacted by any bias in the C:Chl biomass ratio of *Phaeocystis* and diatoms. Specifically, any bias in photoacclimation affects the simulated carbon fields through the impact of the C:Chl ratio on the light limitation factor (Eq. B9 in appendix of manuscript). In this context, a lower C:Chl biomass ratio of *Phaeocystis*, as suggested by observations, would result in a lower light limitation factor (less light limitation, see Eq. B9 in appendix of manuscript), suggesting that we possibly underestimate *Phaeocystis* carbon biomass and its relative importance in the phytoplankton community. Furthermore, a decrease in the light limitation of *Phaeocystis* growth would impact its simulated seasonality and the succession patterns of PFTs, with the lower light limitation being especially critical early in the growth season in the Ross Sea, where/when our model suggests differences in light limitation between *Phaeocystis* and diatoms to be an important driver for the simulated succession from *Phaeocystis* to diatoms throughout the season (see Fig. 5 in the manuscript).

In summary, we acknowledge the sensitivity of our model results to biases in the simulated C:Chl biomass ratios of *Phaeocystis*. In particular, recent advances in our understanding of differences in the adaptation between *Phaeocystis* and diatoms to low levels of light, iron, and temperature warrant a reassessment of model parametrizations and a closer assessment of the simulated Chl:C ratios in future work (see e.g. Strzepek et al., 2019). We have added a statement along these lines in the caveat section:

"As iron-light interactions are key for the simulated Fe:C and Chl:C ratios of SO phytoplankton (Buitenhuis and Geider, 2010) and in light of more recent advances regarding our understanding of the adaptation of SO phytoplankton to persisting low levels of light, iron, and temperature (Strzepek et al., 2019), a reassessment of model parametrizations describing phytoplankton growth and photoacclimation is advisable in future work."

Furthermore, the description and assessment of the new parameter sensitivity experiments is included as section S2 in the revised supplementary material of the manuscript:

Table 1: Overview of parameter sensitivity simulations, varying parameters by +/- 50%. PA=Phaeocystis, D=diatoms. See also Table 1 in the submitted manuscript.

| Run Name | Description | |
|---|---|---|
| Topt150 | Increase $T_{opt}^{PA}$ by 50% | } Param_Topt |
| Topt50 | Decrease $T_{opt}^{PA}$ by 50% | |
| kFe150 | Increase $k_{Fe}^{PA}$ by 50% | } Param_kFe |
| kFe50 | Decrease $k_{Fe}^{PA}$ by 50% | |
| alphaPI150 | Increase $\alpha_{PI}^{PA}$ by 50% | } Param_alphaPI |
| alphaPI50 | Decrease $\alpha_{PI}^{PA}$ by 50% | |
| mortality150 | Increase $\gamma_{m,0}^{PA}$ by 50% | } Param_mortality |
| mortality50 | Decrease $\gamma_{m,0}^{PA}$ by 50% | |
| aggregation150 | Increase $\gamma_{a,0}^{PA}$ by 50% | } Param_aggregation |
| aggregation50 | Decrease $\gamma_{a,0}^{PA}$ by 50% | |
| grazing150 | Increase $\gamma_{g,max}^{PA}$ by 50% | } Param_grazing |
| grazing50 | Decrease $\gamma_{g,max}^{PA}$ by 50% | |
| thetaNmax50 | Increase $\theta_{chl:N,max}^{PA}$ by 50% | } Param_thetaNmax |
| thetaNmax50 | Decrease $\theta_{chl:N,max}^{PA}$ by 50% | |

[Figure]

**Fig. 4:** Annual mean surface chlorophyll concentrations of all phytoplankton (Chl), *Phaeocystis* (PA), and diatoms (D) in the parameter sensitivity simulations (see Table 1 above) relative to the *Baseline* simulation. The model output is averaged over a) 60-90°S and b) the Ross Sea.

With regard to the reviewer's comment on the model tuning, we note that the goal of the tuning of BEC model parameters was to identify that set of phytoplankton growth and loss parameters within the range of observational uncertainty (see Table 1 in the manuscript) that yielded the best agreement in the simulated spatio-temporal variability in phytoplankton community composition between ROMS-BEC and available observational data of carbon biomass concentrations and community structure (e.g., Vogt et al., 2012, Swan et al., 2016). Due to the scarcity in observational data in the Southern Ocean, the complete set of carbon biomass concentrations and community composition was used in the tuning exercise. In the revised manuscript, we have clarified the text in section 2.1 as follows:

"Generally, model parameters for *Phaeocystis* in the *Baseline* setup are chosen to represent the colonial form of *Phaeocystis* whenever information is available in the literature (see e.g. review by Schoemann et al., 2005), and model parameters were tuned to maximize the model-data agreement in the spatio-temporal variability of the phytoplankton community structure between ROMS- BEC and all available observations (see also section 2.3.1)."

**Answers to specific comments (SC):**

**SC1:** The last paragraph of the introduction does not accurately reflect the organization of the paper. This seems like a great place for the authors to more coherently state the purpose of the analyses.
Please see our answer to GC1. We have revised the last paragraph of the introduction.

**SC2:** Line 95: Perhaps it is just me, but I am confused by this sentence. Also, the implication that the model provides "a correct representation of SO phytoplankton biogeography" (emphasis added) seems very presumptuous.
We have rephrased the last sentence of the introduction as follows in the revised version of the manuscript: "[…] we show that the spatially and temporarily varying phytoplankton community composition leaves a distinct imprint on upper ocean carbon cycling and POC export across the SO."

**SC3:** In section 2.1, the authors refer to a "baseline" simulation before it is described. It would be helpful for the authors to refrain from referencing baseline before it is defined.
As the description of the *Phaeocystis* PFT in section 2.1 and the presented parameter choices refer to the *Baseline* setup of ROMS-BEC, we have introduced the "*Baseline*" setup earlier in the revised version of the manuscript to clarify this for the reader. The respective part of section 2.1 now reads:

"Generally, model parameters for *Phaeocystis* in the *Baseline* setup are chosen to represent the colonial form of *Phaeocystis* whenever information is available in the literature (see e.g. review by Schoemann et al., 2005)."

**SC4:** Sect. 2.3: I wonder what the authors mean by "analysis framework" in the section title? To me, growth rate ratios are not an analysis framework, but rather simply a diagnostic variable.
We agree with the reviewer that the growth ratios are diagnostic tools. To avoid confusion, we have renamed section 2.3 to "Data and diagnostics used in the model assessment" in the revised manuscript.

**SC5:** Sect. 2.3.2: The authors should define Betas in the text.
We have added a definition of the "betas" in the revised version of the manuscript. It now reads:

"Further, $\beta_T$, $\beta_N$, and $\beta_I$ describe the logarithmic ratio of the limitation by temperature, nutrients, and light of growth by diatoms and *Phaeocystis*. Thereby, these terms denote the log-normalized contribution of each environmental factor to the simulated relative growth ratio. At high-latitudes south of $60°$ S, the ratio of the nutrient limitation of growth $\beta_N$ corresponds to that of the iron limitation $\beta_{Fe}$ in our model (Fig. S1)."

**SC6:** Line 279, and elsewhere: I think "N" is being used to represent both diazotrophs and nutrients. The authors should restrict its meaning to only one or the other.
We thank the reviewer for this comment. In the revised manuscript, we now use "DZ" to denote diazotrophs in Table 1 and keep "N" for nutrients throughout the text.

**SC7:** Lines 318-323: I think the bias could also be due to poor calibration, especially of the newly introduced Phaeocystis group.
The positive chlorophyll bias at high latitudes was already simulated in the 4-PFT setup of ROMS-BEC (similar in magnitude, see Nissen et al., 2018), suggesting that the implementation of *Phaeocystis* is not the reason for this model behavior. In fact, while a fraction of this bias might be due to parameter choices of the different phytoplankton groups (not only *Phaeocystis*, see e.g. also the overestimated temperature-limited growth rate of diatoms at low temperatures in the model, see Fig. A1 of the manuscript), extensive testing with the 4-PFT setup in Nissen et al. (2018) has suggested that missing model complexity in the zooplankton compartment might drive this bias, as the implementation of more zooplankton functional types has been shown to substantially alter the phytoplankton-zooplankton coupling and hence the simulated chlorophyll concentrations in ecosystem models (Le Quéré et al., 2016).

**SC8:** Line 331: "compared to the 4-PFT"
Changed as suggested.

**SC9:** Line 361-362: "Our model suggests that Phaeocystis is an important member of the high-latitude phytoplankton community." -- I question whether the authors claim that their model suggests something new here is actually in regard to something already known. Furthermore, this is already evidenced by the fact that the authors saw Phaeocystis as important enough to include in the model and write a manuscript about.
We thank the reviewer for this comment. We admit that the chosen formulation does not reflect the state of knowledge before our work. The sentence now reads:

"Overall, our model agrees with observational data that *Phaeocystis* is an important member of the high-latitude phytoplankton community."

**SC10:** Figure 6: I believe the "p ratio" should be defined in the caption or removed.
We have adapted the figure caption in the revised manuscript, and it now reads:

"**Figure 6**. Pathways of particulate organic carbon (POC) formation in the *Baseline* simulation of ROMS-BEC averaged annually over a) 30-90° S and b) 60-90° S. The green and yellow boxes show the relative contribution (%) of *Phaeocystis*, diatoms, coccolithophores, small phytoplankton (SP), and zooplankton (Zoo) to the combined phytoplankton and zooplankton biomass (green) and total POC production (yellow) in the top 100 m, respectively. The arrows denote the relative contribution of the different POC production pathways associated with each PFT (black = grazing by zooplankton, grey = aggregation, blue = non-grazing mortality), given as % of total NPP in the top 100 m. Numbers are printed if ≥0.1% and rounded to the nearest integer if >1%. The sum of all arrows gives the POC production efficiency, i.e., the fraction of NPP which is converted into sinking POC upon biomass loss (p ratio). Note that diazotrophs are not included in this figure due to their minor contribution to NPP in the model domain. «

**Cited literature**

Buesseler, K. O. (1998). The decoupling of production and particulate export in the surface ocean. *Global Biogeochemical Cycles*, *12*(2), 297–310. https://doi.org/10.1029/97GB03366

Buitenhuis, E. T., & Geider, R. J. (2010). A model of phytoplankton acclimation to iron-light colimitation. *Limnology and Oceanography*, *55*(2), 714–724. https://doi.org/10.4319/lo.2009.55.2.0714

DiTullio, G. R., & Smith, W. O. (1996). Spatial patterns in phytoplankton biomass and pigment distributions in the Ross Sea. *Journal of Geophysical Research: Oceans*, *101*(C8), 18467–18477. https://doi.org/10.1029/96JC00034

DiTullio, G. R., Grebmeier, J. M., Arrigo, K. R., Lizotte, M. P., Robinson, D. H., Leventer, A., Barry, J. P., VanWoert, M. L., & Dunbar, R. B. (2000). Rapid and early export of Phaeocystis antarctica blooms in the Ross Sea, Antarctica. *Nature*, *404*(6778), 595–598. https://doi.org/10.1038/35007061

Geider, R. J., MacIntyre, H. L., & Kana, T. M. (1998). A dynamic regulatory model of phytoplanktonic acclimation to light, nutrients, and temperature. *Limnology and Oceanography*, *43*(4), 679–694. https://doi.org/10.4319/lo.1998.43.4.0679

Kaufman, D. E., Friedrichs, M. A. M., Hemmings, J. C. P., & Smith Jr., W. O. (2018). Assimilating bio-optical glider data during a phytoplankton bloom in the southern Ross Sea. *Biogeosciences*, *15*(1), 73–90. https://doi.org/10.5194/bg-15-73-2018

Le Quéré, C., Buitenhuis, E. T., Moriarty, R., Alvain, S., Aumont, O., Bopp, L., … Vallina, S. M. (2016). Role of zooplankton dynamics for Southern Ocean phytoplankton biomass and global biogeochemical cycles. *Biogeosciences*, *13*(14), 4111–4133. https://doi.org/10.5194/bg-13-4111-2016

Mathot, S., Smith, W. O., Carlson, C. A., Garrison, D. L., Gowing, M. M., & Vickers, C. L. (2000). Carbon partitioning within Phaeocystis Antarctica (Prymnesiophyceae) colonies in the Ross Sea, Antarctica. *Journal of Phycology*, *36*(6), 1049–1056. https://doi.org/10.1046/j.1529-8817.2000.99078.x

Nissen, C., Vogt, M., Münnich, M., Gruber, N., & Haumann, F. A. (2018). Factors controlling coccolithophore biogeography in the Southern Ocean. *Biogeosciences*, *15*(22), 6997–7024. https://doi.org/10.5194/bg-15-6997-2018

Schoemann, V., Becquevort, S., Stefels, J., Rousseau, V., & Lancelot, C. (2005). Phaeocystis blooms in the global ocean and their controlling mechanisms: a review. *Journal of Sea Research*, *53*(1–2), 43–66. https://doi.org/10.1016/j.seares.2004.01.008

Smith, W. O., Carlson, C., Ducklow, H., & Hansell, D. (1998). Growth dynamics of Phaeocystis antarctica-dominated plankton assemblages from the Ross Sea. *Marine Ecology Progress Series*, *168*, 229–244. https://doi.org/10.3354/meps168229

Strzepek, R. F., Boyd, P. W., & Sunda, W. G. (2019). Photosynthetic adaptation to low iron, light, and temperature in Southern Ocean phytoplankton. *Proceedings of the National Academy of Sciences*, *116*(10), 4388–4393. https://doi.org/10.1073/pnas.1810886116

Swan, C. M., Vogt, M., Gruber, N., & Laufkoetter, C. (2016). A global seasonal surface ocean climatology of phytoplankton types based on CHEMTAX analysis of HPLC pigments. *Deep Sea Research Part I: Oceanographic Research Papers*, *109*, 137–156. https://doi.org/10.1016/j.dsr.2015.12.002

Vogt, M., O'Brien, C., Peloquin, J., Schoemann, V., Breton, E., Estrada, M., Gibson, J., Karentz, D., Van Leeuwe, M. A., Stefels, J., Widdicombe, C., & Peperzak, L. (2012). Global marine plankton functional type biomass distributions: *Phaeocystis* spp. *Earth System Science Data*, *4*(1), 107–120. https://doi.org/10.5194/essd-4-107-2012

**Answer to referee #4:**

We thank referee #4 for reviewing our manuscript in such great detail. His/Her valuable comments and suggestions have significantly improved the quality of our manuscript.

Below, we include our detailed answers to all comments and questions.

**Answers to general comments (GC):**

**General Comment #1:**
*My main concern with the paper is that given that the outcome is highly sensitive to the model parameterization, the authors should place a very high priority on reporting uncertainties throughout. In its current form uncertainties are seldom reported. I also recommend that the authors conduct more sensitivity tests that examine how the model output changes when more than one parameter is perturbed. The model framework constructed by the authors would be ideal for a thorough exploration of how NPP and export vary with changing parameters, and presents an excellent opportunity to isolate the environmental conditions where model uncertainty is greatest. Additionally, cellular Fe:C ratios should be included in these sensitivity experiments. As Fe:C will make a very large difference to the POC calculated, and is central to the paper title, it is necessary that the authors investigate how Fe:C, which is sensitive to both light and iron, can change the model outcome.*
*I believe that this paper will be much more useful to biogeochemists if error bars can be placed on the model estimates during major revisions, with particular consideration of parameter sensitivity to light and iron limitation.*

**Answer to GC1:**
We thank the reviewer for this comment, which regards the broader topics of "model uncertainty" and "iron cycling" in ROMS-BEC, which we will separately address in the following.

**Reporting uncertainties in an ecosystem model:**
We agree with the reviewer that any quantitative information provided in our manuscript, such as the simulated distribution of chlorophyll, the integrated net primary production (NPP) and the contribution of the different phytoplankton functional types (PFTs) to NPP or the export of particulate organic carbon (POC), is associated with an uncertainty which should ideally be quantified. However, we believe it is important to distinguish between model uncertainty and model sensitivity in the current case.

In any biogeochemical model, the ***model uncertainty*** of any of the above biogeochemical quantities is defined as the variability of the respective quantity across *equally plausible realizations* of the model. Such an uncertainty results from several sources, namely the sensitivity of e.g. the chlorophyll fields to 1) the underlying physical fields (here simulated by ROMS), 2) the chosen structural complexity of the biogeochemical model (here BEC), and 3) the chosen model parameters describing e.g. the growth and loss of phytoplankton biomass in the biogeochemical model (BEC). To the best of our knowledge, there is currently no methodology available in the literature to *computationally efficiently* quantify the combined uncertainty of biogeochemical quantities due to these three factors in numerical ocean biogeochemical models with the complexity of ROMS-BEC, and this quantification is therefore currently not common practice in the biogeochemical modeling community. In the field of climate sciences, *equally plausible* model realizations are often achieved by perturbing the initial conditions, by perturbing certain parameters of a given model, or by comparing a number of different models with different structural complexity and/or model parameters in Monte-Carlo type analysis frameworks (see e.g. IPCC or the CMIP efforts, e.g. Bopp et al., 2013), but the total uncertainty of biogeochemical quantities like NPP is typically not systematically quantified for individual model studies using a single model. Given the complexity of our biogeochemical model, including hundreds of model parameters, and the comparatively high resolution used here, which results in hundreds of cores needed for one single simulation, an assessment of the total model uncertainty is computationally both

beyond the scope of the current work and beyond the limited computational resources available for this work.

However, we argue that the current set-up is suitable for the assessment of the ***model sensitivity*** of the above-mentioned bulk properties to the input parameters of the biogeochemical model, which is what we believe the reviewer had in mind in his/her comment. Hence, we describe how we have 1) quantified the *spatio-temporal variability* of the contribution of the PFTs to NPP and POC export in the model and 2) more systematically quantified the *sensitivity* of the simulated biomass distributions in ROMS-BEC to chosen *Phaeocystis* model parameters in the revised version of the manuscript.

In our study, the results of our *Baseline* setup represent the "*best*" model solution in terms of its agreement with the spatio-temporal distribution and variability of PFTs with available observations (i.e., *Baseline* has the lowest model-data-misfit), with all model parameters chosen within the ranges reported in the literature (see method section of the manuscript for references), and tuned to fit observational evidence. As a means to quantify the spatial variability in the contributions of different PFTs to the seasonally averaged chlorophyll distributions and annually integrated NPP and POC export, we add the area-weighted standard deviation of these quantities to the respective reported means in the abstract, in Table 3, and throughout the result section in the revised version of the manuscript:

Abstract:

all biogeochemically relevant Southern Ocean phytoplankton types. We find that *Phaeocystis* contribute $46\pm21\%$ ($1\sigma$ in space) and $40\pm20\%$ to annual NPP and POC export south of $60°$ S, respectively, making them an important contributor to high-latitude carbon cycling. In our simulation, the relative importance of *Phaeocystis* and diatoms is mainly controlled by spatio-

Table 3:

**Table 3.** Comparison of ROMS-BEC based phytoplankton biomass, production, and export estimates with available observations (given in parentheses). Data sources are given below the Table. The reported uncertainty of the contribution of the PFTs to the simulated integrated NPP corresponds to the area-weighted spatial variability of each PFT's contribution to annual NPP ($1\sigma$).

| | | ROMS-BEC (Data) | |
| --- | --- | --- | --- |
| | | 30-90° S | 60-90° S |
| Surface chlorophyll biomass | total, annual mean [Gg chl] | 40.8 (34.5[a]) | 17.1 (9.5[a]) |
| Diatom carbon biomass | 0-200m, annual mean [Pg C] | 0.059 (global[b]: 0.10-0.94) | 0.015 |
| *Phaeocystis* carbon biomass | 0-200m, annual mean [Pg C] | 0.019 (global[b]: 0.11-0.71) | 0.010 |
| Coccolithophore carbon biomass | 0-200m, annual mean [Pg C] | 0.012 (global[b]: 0.001-0.03) | 0.001 |
| NPP | $Pg\ C\ yr^{-1}$ | 17.2 (12.1-12.5[c]) | 3.0 (0.68-1.7[c]) |
| | Diatoms [%] | 52.0 ($\pm26.2$) | 49.1 ($\pm19.9$) |
| | *Phaeocystis* [%] | 15.3 ($\pm24.5$) | 45.8 ($\pm20.7$) |
| | Coccolithophores [%] | 14.6 ($\pm15.3$) | 0.7 ($\pm1.0$) |
| | SP [%] | 17.2 ($\pm16.1$) | 4.5 ($\pm1.9$) |
| POC export at 100m | $Pg\ C\ yr^{-1}$ | 3.1 (2.3-2.96[d]) | 0.62 (0.21-0.24[d]) |

[a] Monthly climatology from MODIS Aqua (2002-2016, NASA-OBPG, 2014a), SO algorithm (Johnson et al., 2013)

[b] The reported estimates from the MAREDAT data base in Buitenhuis et al. (2013) are global estimates of phytoplankton biomass.

[c] Monthly climatology from MODIS Aqua VGPM (2002-2016, Behrenfeld and Falkowski, 1997; O'Malley, last access: 16 May 2016), NPP climatology from Buitenhuis et al. (2013, 2002-2016)

[d] Monthly output from a biogeochemical inverse model (Schlitzer, 2004) and a data-assimilated model (DeVries and Weber, 2017).

Section 3.1:

dinal trend (see Fig. 2b-d and section 2.3.1, Swan et al., 2016). Averaged over 30-90° S (60-90° S), the simulated relative

360   contributions of *Phaeocystis*, diatoms, and coccolithophores to total chlorophyll in summer are 20±28% (33±34%; subarea mean as shown in Fig. 2b & c ±1$\sigma$ in space), 68±33% (64±33%), and 5±17% (<1±2%), respectively, in good agreement with the CHEMTAX climatology (28% (27%), 46% (48%), and 3% (1%), respectively). Acknowledging the uncertainty in the attribution of the group "Other" in the CHEMTAX data to a model PFT ("Other" includes dinoflagellates, cryptophytes, and chlorophytes here, see section 2.3.1), the model also captures the seasonal evolution of the relative importance of *Phaeocystis*

365   and diatoms reasonably well, both averaged over 30-90° S (Fig. 2b) and at high SO latitudes (Fig. 2c-d). The model overestimates the contribution of *Phaeocystis* in fall (39±14% as compared to 24% in CHEMTAX) and spring (51±22% as compared to 28%) between 60-90° S and in the Ross Sea, respectively (Fig. 2c-d), but the limited number of data points available in the CHEMTAX climatology in this area and the uncertainty in the attribution of pigments in CHEMTAX to the *Phaeocystis* PFT in ROMS-BEC have to be noted (see section 2.3.1).

Section 3.4:

*Phaeocystis* is an important member of the SO phytoplankton community in our model, particularly south of 60°S, where it contributes 46±21% and 40±20% to total annual NPP and POC formation, respectively (Table 3 & Fig. 6). Even when

500   considering the entire region south of 30°S, the contribution of *Phaeocystis* to NPP (15±24%) and POC production (16±22%) is sizeable. The simulated spatial differences in phytoplankton community structure have direct implications for the fate of

Additionally, we have performed a second set of model parameter sensitivity experiments in order to more systematically quantify the *sensitivity* of the simulated distributions of *Phaeocystis* and diatoms and integrated estimates of NPP and POC export in ROMS-BEC to *Phaeocystis* model parameter choices (section S2 of the revised supplementary material). To this end, we have systematically increased/decreased all key *Phaeocystis* parameters by 50%, allowing for an objective ranking of model sensitivities. We varied the following seven parameters of *Phaeocystis* (resulting in a total of 14 new simulations, see Table 1 below): the temperature optimum $T_{opt}$, the half-saturation constant of iron $k_{Fe}$, the maximum chl:N ratio $\theta_{chl:N, max}$, alphaPI, the linear mortality rate, the quadratic mortality rate (aggregation), and the maximum grazing rate of zooplankton on *Phaeocystis*. We note, however, that this systematic assessment does not result in equally plausible realizations of ROMS-BEC, as some of these changes will result in unrealistic biomass distributions of *Phaeocystis* and diatoms when compared to available observations. Therefore, we refer to these experiments as an assessment of *sensitivity* rather than *uncertainty*.

Table 1 (included in the revised manuscript as table S1): Overview of parameter sensitivity simulations, varying parameters by +/- 50%. PA=Phaeocystis, D=diatoms. See also Table 1 in the submitted manuscript.

| Run Name | Description | |
| --- | --- | --- |
| Topt150 | Increase $T_{opt}^{PA}$ by 50% | } Param_Topt |
| Topt50 | Decrease $T_{opt}^{PA}$ by 50% | |
| kFe150 | Increase $k_{Fe}^{PA}$ by 50% | } Param_kFe |
| kFe50 | Decrease $k_{Fe}^{PA}$ by 50% | |
| alphaPI150 | Increase $\alpha_{PI}^{PA}$ by 50% | } Param_alphaPI |
| alphaPI50 | Decrease $\alpha_{PI}^{PA}$ by 50% | |
| mortality150 | Increase $\gamma_{m,0}^{PA}$ by 50% | } Param_mortality |
| mortality50 | Decrease $\gamma_{m,0}^{PA}$ by 50% | |
| aggregation150 | Increase $\gamma_{a,0}^{PA}$ by 50% | } Param_aggregation |
| aggregation50 | Decrease $\gamma_{a,0}^{PA}$ by 50% | |
| grazing150 | Increase $\gamma_{g,max}^{PA}$ by 50% | } Param_grazing |
| grazing50 | Decrease $\gamma_{g,max}^{PA}$ by 50% | |
| thetaNmax50 | Increase $\theta_{chl:N, max}^{PA}$ by 50% | } Param_thetaNmax |
| thetaNmax50 | Decrease $\theta_{chl:N, max}^{PA}$ by 50% | |

We quantify the sensitivity $S$ of any target variable $A$ (here A being one of the following targets: total phytoplankton, *Phaeocystis*, and diatom chlorophyll concentrations, total NPP, and POC export across 100m) to changes in the parameter $X$ as follows, allowing for a ranking of the seven sets of simulations by the magnitude of the sensitivity (see Table 2 below, note that the resulting ranking of model experiments with regard to chlorophyll distributions is insensitive to the choice of chlorophyll rather than carbon here, as the simulated changes in carbon biomass fields of the PFTs are qualitatively similar to those in chlorophyll; not shown):

$$S_X^A = 100 \cdot \frac{A_{X150} - A_{X50}}{A_{X\,Baseline}} \qquad (1)$$

As expected (see also Nissen et al., 2018), we find that both total chlorophyll concentrations and chlorophyll levels of *Phaeocystis* and diatoms are highly sensitive to parameters describing the growth and loss of *Phaeocystis* biomass, with increases of up to 700% (grazing50) and declines of up to >90% (Topt50, thetaNmax50) in *Phaeocystis* biomass between 60-90°S for a 50% change in the associated parameters (see Fig. 1 below). In general, any decline/increase in *Phaeocystis* chlorophyll biomass is associated with an increase/decline in diatom chlorophyll biomass, pointing to the direct competition for resources of these two phytoplankton types at high SO latitudes. Yet, the biomass compensation is not always complete due to non-linearities in the model system (e.g. food web feedbacks), resulting in changes of up to 70% (grazing150) in total chlorophyll levels upon changes in *Phaeocystis* parameters. The ranking of model sensitivities between 60-90°S reveals the highest sensitivity of *Phaeocystis* and diatom chlorophyll concentrations to the maximum grazing rate, the maximum chl:N ratio, the initial slope of the photosynthesis-irradiance curve, and the temperature optimum of *Phaeocystis* growth (Param_grazing, Param_thetaNmax, Param_alphaPI, Param_Topt in Table 1 above and Table 2 below). In comparison, the opposed changes in *Phaeocystis* and diatom chlorophyll levels (see Fig. 1 below) result in lower sensitivities of total chlorophyll levels to changes in *Phaeocystis* parameters in general and a lower ranking of the temperature optimum and thetaNmax experiments in particular (Param_Topt and Param_thetaNmax in Table 2 below).

Table 2 (included in the revised manuscript as table S2): Ranking of the parameter sensitivity experiments by the absolute sensitivity of annual mean total surface chlorophyll ($|S_X^{Chl}|$), *Phaeocystis* chlorophyll ($|S_X^{ChlPA}|$), and diatom chlorophyll (($|S_X^{ChlD}|$) to a +/-50% change in the model parameter X relative to the Baseline setup of ROMS-BEC between 60-90°S and in the Ross Sea, respectively. The sensitivity S (%) is quantified using Eq. 1.
See Table 1 above for details on the experimental setup and Fig. S12 for details on the resulting chlorophyll fields in ROMS-BEC in each experiment. Note that the simulated changes in carbon biomass fields are qualitatively similar to those of chlorophyll (not shown) and that the ranking shown here is therefore insensitive to the choice of chlorophyll in the analysis.

| | Ranking ($|S_X^{Chl}|$ in %) | Ranking ($|S_X^{ChlPA}|$ in %) | Ranking ($|S_X^{ChlD}|$ in %) |
|---|---|---|---|
| **60-90°S** | | | |
| | 1. Param_alphaPI (63.6) | 1. Param_grazing (693.1) | 1. Param_alphaPI (153.4) |
| | 2. Param_grazing (48.3) | 2. Param_thetaNmax (390.9) | 2. Param_thetaNmax (149.6) |
| | 3. Param_mortality (40.6) | 3. Param_Topt (306.8) | 3. Param_Topt (132.7) |
| | 4. Param_kFe (39.8) | 4. Param_alphaPI (259.4) | 4. Param_grazing (128.3) |
| | 5. Param_Topt (37.5) | 5. Param_kFe (209.1) | 5. Param_kFe (109.6) |
| | 6. Param_thetaNmax (33.0) | 6. Param_mortality (178.0) | 6. Param_mortality (101.8) |
| | 7. Param_aggregation (6.4) | 7. Param_aggregation (65.1) | 7. Param_aggregation (10.2) |
| **Ross Sea** | | | |
| | 1. Param_alphaPI (76.3) | 1. Param_grazing (360.3) | 1. Param_thetaNmax (189.1) |
| | 2. Param_mortality (53.3) | 2. Param_thetaNmax (288.9) | 2. Param_alphaPI (189.1) |
| | 3. Param_thetaNmax (46.4) | 3. Param_Topt (194.2) | 3. Param_Topt (142.1) |
| | 4. Param_Topt (41.6) | 4. Param_alphaPI (188.3) | 4. Param_grazing (129.8) |
| | 5. Param_kFe (41.3) | 5. Param_kFe (126.2) | 5. Param_mortality (126.7) |
| | 6. Param_grazing (19.2) | 6. Param_mortality (114.8) | 6. Param_kFe (114.3) |
| | 7. Param_aggregation (12.3) | 7. Param_aggregation (59.5) | 7. Param_aggregation (9.0) |

[Figure]

**Fig. 1** (included in the revised manuscript as Fig. S13): Annual mean surface chlorophyll concentrations of all phytoplankton (Chl), *Phaeocystis* (PA), and diatoms (D) in the parameter sensitivity simulations (see Table 1 above) relative to the *Baseline* simulation. The model output is averaged over a) 60-90°S and b) the Ross Sea.

In comparison to the ranking of model experiments for total chlorophyll, the model sensitivities for NPP and POC export across 100 m are similar in magnitude both between 60-90°S and in the Ross Sea (20-90%, compare Table 2 above & Table 3 below). Additionally, the ranking of model experiments for NPP and POC export reveals only small differences to the ranking of model sensitivities for total chlorophyll: While the experiments Param_alphaPI and Param_grazing consistently rank amongst the top two most sensitive experiments for NPP and POC export and between 60-90°S for total chlorophyll concentrations, the experiments Param_mortality/Param_Topt are less/more important for NPP and POC than for total chlorophyll levels in ROMS-BEC (compare Table 2 & 3). In summary, this demonstrates the large model sensitivity of bulk biogeochemical quantities to parameter choices describing the temperature and light dependence of Phaeocystis growth and zooplankton grazing.

Table 3 (included in the revised manuscript as Table S3): Ranking of the parameter sensitivity experiments by the absolute sensitivity of annually integrated NPP ($|S_X^{NPP}|$) and POC export across 100 m ($|S_X^{POC100m}|$) to a +/-50% change in the model parameter X relative to the Baseline setup of ROMS-BEC between 60-90°S and in the Ross Sea, respectively. The sensitivity S (%) is quantified using Eq. 1. See Table 1 above for details on the experimental setup.

| | Ranking ($|S_X^{NPP}|$ in %) | Ranking ($|S_X^{POC100m}|$ in %) |
|---|---|---|
| **60-90°S** | | |
| | 1. Param_grazing (68.4) | 1. Param_grazing (86.4) |
| | 2. Param_alphaPI (46.7) | 2. Param_alphaPI (35.4) |
| | 3. Param_Topt (43.6) | 3. Param_Topt (26.7) |
| | 4. Param_kFe (23.6) | 4. Param_mortality (12.9) |
| | 5. Param_thetaNmax (23.4) | 5. Param_kFe (11.6) |
| | 6. Param_mortality (11.6) | 6. Param_thetaNmax (10.7) |
| | 7. Param_aggregation (7.6) | 7. Param_aggregation (1.4) |
| | | |
| **Ross Sea** | | |
| | 1. Param_grazing (55.6) | 1. Param_grazing (71.9) |
| | 2. Param_alphaPI (48.5) | 2. Param_alphaPI (39.0) |
| | 3. Param_Topt (44.0) | 3. Param_Topt (26.9) |
| | 4. Param_thetaNmax (24.7) | 4. Param_thetaNmax (11.9) |
| | 5. Param_kFe (20.4) | 5. Param_kFe (10.5) |
| | 6. Param_aggregation (11.6) | 6. Param_mortality (10.2) |
| | 7. Param_mortality (8.3) | 7. Param_aggregation (2.6) |

Given computational constraints, we focused on *Phaeocystis* parameters here, but acknowledge that the simulated biomass distributions, NPP, and POC export are equally sensitive to parameters of the other PFTs and that the quantitative results might change when varying multiple parameters at once. We agree with the reviewer that this type of sensitivity experiments would ideally be performed in addition to the single-parameter perturbations, but given the number of available parameters in ROMS-BEC, the systematic assessment of these co-variation landscapes is beyond the scope of this study. We have added the above Tables and Figure as Tables S1, S2 & S3 and Fig. S13 to the revised supplementary information, together with the motivation of the experiments and the description of the main findings discussed above. Furthermore, we have added the new simulations to Table 2 and section 2.2 in the main text, as well as added small revisions to the appendix of the revised manuscript:

[revised manuscript text omitted]

740 of sensitivity experiments (competition experiments, runs 1-8 in Table 2). Overall, the simulated surface *Phaeocystis* biomass concentrations change by $\gtrsim$ ±50% for each of the experiments in the high-latitude SO (Fig. A2). Between 60-90° S and in the Ross Sea, the largest increases in *Phaeocystis* biomass concentrations are simulated for THETA_N_MAX (+332% and +217%, respectively, Fig. A2b & c) and AGGREGATION (+112% and +96%, respectively, Fig. A2b & c), whereas the strongest decline is simulated for ALPHA$_{PI}$ (-76% and -87%, respectively, Fig. A2b & c). As a response to changes in *Phaeocystis* parameters,

745 diatom biomass changes overall more than that of SP on a basin scale, suggesting *Phaeocystis* is indeed mostly competing with diatoms for resources in the high-latitude SO. Between 60-90° S, the magnitude of change is similar for the experiments TEMPERATURE (-73%), ALPHA$_{PI}$ (-76%), and IRON (+70%), while in the Ross Sea, the response in IRON is substantially smaller (+17%) than that for the other two experiments (-82% and -87% for TEMPERATURE and ALPHA$_{PI}$, respectively; Fig. A2b & c). This supports our findings from section 3.3, namely that the difference in light sensitivity between *Phaeocystis*

750 and diatoms is more important in coastal areas than on a basin scale in controlling the relative importance of *Phaeocystis* for total phytoplankton biomass.

**Iron cycling in BEC:**

We thank the reviewer for raising this important point. In the following, we will describe how Fe:C uptake ratios of diatoms and *Phaeocystis* are parametrized in the model and how these choices impact the simulated biomass distributions in ROMS-BEC. For more details on other aspects of the cycling of iron in BEC (e.g. scavenging, external sources), we kindly refer the reviewer to our answer to SC59 below.

In BEC, all of the dissolved iron (Fe) pool is assumed to be bioavailable to phytoplankton, whose Fe:C uptake stoichiometry for a given carbon uptake during photosynthesis varies from 3-20 μmol mol$^{-1}$ for all PFTs except diazotrophs (12-60 μmol mol$^{-1}$). In BEC, the Fe:C uptake ratio depends on surrounding Fe concentrations and the half-saturation constants of iron of the respective PFT (k$_{Fe}$, see Fig. 2 below for diatoms and *Phaeocystis*).  Generally, when seawater Fe concentrations fall below a critical level (specific to the PFT's half-saturation constant), the PFT reduces its cellular Fe:C requirements. As a result, diatoms generally have higher intracellular Fe:C ratios than *Phaeocystis* colonies in ROMS-BEC, and Fe:C ratios of all phytoplankton are highest in winter and lowest in summer, in concert with the seasonal evolution of upper ocean Fe levels (see Fig. 3 below). Photoacclimation affects the photosynthetic carbon uptake of phytoplankton (see model equations in

the appendix of the submitted manuscript) and thereby further modifies the intracellular Fe:C ratio of each PFT (not shown here).

As the Fe uptake per unit of C of phytoplankton is regulated by each PFT's $k_{Fe}$ in ROMS-BEC, these differences between *Phaeocystis* and diatoms are eliminated in the simulation IRON, in which the kFe of *Phaeocystis* is set to the value of diatoms (see method section 2.2 of the manuscript). In general, any simulated changes in the biomass distributions in this experiment relative to the *Baseline* setup result from the interaction of two factors: First, the lower $k_{Fe}$ of *Phaeocystis* in the IRON experiment (see Table 2 of the manuscript) leads to a lower Fe limitation of photosynthetic growth and hence more C uptake of *Phaeocystis* in the model, facilitating a higher buildup of biomass. Second, the higher Fe uptake per unit of C of *Phaeocystis* in the IRON experiment leads to a quicker Fe depletion, which ultimately drives *Phaeocystis* back into severe Fe limitation, slowing down the accumulation of carbon biomass.

In ROMS-BEC, the annual mean surface *Phaeocystis* biomass between 60-90°S in the IRON experiment increases to 170% of the concentration simulated in the *Baseline* setup, while the simulated average diatom biomass concentration declines to 65% of its *Baseline* value (see Fig. A2 in manuscript; note that the values in Fig. A2 refer to each PFT's chlorophyll biomass, but that the relative change in carbon biomass are comparable in magnitude). As expected, in the experiments kFe150/kFe50, the simulated changes are larger (25%/234% and 144%/34% for *Phaeocystis* and diatoms, respectively, see Fig. 1 above), resulting directly from the larger perturbation of kFe compared to the experiment IRON. Overall, this suggests that changes in the iron uptake, through changes in both iron limitation in photosynthesis and the Fe:C uptake ratio, significantly alter the relative abundance of *Phaeocystis* and diatoms in the high-latitude Southern Ocean. Yet, as a result of the at least partially compensating changes of *Phaeocystis* and diatom biomass for the simulated total phytoplankton biomass, the simulated changes in POC export between 60-90°S are rather small compared to the changes in community composition in ROMS-BEC upon changes in iron uptake dynamics (103.8%/94.8%/106.5% of *Baseline* in the experiments IRON/kFe150/kFe50).
While this suggests a rather small sensitivity of bulk properties like POC export or total phytoplankton biomass to the phytoplankton Fe:C uptake ratios, the large simulated changes in the phytoplankton community structure imply a large sensitivity of the exact pathways of carbon from its uptake to its export to the chosen Fe uptake parameters in ROMS-BEC. Further, we acknowledge that the current parametrization of Fe:C uptake ratios in BEC neglects more direct impacts of light availability (see e.g. Strzepek et al., 2019), the effect of which on the competition of *Phaeocystis* and diatoms and carbon export fluxes we can currently not assess.

[Figure]

**Fig. 2**: The Fe:C uptake ratio of diatoms (red) and *Phaeocystis* (blue) in ROMS-BEC as a function of Fe levels.

In the revised version of the manuscript, we have added information on the Fe:C uptake ratios of phytoplankton to the method section. It reads:

"In ROMS-BEC, phytoplankton Fe uptake relative to the uptake of C varies as a function of seawater Fe levels and decreases linearly below a critical concentration which is specific to each PFT's $k_{Fe}$ (see Eq. B11). In concert with the seasonal evolution of upper ocean Fe levels, the Fe:C ratios of all PFTs are highest in winter and lowest in summer (not shown). As a result of their higher $k_{Fe}$ in the model, *Phaeocystis* generally have lower Fe:C uptake ratios than diatoms."

Furthermore, we have added the model equations to the appendix in the revised manuscript:

In BEC, the Fe:C ratio $\theta_{Fe:C}^{i}$ [$\mu mol\ mol^{-3}$] of growth by phytoplankton $i$ varies between the maximum Fe:C ratio $\theta_{Fe:C,max}^{i}$ at high seawater Fe concentrations and the minimum Fe:C ratio $\theta_{Fe:C,min}^{i}$ at very low Fe concentrations. Below a critical surrounding Fe concentration, which depends on each PFT's half-saturation constant of iron $k_{Fe}^{i}$ (see Table 1), the ratio is reduced from the maximum Fe:C ratio following:

$$\theta_{Fe:C}^{i} = \theta_{Fe:C,max}^{i} \tag{B11}$$

$$\theta_{Fe:C}^{i} = \max\left(\theta_{Fe:C}^{i} \cdot \frac{[Fe]}{9 \cdot k_{Fe}^{i}}, \theta_{Fe:C,min}^{i}\right) \qquad \text{where } [Fe] < 9 \cdot k_{Fe}^{i} \tag{B12}$$

For this study, $\theta_{Fe:C,max}^{i}$ is 60 for diazotrophs and 20 for all other PFTs, and $\theta_{Fe:C,min}^{i}$ is 12 for diazotrophs and 3 for all other PFTs.

[Figure]

**Fig. 3**: Top: Monthly averaged surface iron concentrations in the Ross Sea. Bottom: The intracellular Fe:C ratio of diatoms (red) and *Phaeocystis* (blue). Both panels show output from the *Baseline* simulation of ROMS-BEC.

**Answers to specific comments (SC):**

**SC1:** Line 5: "improved" instead of "extended"
We have changed the sentence as follows:

"We improved ROMS-BEC by adding an explicit parameterization of *Phaeocystis* colonies, so that the model, together with the previous addition of an explicit coccolithophore type, now includes all biogeochemically relevant Southern Ocean phytoplankton types."

**SC2:** Line 7: This implies solitary Phaeocystis are not biogeochemically relevant, when commonly used to understand succession in the Ross sea
We kindly refer the reviewer to our answer to SC15 & SC18.

**SC3:** Line 7: Please report uncertainties
As described in more detail in our answer to GC1, we quantify the spatial variability of the contribution of *Phaeocystis* to integrated annual NPP and POC export as the area-weighted standard deviation of the reported averages in the revised manuscript. The abstract now reads:

"We find that *Phaeocystis* contribute 46±21% (1σ in space) and 40±20% to annual NPP and POC export south of 60° S, respectively, making them an important contributor to high- latitude carbon cycling."

**SC4:** Line 9: Saying "temporal variability" here implies spatial variability is not a considerable factor, when in the following line you say that there is a difference at the coast
In order to clarify, we have rephrased this part of the abstract as follows:

"In our simulation, the relative importance of *Phaeocystis* and diatoms is mainly controlled by spatio-temporal variability in temperature and iron availability. In addition, in more coastal areas, such as the Ross Sea, the higher light sensitivity of *Phaeocystis* at low irradiances promotes the succession from *Phaeocystis* to diatoms. "

**SC5:** Line 11: Remove "Still," from this sentence, it's unclear how this sentence follows from the previous
Changed as suggested in the revised version of the manuscript

**SC6:** Line 24-25: There is no previous statement that there is observed succession, just that the other groups contribute to biomass. Please include such a statement and provide references.
In the revised version of the manuscript, to link spatial and temporal variability of phytoplankton community structure and its importance for biogeochemical cycles, we have changed the respective sentence to also state *when* coccolithophores and *Phaeocystis* contribute substantially to biomass. It now reads:

"However, calcifying coccolithophores and dimethylsulfide (DMS) producing *Phaeocystis* have been found to contribute in a significant way to total phytoplankton biomass in summer/fall at subantarctic (Balch et al., 2016; Nissen et al., 2018) and in spring/summer at high latitudes, respectively (Smith and Gordon, 1997; Arrigo et al., 1999; DiTullio et al., 2000; Poulton et al., 2007; Arrigo et al., 2017), thus suggesting that the succession and competition of different plankton groups governs biogeochemical cycles at the (sub)regional scale."

**SC7:** Line 32: Since there are few and not zero estimates, please briefly reference them here.
In the revised manuscript, we have added references that quantify the contributions of different members of the high-latitude phytoplankton community to Southern Ocean NPP and POC export:

"While a number of recent studies have elucidated the importance of coccolithophores for subantarctic carbon cycling (e.g. Rosengard et al., 2015; Balch et al., 2016; Nissen et al., 2018; Rigual Hernández

et al., 2020), few estimates quantify the role of present and future high-latitude SO phytoplankton community structure for ecosystem services such as NPP and carbon export (e.g. Wang and Moore, 2011; Yager et al., 2016)."

**SC8:** Line 48: This is unclear. Do you mean that grazing would not be as significant of a loss as aggregation?
Yes, this is what we mean. Since *Phaeocystis* colonies are more likely to form aggregates than single cells and since they are additionally less likely to be grazed than single cells (both purely based on size assumptions), the observed dominance of colonies over solitary cells in summer (see e.g. Smith et al., 2003) likely leads to relatively more important aggregation than grazing for total *Phaeocystis* biomass loss. We have rephrased the respective sentence to clarify:

"Its alternation between solitary cells of a few µm in diameter and gelatinous colonies of several mm to cm in diameter (e.g. Rousseau et al., 1994; Peperzak, 2000; Chen et al., 2002; Bender et al., 2018) directly impacts community biomass partitioning and the relative importance of aggregation, viral lysis, and grazing for *Phaeocystis* biomass losses, its susceptibility to zooplankton grazing relative to that of diatoms (Granéli et al., 1993; Smith et al., 2003), and ultimately the export of particulate organic carbon (POC; Schoemann et al., 2005). With *Phaeocystis* colonies typically dominating over solitary cells during the SO growing season (Smith et al., 2003) and with larger cells being more likely to form aggregates and less likely to be grazed by microzooplankton (Granéli et al., 1993; Caron et al., 2000; Schoemann et al., 2005; Nejstgaard et al., 2007), *Phaeocystis* biomass loss via aggregation possibly increases in relative importance at the expense of grazing as more colonies are formed and colony size increases (Tang et al., 2008). "

**SC9:** Line 52: "expensive" instead of "difficult"
Changed as suggested in the revised version of the manuscript

**SC10:** Line 66: Reference Yager et al., 2016
We have added the reference as suggested. See also SC12.

**SC11:** Line 67: Zooplankton grazing rates on *Phaeocystis* are low (Yang et al., 2016)
We have added the reference as suggested. See also SC12.

**SC12:** Line 68: There are also arguments that the hydrography results in the resurfacing of any sunken Phaeocystis-associated POC (Lee et al., 2017)
We thank the reviewer for the additional references which we were not aware of. We have changed this sentence of the introduction, which now reads:

"While some have found blooms of *Phaeocystis* to be important vectors of carbon transfer to depth through the formation of aggregates (Asper and Smith, 1999; DiTullio et al., 2000; Ducklow et al., 2015; Yager et al., 2016; Asper and Smith, 2019), others suggest their biomass losses to be efficiently retained in the upper ocean by local circulation (Lee et al., 2017) and degraded in the upper water column through bacterial and zooplankton activity (Gowing et al., 2001; Accornero et al., 2003; Reigstad and Wassmann, 2007; Yang et al., 2016), making *Phaeocystis* a minor contributor to SO POC export.

**SC13:** Line 79: State how the referenced models parameterize Phaeocystis differently, and the possible consequences on the model outcome
We have added a statement addressing this comment in the introduction of the revised version of the manuscript:

"In this context, whether the model explicitly represents both *Phaeocystis* life stages (Pasquer et al., 2005; Kaufman et al., 2017; Losa et al., 2019) or only the colonial stage (Wang and 85 Moore, 2011; Le Quéré et al., 2016) is key, as single cells are known to have lower iron requirements than *Phaeocystis* colonies (Veldhuis et al., 1991)."

**SC14:** Line 88: The introduction is missing a description of observed succession in SO sectors outside of the Ross Sea

We thank the reviewer for this comment, but would like to clarify. In the introduction of the submitted manuscript, three of the six cited papers on Southern Ocean phytoplankton succession include a discussion from regions outside of the Ross Sea, namely from the Western Antarctic Peninsula (Arrigo et al., 2017) and from the circumpolar Southern Ocean (Green et al., 2006; Alvain et al., 2008), and all cited studies describe a succession from *Phaeocystis* to diatoms throughout spring and summer. Furthermore, as the Ross Sea is one of the key areas for the competition of *Phaeocystis* and diatoms in the Southern Ocean and was therefore chosen as one of the focus regions in our manuscript, the cited papers on phytoplankton succession in the introduction reflect past research efforts, which, as far as we are aware of, focused more heavily on the Ross Sea than on other sectors.

**SC15:** Line 93: Why did you not include solitary Phaeocystis when it's been used in other successful models (such as those published by Kaufman)?

We fully agree with the reviewer that better constraining the role of life cycle transitions of *Phaeocystis* for Southern Ocean carbon cycling is of high interest. Yet, for this study, we decided to focus on the colonial stage of *Phaeocystis* for two reasons. To the best of our knowledge, colonial *Phaeocystis* is dominant over solitary cells in terms of total *Phaeocystis* biomass during late spring and summer (e.g. Smith et al., 2003) and likely dominates for downward carbon fluxes (e.g. DiTullio et al., 2000; Yang et al., 2016), making the colonial stage the more relevant *Phaeocystis* life stage for our study. Furthermore, there is currently only 390 *Phaeocystis* biomass data points in the whole Southern Ocean, and the distinction between colonial and solitary Phaeocystis is often difficult (Vogt et al. 2012), impeding the basin-wide model evaluation of both *Phaeocystis* life stages, especially on a seasonal scale (see also section 4.3 of the manuscript). We note that single-celled *Phaeocystis* are implicitly included in the small phytoplankton group in ROMS-BEC (see method section 2.1). Thereby, while not being able to distinguish single-celled *Phaeocystis* from other nanophytoplankton cells in the model, the contribution of this *Phaeocystis* life stage to total phytoplankton biomass is contained in the estimate for the small phytoplankton group.

**SC16:** Line 95: Please give a brief statement here on how the model was validated.

In the revised manuscript, we have added a statement describing the model validation to this paragraph:

"Using available observations, such as satellite-derived chlorophyll concentrations, carbon biomass and pigment data, we first validate the simulated phytoplankton distributions and community structure across the SO and then particularly focus on the temporal variability of diatoms and *Phaeocystis* in the high-latitude SO."

As a response to a comment by referee #3, we have further modified the last paragraph to better reflect the structure of the manuscript. The whole paragraph now reads:

"In this study, we investigate the competition between *Phaeocystis* and diatoms and its implications for carbon cycling using a regional coupled physical-biogeochemical-ecological model configured at eddy-permitting resolution for the SO (ROMS-BEC, Nissen et al., 2018). To address the missing link between SO phytoplankton biogeography, ecosystem function, and the SO carbon cycle, we have added *Phaeocystis* colonies as an additional PFT to the model, so that it includes all major known biogeochemically relevant phytoplankton types of the SO (see e.g. Buesseler et al., 1998; DiTullio et al., 2000). Using available observations, such as satellite-derived chlorophyll concentrations, carbon biomass and pigment data, we first validate the simulated phytoplankton distributions and community structure across the SO and then particularly focus on the temporal variability of diatoms and *Phaeocystis* in the high-latitude SO. After assessing the relative importance of bottom-up and top-down factors in controlling the contribution of *Phaeocystis* colonies and diatoms to total phytoplankton biomass over a complete annual cycle in the high-latitude SO, we show that the spatially and temporarily varying phytoplankton community composition leaves a distinct, PFT-specific imprint on upper ocean carbon cycling and POC export across the SO ."

**SC17:** Line 102: does the addition of the new PFT affect the validation metrics done in Nissen et al. 2018?

In comparison to the 4-PFT setup of ROMS-BEC in Nissen et al. (2018), the model performance has improved in the 5-PFT setup of this study, as described in section 3.1 and seen in Fig. 2 & S7 of the manuscript. In summary, general trends like the too high chlorophyll biomass and NPP at high latitudes and the associated too low macronutrient concentrations remain also in the 5-PFT setup presented in this study, but the existing biases in these biogeochemical tracers are reduced upon the implementation of *Phaeocystis* into the model.

**SC18:** Line 107: isn't solitary Phaeocystis also important in the SO?

We agree with the reviewer that solitary cells can temporarily be more important for total *Phaeocystis* biomass than its colonial life form, but to the best of our knowledge, at the bloom peak, the colonial form typically dominates over solitary cells in the Southern Ocean (late spring/summer, e.g. Smith et al., 2003). In the revised version of the manuscript, we have rephrased the respective sentence to read:

"Here, we extend the version of Nissen et al. (2018) to include an explicit parameterization of colonial *Phaeocystis antarctica*, which is the only bloom-forming species of *Phaeocystis* occurring in the SO (Schoemann et al., 2005) and which typically dominates over solitary cells when SO *Phaeocystis* biomass levels are highest (Smith et al., 2003)."

Furthermore, as stated in our response to SC15 above, single-celled *Phaeocystis* are implicitly included in the small phytoplankton group in ROMS-BEC.

**SC19:** Line 110: Wouldn't you need to assume a minimum cell concentration for this to be valid?

Here, we assume enough cells to be available at any time in the small phytoplankton functional type of ROMS-BEC, in the same way it was previously done by Wang & Moore (2011). As model parameters for *Phaeocystis* growth are chosen to reflect the colonial life stage in ROMS-BEC and as biomass accumulation of this phytoplankton type thus only accelerates when environmental conditions are favorable for *Phaeocystis* colonies, we only expect a small sensitivity of the simulated *Phaeocystis* seasonality to the explicit inclusion of a minimum cell concentration for colony formation.

**SC20:** Line 114: Please give your rationale for using this function instead of Eppley

As indicated in the manuscript (section 2.1), the new phytoplankton functional type *Phaeocystis* represents a single species (*Phaeocystis Antarctica*) rather than a multitude of species as e.g. in the case of diatoms. As individual phytoplankton species typically show an optimum temperature for growth in laboratory experiments, above and below which its growth is slowed down (see Fig. 4 below), this justifies the choice to use an optimum curve to describe temperature-dependent growth of the *Phaeocystis* functional type in ROMS-BEC. In contrast, within the model PFT "diatoms", we do not model a specific species of diatoms, but the whole diatom community (typical PFT approach; Le Quéré et al., 2005), and the use of a so-called Q10-function (Eppley 1972) can hence be interpreted as the overlap of numerous optimum curves of numerous individual species.

[Figure]

**Fig. 4:** Growth rates as a function of temperature for example high-latitude SO species of diatoms and *Phaeocystis* (Boyd 2019).

**SC21:** Table 1: Why did you choose a slightly different grazing rate for diatoms and Phaeocystis? This choice is motivated based on size-mismatch assumptions between diatoms and *Phaeocystis* colonies and the single zooplankton grazer in ROMS-BEC (see section 2.1 of the manuscript).

**SC22:** Line 149: shouldn't alpha be sensitive to the iron concentration? (Strzepek et al., 2019) We thank the reviewer for pointing us to the paper by Strzepek et al. (2019). In ROMS-BEC, the sensitivity of phytoplankton to changes in light intensity at low light, i.e., alpha PI, is set as a constant for each phytoplankton functional type (see Table 1 of the manuscript). However, the light limitation formulation also accounts for effects of photoacclimation (see appendix of the manuscript; Geider et al., 1998), thereby allowing for interactions between light and nutrient availability (iron availability in the case of the high SO latitudes; see Fig. 5 below). We note that while the respective curves for diatoms and *Phaeocystis* look rather similar in Fig. 5 below (compare e.g. the two solid lines), differences in the model simulation are larger at any location and any given point in time due to differences in their temperature growth limitation function, which further modifies the light limitation factor (not considered in Fig. 5 below, but see Eq. B9 of the manuscript), their iron half-saturation constants and the resulting differences in their nutrient limitation factor. Still, we acknowledge that this parametrization should be re-assessed in the light of recent advances (such as the study by Strzepek et al.) in future work.

[Figure]

**Fig. 5**: Light limitation [n.d.] as a function of light intensity (PAR, W m$^{-2}$) in ROMS-BEC for growth by diatoms (red) and *Phaeocystis* (blue). A light limitation factor of 1 denotes no light limitation of phytoplankton growth. The dashed line represents a nutrient limited case, the dotted line a nutrient replete case. For the computation, the annual mean surface C:Chl ratio of diatoms (54.8 mg C (mg chl)$^{-1}$) and *Phaeocystis* (83.7 mg C (mg chl)$^{-1}$) between 60-90°S is used. Note that in the model, in addition to nutrient limitation, temperature stress further modifies this ratio, see Eq. B9 of the manuscript. Consequently, while the respective curves for diatoms and *Phaeocystis* look rather similar here (compare e.g. the two solid lines), differences in the model simulation are larger at any location and any given point in time due to differences in their temperature growth limitation function, their iron half-saturation constants, and the resulting differences in their nutrient limitation factor.

**SC23:** Line 162: this is a very small difference, what is the net sensitivity of the output to this parameter?
Admittedly, the difference in the maximum grazing rate between grazing on diatoms (3.8 d$^{-1}$) and grazing on *Phaeocystis* (3.6 d$^{-1}$) appears rather small. Yet, high-latitude biomass distributions in ROMS-BEC are rather sensitive to this difference. In fact, in the experiment GRAZING, in which we set the maximum grazing rate on *Phaeocystis* to the value of diatoms, i.e., we increase the grazing pressure on *Phaeocystis* (see section 2.2 and Table 2 of the manuscript), annual mean *Phaeocystis* biomass concentrations decrease to 52% and even 38% of the levels in the *Baseline* simulation between 60-90°S and in the Ross Sea, respectively (see Fig. A2 of the manuscript). At the same time, biomass levels of diatoms increase to 121% and 149% of the levels in the *Baseline* simulation,

demonstrating the rather large sensitivities to this parameter. We note, however, that by choosing the same grazing rate on diatoms and *Phaeocystis*, the resulting phytoplankton community structure at high SO latitudes shows a larger discrepancy to the observed community structure (compare e.g. Fig. A2 to Fig. 2 in the manuscript), suggesting that the choice of a higher maximum grazing rate on diatoms than on *Phaeocystis* in our model setup is justified as it results in a more ecologically sound ecosystem structure.

**SC24:** Line 190: what day of year are you initializing with?
All model simulations are started on January, 1 of the respective year. Please see also our answer to SC25 for a discussion of the sensitivity of the results to the chosen initial conditions of the phytoplankton community composition.

**SC25:** Line 194: what is the sensitivity of the model outcome to your initial community composition?
The simulated phytoplankton biomass concentrations assessed in this manuscript are averaged over the last five full annual cycles of a 20-year long simulation (see section 2.2 of the manuscript). In fact, the simulated phytoplankton biomass distributions for this analysis period are controlled more by chosen model parameters than by the chosen initial biomass distributions. Initial chlorophyll fields of each PFT are derived using satellite-derived total chlorophyll concentrations and a fixed partitioning onto the model PFTs. For our simulations, the chosen partitioning (90% small phytoplankton, 4% diatoms and coccolithophores, 1% diazotrophs and *Phaeocystis*, see method section 2.2) is admittedly motivated by the phytoplankton community composition at the open northern boundary at 24°S, i.e., in the middle of the subtropical gyre, where small phytoplankton (e.g. *Prochlorococcus* and *Synechococcus*) dominate. In contrast, at high SO latitudes, large phytoplankton types, such as diatoms and *Phaeocystis* dominate (e.g. Swan et al., 2016). Nevertheless, phytoplankton distributions at high latitudes are quickly in steady state, as biomass levels of all phytoplankton types decrease to very low levels in the high-latitude winter months (due to the absence of light), resulting in a phytoplankton community close to equilibrium already in the 2$^{nd}$ growth season. Therefore, by analyzing the years 15-20 of each model simulation, our results are independent of the chosen initial community composition.

**SC26:** Line 212: diatom and phaeo cellular Fe:C ratios should also be informed by light and iron limitation (Strzepek et al., 2011)
As described in more detail in our answer to GC1, the Fe:C uptake ratios of phytoplankton are a function of the surrounding Fe concentrations in the version of ROMS-EBC used for this study. In addition, photoacclimation affects the cellular Fe:C content of phytoplankton by impacting the light limitation factor (see Eq. B9 in the manuscript), which is used to calculate the photosynthetic carbon uptake in the model.

**SC27:** Line 240: Top 50 m is not deep enough for analyzing export.
We agree with the reviewer on this statement. This is why, for the analysis of export fluxes, the top 100m are assessed (see Fig. 6 and Table 3 of the manuscript), but we note that in our model, at high latitudes, ~80% of total biomass can be found in the top 50m. The reason for choosing the top 50m for the analysis of the spatial distribution of the PTFs in ROMS-BEC was thus twofold. First, we decided on this depth level for Fig. 1 and the niche analysis for an easy comparison with the plots from the 4-PFT setup of ROMS-BEC presented in Nissen et al. (2018). Second, most of the available phytoplankton carbon biomass validation data are from the top 50m of the water column.

**SC28:** Line 246: References to supplemental figures are not in order
We thank the reviewer for pointing this out. We have corrected the order in the revised manuscript.

**SC29:** Line 250: What does the outcome look like with an attempt at a quantitative comparison? This would at least be useful to see in the supplemental material.
The study by Brun et al. (2015) is based on all available observations in the MAREDAT database (e.g. for *Phaeocystis*, see Vogt et al., 2012), but environmental niche centers and widths for each phytoplankton functional type are presented for the whole year only, as data coverage is generally low and skewed towards the summer season, especially in the Southern Ocean. Therefore, we believe that

constructing monthly niche centers based on the few available observations would not be meaningful at the moment. Since monthly niche centers are therefore not provided by Brun et al. (2015), we decided to compare the provided niche centers to December-March averages in ROMS-BEC and to thereby focus the quantitative comparison on the seasonal scale (see section 3.3 of the manuscript).

**SC30:** Line 263: This doesn't seem like an insurmountable problem; satellite chlorophyll data should be employed for model validation. Binning and temporal averaging are potential workarounds for the issues presented.

Data coverage of satellite-chlorophyll in the focus area of this study, i.e., the Southern Ocean south of 60°S, is low, already at monthly temporal resolution, but especially at daily resolution (see Fig. S4 in the supplement). As correctly pointed out by the reviewer, this problem can be surmounted by aggregating data in time. For this reason, we use satellite-derived chlorophyll concentrations to validate the simulated summer average chlorophyll field of ROMS-BEC (Fig. 1 of the manuscript). Yet, the assessment of phytoplankton bloom metrics requires a higher temporal frequency in the chlorophyll time series than monthly (ideally daily). As satellite data coverage is highest in the summer months (December-March), especially south of 60°S (Fig. S4 in the supplement), which coincides with peak phytoplankton biomass levels, we have decided to focus the discussion of bloom phenology on the bloom peak in this manuscript, instead of focusing on bloom initiation (typically in spring, see also our answer to SC32). Furthermore, regarding the seasonality of phytoplankton functional types, the cited literature discussing diatoms and *Phaeocystis* phenology often refers to the bloom peak of these two (see e.g. Peloquin & Smith 2007, Smith et al., 2011), thereby facilitating a direct comparison with our model output.

**SC31:** Line 263: if you are using timing from the literature, why not use bloom initiation from the literature as well?

Please see our answer to SC30.

**SC32:** Line 265: I recommend you also validate using bloom initiation as well.

Defining the bloom start as the day at which total chlorophyll levels first surpass 105% of the annual median chlorophyll concentrations (see Nissen et al., 2018), the total chlorophyll bloom south of 60°S starts in late September in ROMS-BEC (week 11, calendar starts in July), at least 2 months earlier than suggested by satellite-derived chlorophyll data (see e.g. Thomalla et al., 2011). This is consistent with the difference in the timing of the bloom peak discussed in the manuscript (section 3.2). In the revised manuscript, we have added a statement along these lines in section 3.2:

"Maximum total chlorophyll concentrations are simulated for the first half of December across latitudes in ROMS-BEC (solid blue line in Fig. 3a), and at high SO latitudes south of 60° S, total chlorophyll blooms start already in late September in the model (not shown). Thereby, the model-derived timing of total chlorophyll bloom start and peak is 2-3 and 1-2 months earlier, respectively, than satellite-derived estimates (for bloom peak, see black line in Fig. 3a, for bloom start, see e.g. Thomalla et al., 2011)."

**SC33:** Line 314: Could the overestimate have to do with modeled Fe:Chl ratios?

The recent study by Strzepek et al. (2019) suggests that Southern Ocean phytoplankton have higher photosynthetic rates at low iron, light, and temperature than temperate phytoplankton, suggesting that the formulations describing photoacclimation used here are possibly not applicable globally (see Geider et al., 1998) and should be reassessed in ROMS-BEC in future work. Yet, the overestimation is also seen in the carbon biomass distributions (see Fig. 1 & S5 in the manuscript), suggesting that factors describing the carbon uptake and/or loss are dominantly driving the high biomass bias. Please also see our answer to GC34.

**SC34:** Line 315: Please state the reason why Chl was overestimated in Nissen et al., 2018

Biases in simulated chlorophyll levels in ocean biogeochemistry models can be caused by a bias either in physics (temperature, mixed layer depth, thus impacting light availability) or in biology (growth or loss rates). Generally, in ROMS-BEC, temperature is biased high and the mixed layer is biased shallow, i.e., light availability is biased high, both favoring the accumulation in phytoplankton

biomass (Nissen et al., 2018). In Nissen et al. (2018), we have tested the effect of the biases in the physical fields by correcting the temperature and light field only in the BEC-subroutine (i.e., not affecting ocean dynamics). In conclusion, the simulated biases in temperature and light availability are not large enough to explain the simulated bias in chlorophyll concentration (~80-90% remained unexplained, see Nissen et al., 2018).

Hence, biological factors must be the reason for the bias. While biases in the chosen maximum growth rates of phytoplankton functional groups certainly contribute to the simulated chlorophyll bias at high SO latitudes (see e.g. Fig. A1 of the manuscript for the positive bias in the temperature-limited growth rate of diatoms at low temperatures compared to available laboratory data), correcting this unavoidably leads to a negative chlorophyll bias at subantarctic latitudes in the model, demonstrating the difficulty of simulating the whole diatom community with a single model functional type (see also discussion in Losa et al., 2019). The study by Le Quéré et al. (2016) demonstrates how adding additional complexity in the zooplankton compartment can reduce high-latitude chlorophyll biases by directly affecting total grazing rates on phytoplankton and the coupling of phytoplankton and zooplankton in the model.

Since ROMS-BEC currently only includes a single zooplankton functional type and given that we have tested the impact of all other potential factors causing the simulated bias in total chlorophyll levels at high SO latitudes in ROMS-BEC, this is the most likely reason for the positive chlorophyll bias at high latitudes, which yet needs to be tested in our model.

We kindly refer the reviewer to L. 320-323 in the manuscript, where we write:
"[…] the bias is likely due to a combination of underestimated high-latitude chlorophyll concentrations in satellite-derived products (Johnson et al., 2013) and the missing complexity in the zooplankton compartment in ROMS-BEC, as biases in the simulated physical fields (temperature, light) have been shown to only explain a minor fraction of the simulated high-latitude biomass overestimation (Nissen et al., 2018)."

**SC35:** Figure 1: The model is not capturing spatial variability in chlorophyll concentration- if it's truly due to a latitudinal bias in the ocean color product, please validate against data from shipboard CTDS
As explained in more detail in our answer to SC34 above, missing complexity in the zooplankton compartment is currently our leading hypothesis for causing the positive chlorophyll bias at high SO latitudes in ROMS-BEC. While Johnsen et al. (2013) show that at chlorophyll concentrations >2mg chl m$^{-3}$, their satellite-derived chlorophyll fields are typically 20-30% lower than in-situ chlorophyll concentrations (based on HPLC data), this underestimation is, however, not enough to fully account for the positive bias simulated by ROMS-BEC. Please see also our answer to SC 36 below.

**SC36:** Line 320: How does modeled Chl compare to measurements from CTDs, gliders, BGC Argo floats, etc. from the region? Then you can determine if it's a satellite underestimate issue.
Fluorescence-based chlorophyll observations (ship-based, floats etc.) reveal concentrations of up to 10 mg chl m$^{-3}$ in the high-latitude SO (see Fig. 6 below), in agreement with the total chlorophyll distribution simulated by ROMS-BEC (Fig. 1 of the manuscript). However, while acknowledging the data scarcity in the in situ data (see Fig. 6 below), the very high chlorophyll concentrations (>2 mg chl m$^{-3}$) appear to be too wide-spread in ROMS-BEC when comparing to Fig. 6 below, which also shows areas with concentrations <2 mg chl m$^{-3}$ at latitudes >60°S.

[Figure]

[Figure]

**Fig. 6:** Median total chlorophyll a concentration from fluorescence-based chlorophyll observations (ship-based, floats etc.; mg chl m$^{-3}$) scaled to a 3° spatial resolution for a) 0-25m and b) 25-50m (Sauzède et al., 2015). Data are from all months, but data availability is skewed to the summer months (see Sauzède et al., 2015).

**SC37:** Line 324: "distinct" instead of "distinctly different"
Changed as suggested.

**SC38:** Table 3: Please include confidence intervals on ROMS-BEC values
Please see our answer to GC1 for details. We revised Table 3 as follows:

**Table 3.** Comparison of ROMS-BEC based phytoplankton biomass, production, and export estimates with available observations (given in parentheses). Data sources are given below the Table. The reported uncertainty of the contribution of the PFTs to the simulated integrated NPP corresponds to the area-weighted spatial variability of each PFT's contribution to annual NPP (1$\sigma$ in space).

| | | ROMS-BEC (Data) | |
| --- | --- | --- | --- |
| | | 30-90° S | 60-90° S |
| Surface chlorophyll biomass | total, annual mean [Gg chl] | 40.8 (34.5[a]) | 17.1 (9.5[a]) |
| Diatom carbon biomass | 0-200m, annual mean [Pg C] | 0.059 (global[b]: 0.10-0.94) | 0.015 |
| *Phaeocystis* carbon biomass | 0-200m, annual mean [Pg C] | 0.019 (global[b]: 0.11-0.71) | 0.010 |
| Coccolithophore carbon biomass | 0-200m, annual mean [Pg C] | 0.012 (global[b]: 0.001-0.03) | 0.001 |
| NPP | Pg C yr$^{-1}$ | 17.2 (12.1-12.5[c]) | 3.0 (0.68-1.7[c]) |
| | Diatoms [%] | 52.0 (±26.2) | 49.1 (±19.9) |
| | *Phaeocystis* [%] | 15.3 (±24.5) | 45.8 (±20.7) |
| | Coccolithophores [%] | 14.6 (±15.3) | 0.7 (±1.0) |
| | SP [%] | 17.2 (±16.1) | 4.5 (±1.9) |
| POC export at 100m | Pg C yr$^{-1}$ | 3.1 (2.3-2.96[d]) | 0.62 (0.21-0.24[d]) |

[a] Monthly climatology from MODIS Aqua (2002-2016, NASA-OBPG, 2014a), SO algorithm (Johnson et al., 2013)
[b] The reported estimates from the MAREDAT data base in Buitenhuis et al. (2013) are global estimates of phytoplankton biomass.
[c] Monthly climatology from MODIS Aqua VGPM (2002-2016, Behrenfeld and Falkowski, 1997; O'Malley, last access: 16 May 2016), NPP climatology from Buitenhuis et al. (2013, 2002-2016)
[d] Monthly output from a biogeochemical inverse model (Schlitzer, 2004) and a data-assimilated model (DeVries and Weber, 2017).

**SC39:** Table 3: Why do you use a 100 m depth horizon for export? The 0.1% light depth horizon is more biogeochemically relevant (Buesseler et al., 2020)
We thank the reviewer for this comment. Unsurprisingly, the simulated POC export between 30-90°S and 60-90°S are lower when using the depth at which PAR corresponds to 0.1% of the incoming PAR at the surface (1.9 and 0.4 Pg C yr$^{-1}$, respectively, based on monthly averages) than when using a fixed depth of 100 m (3.1 and 0.62 PgC yr$^{-1}$), as the former is often at depths greater than 100 m in the focus area of this study (at around 110-120 m, see Buesseler et al., 2020). Yet, we note that our estimates of the contribution of diatoms and *Phaeocystis* to POC export is largely unaffected by this difference, as virtually all POC in ROMS-BEC is produced above 100 m (not shown) and as the model currently only includes one class of POC, meaning that the remineralization of POC with depth is identical for POC originating from diatoms and *Phaeocystis* and thereby conserving the contribution of different functional types to POC production with depth. Further, we decided to report the export fluxes of POC at 100 m because this is still the norm in the field of biogeochemical modeling (see e.g. Bopp et al., 2013, Laufkötter et al., 2016)

**SC40:** Line 338: It would increase model confidence to include whether coccolithophore biomass corresponds to positioning of the great calcite belt.
We thank the reviewer for this comment. Indeed, the distribution of coccolithophore biomass in ROMS-BEC agrees well with the location of the "Great Calcite Belt" at subantarctic latitudes (Balch et al., 2011; Nissen et al., 2018). We have added this information in the revised version of the manuscript:

"In contrast to both *Phaeocystis* and diatoms, the simulated biomass levels of coccolithophores are highest in the subantarctic (highest concentrations of 3 mmol C m$^{-3}$ on the Patagonian Shelf, Fig. 1e & S3d). Overall, their simulated SO biogeography agrees well with the position of the "Great Calcite Belt" (Balch et al., 2011, 2016) and remains largely unchanged compared to the 4-PFT setup (Nissen et al., 2018)."

**SC41:** Line 342: Please include standard deviations on these percentages.
Following our answer to GC1, we have included the spatial variability of each PFT's relative contribution to the seasonally averaged total chlorophyll levels in the revised manuscript, expressed as one standard deviation within the respective subarea. The sentence now reads:

"Averaged over 30-90°S (60-90°S), the simulated relative contributions of *Phaeocystis*, diatoms, and coccolithophores to total chlorophyll in summer are 20±28% (33±34%; subarea mean as shown in Fig. 2b & c ±1σ in space), 68±33% (64±33%), and 5±17% (<1±2%), respectively, in good agreement with the CHEMTAX climatology (28% (27%), 46% (48%), and 3% (1%), respectively)."

**SC42:** Figure 2: Please use box-and-whisker plots instead of pie charts to display categorical data and the associated error.
In the revised manuscript, we have reported the standard deviation as an estimate of the spatial variability of the contribution of the PFTs to seasonally averaged chlorophyll (see SC41 above & SC43 below). Given that the focus of the manuscript is on the simulated average community composition in key areas, we have decided to keep the Figure as is.

**SC43:** Line 347: Again, please include error on your reported percentages.
We have adapted the respective sentence to now include a quantification of the spatial variability (see also GC1 and S41 above):

"The model overestimates the contribution of *Phaeocystis* in fall (39±14% as compared to 24% in CHEMTAX) and spring (51±22% as compared to 28%) between 60-90° S and in the Ross Sea, respectively (Fig. 2c-d), but the limited number of data points available in the CHEMTAX climatology in this area and the uncertainty in the attribution of pigments in CHEMTAX to the *Phaeocystis* PFT in ROMS-BEC have to be noted (see section 2.3.1)."

**SC44:** Line 361: "agrees" instead of "suggests"
Changed as suggested in the revised version of the manuscript

**SC45:** Line 370: Please also include a comparison to bloom initiation.
Please see our comment to SC32 above.

**SC46:** Line 374: This doesn't seem right. Phaeocystis are often associated with bloom peaks in coastal polynyas.
We thank the reviewer for this valuable comment and have corrected the text to reflect this imprecision. In fact, analyzing the model output in more detail revealed that our statement only holds for the broad spatial averages presented in Figure 2 & 3. In fact, when looking at the contribution of *Phaeocystis* to total phytoplankton chlorophyll biomass at the day of the total chlorophyll bloom peak in ROMS-BEC, we see that besides in parts of the open ocean, *Phaeocystis* dominate many coastal regions at bloom peak, e.g. in the Ross Sea and in the Amundsen polynya (see Fig. 7 below), in agreement with observations (see e.g. Yager et al., 2016). The respective sentence in the revised manuscript now reads:

"As diatoms dominate the phytoplankton community at peak total chlorophyll concentrations for all latitudinal averages in the model domain (compare their bloom timing in Fig. 3c to Fig. 3a and to the simulated community composition in Fig. 2b-d, but note that *Phaeocystis* often dominate in coastal areas, not shown), the mismatch in timing is likely related to the representation of this PFT in the model, and is possibly at least partly caused by their comparatively high growth rates at low temperatures (see Fig. A1a)."

[Figure]

**Fig. 7:** Relative contribution of Phaeocystis to total chlorophyll at the day of the annually maximum chlorophyll concentration in the *Baseline* setup of ROMS-BEC.

**SC47:** Figure 3: Why don't Phaeocystis concentrations reach values much higher than 3 ug/L? Much higher concentrations have been observed.

Figure 3b of the manuscript shows the simulated zonally averaged daily surface chlorophyll concentrations of *Phaeocystis*. When plotting the annual maximum of daily averaged *Phaeocystis* chlorophyll concentrations at each grid cell in the *Baseline* setup of ROMS-BEC, the simulated concentrations exceed 10 mg chl m$^{-3}$ locally (especially in coastal areas, see Figure 8 below), thus significantly higher than the zonal average suggests and in agreement with observed *Phaeocystis* chlorophyll concentrations. We have added this information to section 3.2 of the revised manuscript:

"In contrast to diatoms, maximum zonally averaged chlorophyll concentrations of *Phaeocystis* are simulated for late November or early December across most latitudes in the model (only around 70°S a peak in late January is simulated, Fig. 3b; note that locally, maximum *Phaeocystis* chlorophyll concentrations exceed 10 mg chl m$^{-3}$, not shown here)."

[Figure]

**Fig. 8:** Annual maximum of daily averaged *Phaeocystis* chlorophyll concentration [mg chl m$^{-3}$] in the *Baseline* setup of ROMS-BEC.

**SC48:** Line 382: How specifically does the diatom parameterization drive this bias?

In general, total chlorophyll concentrations peak too early in ROMS-BEC, especially at high latitudes. As the phytoplankton community is mainly composed of diatoms and *Phaeocystis* in this area, biases in the chlorophyll seasonality of either of the two could contribute to this bias. Since the chlorophyll seasonality of *Phaeocystis* is in broad agreement with the reported seasonality of DMS concentrations in the high SO latitudes, we have concluded that diatoms mainly control this bias in total chlorophyll concentrations. Any too quick accumulation of diatom biomass (and hence chlorophyll) early in the

season is caused by an imbalance of their growth and loss rates at those times. In particular, if their growth rates were biased high in the model (e.g. due to a too high maximum growth rate in this area, see Fig. A1 of the manuscript), whereas e.g. zooplankton grazing were lagging behind and were biased low, this would result in a decoupling of diatoms and zooplankton, allowing for the build-up of high diatom biomass levels early in the growth season.

We have adapted the manuscript as follows:
"This further corroborates the hypothesis that the bias in the timing of maximum total chlorophyll levels in ROMS-BEC is likely caused by how diatoms are parameterized in the model (see e.g. the rather high temperature-limited growth rate of diatoms at low temperatures compared to available laboratory data, see Fig. A1). "

**SC49:** Line 405: Wouldn't the lower cell density associated with deeper MLD bias this assessment? It is known that Phaeocystis thrive under lower light conditions than diatoms, so this doesn't seem right.
We agree with the reviewer, that based on available laboratory and in situ studies, it is known that *Phaeocystis* colonies cope better with low light environments than diatoms (see introduction and method section of the manuscript). However, we are not sure we understand how this relates to the topic discussed in this part of the manuscript. Here, assuming that *Phaeocystis surface* chlorophyll concentrations are most representative for air-sea fluxes of DMSP & DMS, we compare the timing of the simulated maximum *surface* chlorophyll concentrations of *Phaeocystis* in ROMS-BEC to the reported timing of peak DMSP & DMS concentrations in the atmosphere. Since we find a good agreement between the two, we conclude that biases in *Phaeocystis* phenology are possibly small and that the bias in total phytoplankton chlorophyll phenology must be driven by biases in the seasonal evolution of diatom chlorophyll.

**SC50:** Line 408: It is important to note here that much of the SO is light limited, in particular the canyons near the WAP (Carvalho et al., 2016).
We fully agree with the reviewer and apologize for any confusion. In L. 408 of the manuscript, we refer to iron as being the most limiting amongst all *nutrients* for phytoplankton growth in ROMS-BEC (Fig. S1). In the revised version of the manuscript, we have changed the respective sentence to make this clearer, and it now reads:

"With regard to iron, the two PFTs do not occupy distinct ecological niches in ROMS-BEC (niche centers at 0.32 μmol m$^{-3}$ for both PFTs, see Fig. S9). Yet, *as all simulated phytoplankton growth is most limited by iron availability in the high-latitude SO compared to the availability of other nutrients* (Fig. S1), this suggests that the spatio-temporal averaging applied for the niche analysis here potentially precludes the assessment of the role of iron in the competition between *Phaeocystis* and diatoms, especially on a sub-seasonal scale."

**SC51:** Line 414: If this analysis is not useful for the scientific questions proposed, it should be either removed or moved to the supplemental information.
We thank the reviewer for this comment and apologize for the confusion. The analysis of environmental niches of phytoplankton in ROMS-BEC and the comparison of these niches with those reported for individual phytoplankton taxa. Here, in agreement with observations, our analysis reveals a separation of diatoms and *Phaeocystis* in NO3 and temperature space. However, maybe surprisingly, this analysis does not reveal any difference in the niche regarding MLPAR and iron, likely due to model biases (MLPAR) and the temporal averaging (iron; due to observational data scarcity, see also SC29 above). As this analysis serves as a link between the evaluation of the simulated distributions of diatoms and *Phaeocystis* and the analysis of the factors controlling these distributions, we have decided to keep it in the main text.

**SC52:** Line 422: If this is the case, why do Phaeocystis never dominate at the bloom peak in the model?
We thank the reviewer for this comment. In fact, the analysis of the bottom-up factors only partly informs about the realized biomass distributions, as these are a result of bottom-up and top-down factors at any given time and location (see Fig. 5 and Lines 446ff of the manuscript). Thus, if it was

only bottom-up factors controlling which phytoplankton type was the dominant one, *Phaeocystis* should outcompete diatoms over much of the Southern Ocean in ROMS-BEC. Therefore, in our model, bottom-up factors only *promote the accumulation* of *Phaeocystis* biomass relative to that of diatoms, whereas top-down factors achieve the opposite. As a result, *Phaeocystis* only successfully outcompete diatoms in terms of their contribution to total biomass at certain locations and at certain times of the year (see Fig. 5i & j, but see also Fig. 7 above).

**SC53:** Figure 4: The temperature range covered here is very large- where are water column temperatures getting so hot in the model? This may be killing off the *Phaeocystis*.
Figure 4 in the manuscript is shown for the area south of 40°S. As shown in the Figure 9 below, top 50m and DJFM average temperatures close to 40°S reach 20°C, in agreement with observed temperatures (Locarnini et al., 2013). We note, however, that the temperature between 60-90°S, where *Phaeocystis Antarctica* is a key player of the phytoplankton community, ranges from ~-2-8°C in the model (see red bars next to the temperature axes in Fig. 4 of the manuscript), resulting in zero growth of *Phaeocystis* north of ~60°S in ROMS-BEC (see temperature function in Fig. A1 of the manuscript), in agreement with laboratory studies (Buma et al., 1991) and observations (Schoemann et al., 2005).

[Figure]

**Fig. 9:** Top 50 m and DJFM average temperature [°C] in ROMS-BEC (30-90°S).

**SC54:** Line 430: It would be useful to see how photoacclimation effects on the Fe:C ratio would affect the outcome.
Please see our answer to SC26 and GC1. As stated there, photoacclimation only impacts the Fe:C ratio of phytoplankton through its impact on the light limitation factor and hence photosynthetic carbon uptake. Thereby, the effects of photoacclimation are included in the assessment of the differences in the light limitation factor in Fig. 5 of the manuscript.

**SC55:** Figure 5: Iron is green here, not blue.
In fact, it is shown in blue.

**SC56:** Line 442: It would also be good to see Pine Island Polynya and the Amundsen Sea Polynya to compare with the Ross Sea.
Overall, the seasonal evolution of the contribution of each PFT to total phytoplankton biomass concentrations in the Amundsen Sea (averaged over the whole coastal area south of 71°S and between 240-260°W) is very similar to those in the Ross Sea, with *Phaeocystis* generally being more important for total phytoplankton biomass in the Amundsen Sea than in the Ross Sea (compare Fig. 2 in the manuscript to Fig. 10a below). Furthermore, the factors controlling the biomass distributions in the Amundsen Sea are overall similar to those in the Ross Sea, with advantages in light limitation of *Phaeocystis* relative to diatoms being larger than in the Ross Sea (compare Fig. 5 in the manuscript to Fig. 10b below), explaining the higher relative importance of *Phaeocystis* in the Amundsen Sea.

[Figure]

**Fig. 10:** Same as Fig. 2 & 5 in the manuscript, but averaged over the Amundsen Sea.

**SC57:** Line 445: In the model they should be considered, but what is your confidence in the model when it cannot reproduce a Chl max in Phaeocystis?
As shown in Fig. 8 above, the model simulates maximum daily chlorophyll concentrations of >10 mg chl m⁻ locally for *Phaeocystis*, in agreement with observations, which are not visible due to the broad spatial averaging in Fig. 3 of the manuscript. We are therefore confident in the simulated interplay of bottom-up and top-down factors in controlling the high-latitude competition between diatoms and *Phaeocystis* in our model. We also refer the reviewer to our answer to SC47 & SC52.

**SC58:** Line 457: It is necessary to know how sensitive the model is to the range of these parameters to determine how likely this result is to mirror reality.
As shown in Fig. A2 of the submitted manuscript, the simulated high-latitude chlorophyll fields of *Phaeocystis* and diatoms are highly sensitive to the chosen model parameters for *Phaeocystis* biomass loss rates (experiments GRAZING, AGGREGATION, and MORTALITY). The magnitude of change in the chlorophyll fields is largest when neglecting parameter differences between *Phaeocystis* and diatoms for aggregation (followed in decreasing order by mortality and grazing), which is in agreement with the analysis shown in Fig. 5 and discussed in section 3.3 of the manuscript.

Yet, the additionally performed parameter sensitivity experiments in this round of revisions revealed a different picture regarding the ranking of the experiments, with the largest simulated change in *Phaeocystis* and diatom chlorophyll concentrations when varying the maximum zooplankton grazing rate on *Phaeocystis* by +/-50%, followed by the response when varying the mortality and aggregation rates (see GC1 and supplementary material of the revised manuscript). However, we note that the relative change in the model parameters was different for the experiments shown in Fig. A2, explaining the differences in the simulated response and making the two sets of experiments not directly comparable.

Given that the model parameters in the *Baseline* simulation of ROMS-BEC are chosen to best reflect the observed high-latitude phytoplankton community structure throughout the year, we are more confident in the qualitative dynamics simulated by the model than in the quantitative results, i.e. the relative importance of bottom-up and top-down factors, due to the large parameter uncertainty demonstrated in our sensitivity experiments (and with uncertainty resulting from interactions of multiple parameters not having been assessed here) and the scarcity of the observational data used to constrain the model.

**SC59:** Line 467: It would be great to include some necessary information in this manuscript about the way iron is cycled in ROMS-BEC. How are organic ligands parameterized? How about scavenging processes? Is the relief from iron limitation in the Ross Sea driven by wind-driven sediment resuspension (McGillicuddy et al., 2015)?

Besides uptake by phytoplankton (please see also our answer to GC1 for details on the model parametrizations of the Fe:C uptake ratios of phytoplankton in ROMS-BEC), particle scavenging is the second loss pathway of dissolved iron (Fe). Scavenging rates of Fe are a function of surrounding Fe concentrations (to crudely account for the effect of iron-binding ligands, which are not explicitly included in BEC) and available sinking particles (particulate organic carbon, as well as the ballast materials calcite, opal, and dust). Overall, the higher the concentrations of Fe and particles, the higher is the loss of Fe through scavenging. For more details, the reviewer is kindly referred to Moore & Braucher 2008 and Lima et al. (2014), which thoroughly describe and discuss the cycling of iron and treatment of particles in BEC.

In the version of ROMS-BEC used here, sediments (and hence wind/circulation-driven sediment resuspension) are not explicitly modeled, and all particles are immediately buried, i.e., lost from the system, or remineralized when reaching the ocean floor. In the model, fluxes of iron from the sediments are supplied to the bottom model layer and parametrized as a function of the particle flux to the sediment and bottom water oxygen concentrations following Dale et al. (2015). As a consequence, even though sediment resuspension is not explicitly modeled, wind-driven mixing of the water column is a key process in supplying the upper water column with dissolved iron released from the sediments. This is especially true at high SO latitudes, where atmospheric deposition is low (Mahowald et al., 2009). Here, any seasonal change in the supply of iron to the mixed layer is the result of entrainment of higher iron concentrations from below, which result from the remineralization of sinking particulate organic matter.

**SC60:** Line 475: Please include error bars on these estimates. Propagate the error using the sensitivity analyses, and the standard deviation across the domain.

Following our answer to GC1, we have included the spatial variability of each PFT's relative contribution to the NPP and POC export in the revised manuscript, expressed as one standard deviation within the respective subarea. The sentence now reads:

"*Phaeocystis* is an important member of the SO phytoplankton community in our model, particularly south of 60°S, where it contributes 46±21% and 40±20% to total annual NPP and POC formation, respectively (Table 3 & Fig. 6). Even when considering the entire region south of 30°S, the contribution of *Phaeocystis* to NPP (15±24%) and POC production (16±22%) is sizeable."

**SC61:** Figure 6: Please report uncertainty on the numbers in this figure

A thorough assessment of the uncertainty of all fluxes presented in Fig. 6 to describe the routing of carbon through the phytoplankton and zooplankton compartments in ROMS-BEC would be computationally expensive and is beyond the scope of this study. Please see also our answer to GC1.

**SC62:** Line 492: It would be helpful to see a breakdown of these fluxes in comparison to other modeled and observed export in the SO.

We appreciate this comment. While there is a number of observational studies assessing individual pathways of carbon through the system (see e.g. reviews for diatoms and *Phaeocystis* by Sarthou et al., 2005 and Reigstad et al., 2007, respectively), there is, to the best of our knowledge, no observation-based study available which holistically quantifies the relative importance of aggregation, grazing, and non-grazing mortality (e.g. viral lysis) for the biomass losses of different PFTs in the Southern Ocean. Yet, assumptions made in ROMS-BEC when choosing model parameters are motivated by available in situ or laboratory based studies which focused on individual aspects of the routing of carbon through the system (see method section of the manuscript). With regards to biogeochemical models, Laufkötter et al. (2016) present a detailed comparison of the routing of carbon through the high-latitude ecosystem in state-of-the-art ocean biogeochemistry models (see Figure 11 below). In summary, the importance of different pathways, namely grazing, aggregation, or mortality, for POC production at high latitudes varies widely, with some models suggesting diatom aggregation to be the most important pathway (REcoM and BEC), whereas others point to the big importance of zooplankton

mortality. Overall, acknowledging that all of these models only include two phytoplankton functional types, making a direct quantitative comparison with the results in ROMS-BEC difficult, this points to the uncertainty surrounding the representation of particle treatment in state-of-the-art ocean biogeochemistry models and highlights the need for more observational/laboratory studies to guide modelers in improving these parametrizations. In this context, on-going projects like EXPORTS (https://oceanexports.org/about.html) are an important contribution in the field, shedding light on how carbon is routed through the upper ocean.

[Figure]

**Fig. 11**: Routing of carbon through the ecosystem in the high latitudes south/north of 60°S/N in REcoM, BEC (CESM version), TOPAZ, and PISCES. Adapted from Laufkötter et al. (2016). The boxes denote the relative contribution of each functional type to total biomass (green) and particle production (yellow), respectively. The arrows show, from left to right, diatom aggregation, diatom grazing by zooplankton, zooplankton mortality, nanophytoplankton grazing by zooplankton, and nanophytoplankton aggregation.

**SC63:** Figure 7: Since there is a wide potential range of aggregation rates, how would this figure change when testing using that range?
As expected, changing the quadratic loss rate in aggregation in ROMS-BEC leads to a shift in the relative importance of grazing and aggregation for total phytoplankton biomass loss. Based on size assumptions, generally higher aggregation rates are assumed for *Phaeocystis* colonies than for diatoms in ROMS-BEC (see method section of the submitted manuscript). In fact, the difference in the quadratic loss rate parameter between *Phaeocystis* and diatoms (see Table 1 of the manuscript) is one of the reasons why the model simulates the differences in aggregation vs grazing for these two phytoplankton types (Fig. 7 of the manuscript). However, we note that in ROMS-BEC, the simulated aggregation rate for a given phytoplankton PFT at any point in time is additionally a function of the quadratic biomass at the given location (see model equations in the appendix of the manuscript), meaning that ROMS-BEC simulates substantial spatio-temporal variability in phytoplankton biomass aggregation rates as a direct result of the simulated spatio-temporal variability in carbon biomass concentrations.

**SC64:** Line 497: Why is the peak in Phaeocystis so late? This is much later than observations.
We thank the reviewer for this comment. We point out that in section 3.4 of the manuscript, we only describe the carbon cycling for very broad spatial averages (30-90°S and 60-90°S). Admittedly, this makes it difficult to compare the simulated timing of maximum POC production of *Phaeocystis* to available observations taken at a more regional scale. In fact, the timing of peak *Phaeocystis* POC production and its peak contribution to total POC production is earlier in the Ross Sea (~November/December, see Fig. 12 below) than between 60-90°S, thus reconciling our model results with the observations. As a response to a comment by reviewer #3, the carbon cycling figures for the Ross Sea (Fig. 12 below) were added to the supplementary material of the revised version of the manuscript.

[Figure]

**Fig. 12**: Carbon cycling in the Ross Sea: The upper panel corresponds to Fig. 6 in the main text, the lower panels shows the quantities from Fig. 7 for the Ross Sea.

**SC65:** Line 525: I find this result hard to believe when the model is not taking into account changing cellular iron quotas as iron limitation shifts.
The Fe:C uptake ratio of the different phytoplankton PFTs in ROMS-BEC varies as a function of the Fe concentrations (and thereby also the cellular iron quotas). Please see our answer to GC1 for more details.

**SC66:** Line 539: More than Kfe needs to be constrained with environmental data. Alpha, Fe:C, Pmax are all also sensitive to light/iron limitation conditions.
Agreed. We have rephrased the sentence as follows:

"Thus, in order to include light-iron inter- actions in future modeling efforts with *Phaeocystis* and to assess their impact on the competition of *Phaeocystis* with diatoms throughout the SO, additional measurements are needed for how $k_{Fe}$, but also e.g. $\alpha_{PI}$ and the Fe:C uptake ratio of phytoplankton vary as a function of the surrounding light level."

**SC67:** Line 546: There is agreement in some conditions, but elsewhere there seems to be large discrepancies. Additionally, surface chlorophyll variability is not represented well by the model.
While the simulated distribution of chlorophyll concentrations in ROMS-BEC is admittedly far from in perfect agreement with observation, we point the reviewer to our answers of e.g. SC 46 & SC 47, demonstrating that important features of the variability in *Phaeocystis* chlorophyll concentrations are simulated by the model. Accordingly, we have rephrased the sentence as follows:

„Furthermore, the simulated spatio-temporal variability of the high-latitude phytoplankton community structure is in agreement with that suggested by available pigment data (Fig. 2)."

**SC68:** Line 555: It would be useful to conduct more sensitivity experiments while varying the aggregation parameter alongside other model parameters, in light of this finding.
Please see our comment to GC1 above.

**SC69:** Line 565: This paragraph seems somewhat redundant with the above paragraph. Please restructure.
We have combined the two paragraphs into a single, slightly modified paragraph in the revised manuscript. It reads:

Loss processes, such as aggregation and grazing, clearly matter for the competitive advantage of one PFT over another, but these loss processes are generally not well quantified and often not studied with sufficient detail. For example, while the modeling study by Le Quéré et al. (2016) demonstrates the importance of such top-down control for total SO phytoplankton

575 biomass concentrations, an analysis of the impact on phytoplankton community structure is yet to be done. In fact, in the literature, only few studies discuss the role of top-down factors for the relative importance of *Phaeocystis* and diatoms in the high-latitude SO (Granéli et al., 1993; van Hilst and Smith, 2002). Consequently, very little quantitative information exists to constrain model parameters (see section 2.1) or to validate the simulated non-grazing mortality, grazing, or aggregation loss rates of *Phaeocystis* and diatoms over time. In agreement with our results, aggregation has been suggested to be an important

580 process facilitating high POC export when *Phaeocystis* biomass is high (Asper and Smith, 1999; Ducklow et al., 2015; Asper and Smith, 2019), but to what extent this process significantly contributes to the observed relative importance of *Phaeocystis* and diatoms throughout the year in the high-latitude SO remains largely unknown. Certainly, the simulated aggregation rates in the model and their impact on spatio-temporal distributions of PFT biomass concentrations and rates of NPP are associated with substantial uncertainty due to the immediate conversion of biomass to sinking detritus in the model, the equal treatment

585 of POC originating from all PFTs, the neglect of disaggregation, and due to the calculation of aggregation rates based on the biomass concentrations of individual PFTs rather than all PFTs or even particles combined (see e.g. Turner, 2015). Given that the simulated biomass distributions in ROMS-BEC are most sensitive to differences in parameters describing non-grazing mortality (e.g. viral lysis) and aggregation (Fig. A2 & S11), any changes in these loss processes will significantly impact the relative abundance of *Phaeocystis* and diatoms in the SO. Additionally, as discussed in Nissen et al. (2018), the lack of multiple

590 zooplankton groups in the SO model (Le Quéré et al., 2016) and the parametrization of the single zooplankton grazer using fixed prey preferences and separate grazing on each prey using a Holling Type II function (Holling, 1959), which thus precludes a saturation of feeding at high total phytoplankton biomass, are major limitations of ROMS-BEC. To what extent accounting implicitly for grazing by higher trophic levels in the non-grazing mortality term makes up for not including more zooplankton PFTs remains unclear. Nevertheless, by changing the overall coupling between phytoplankton and zooplankton and through

595 the distinct grazing preferences of the different zooplankton types, the addition of larger zooplankton grazers would likely change the simulated temporal evolution of *Phaeocystis* and diatom biomass in the model (Le Quéré et al., 2016). Therefore, the above mentioned uncertainties should be addressed by future in situ or laboratory measurements in order to better constrain the simulated biomass loss processes, as our findings suggest these to be necessary to explain the seasonal evolution of the relative importance of *Phaeocystis* and diatoms in the high-latitude SO.

**SC70:** Line 579: Please provide uncertainty on this percentage.
In agreement with our answer to GC1, we have added the spatial variability of the contribution of *Phaeocystis* to SO NPP in the revised manuscript. For the estimation of the contribution of SO *Phaeocystis* to global NPP we have used the simulated integrated NPP of *Phaeocystis* in ROMS-BEC, whose uncertainty is not straightforward to quantify and is beyond the scope of this study (see GC1). The respective part of the manuscript now reads:

"Based on our model results, *Phaeocystis* is a substantial contributor to global NPP and POC export. Comparing the integrated NPP and POC export between 30-90° S in ROMS-BEC with data-based estimates of global NPP and POC export suggests that SO *Phaeocystis* alone contribute about 5% to globally integrated NPP ($58\pm7$ Pg C yr$^{-1}$, Buitenhuis et al., 2013), and about the same percentage to global POC export ($9.1\pm0.2$ Pg C yr$^{-1}$, DeVries and Weber, 2017). Thereby, our simulated contribution of *Phaeocystis* to global NPP is higher than that found in the previous modeling study by Wang and Moore (2011), particularly at higher latitudes, where Wang and Moore (2011) diagnosed a contribution of 23% to NPP south of 60°S ($46\pm21\%$ in ROMS-BEC)."

**SC71:** Line 587: Please propagate this uncertainty into your NPP, POC, and export estimates.
Please see our answer to GC1 above.

**SC72:** Line 590: Considering the discrepancies between the observations and the model, please include confidence intervals on your estimates.
Please see our answer to GC1 above.

**SC73:** Line 608: Why can't you assess these effects? Assessing horizontal fluxes of model tracers should be relatively straightforward.

We completely agree with the reviewer in that the assessment of the simulated physical fluxes of the biogeochemical tracers would allow for a quantification of the spatial/temporal decoupling of POC production and export. However, particle sinking of e.g. POC is currently treated implicitly in ROMS-BEC, meaning that particles are instantaneously distributed in the vertical and remineralized following an exponential curve (Lima et al., 2014). Therefore, sinking particles are not laterally advected with physical fluxes in the version of ROMS-BEC used here, making it impossible to assess the decoupling between POC production and export. In the revised version of the manuscript, we have rephrased this part of the manuscript to better explain the implications of the implicit particle treatment. It how reads:

"In this regard, the simulated strong temporal coupling between POC fluxes and biomass distributions in ROMS- BEC is a direct result of the model formulations describing particle sinking (particles sink implicitly, i.e., they are not laterally advected, Lima et al., 2014)."

The implementation of an explicit treatment of sinking particles, which are then subject to lateral fluxes with ocean circulation, is on-going work in the research group at ETH Zürich, but goes beyond our study.

**SC74:** Line 616: How are you accounting for variability in mixed layer depth in the DMSP calculation?
In the calculation presented in the manuscript, we are not accounting for any variability in mixed layer depth at the moment, but instead use the approximated DMS production integrated over the top 10 m (calculated using a range of conversion efficiencies to get from the DMSP produced by *Phaeocystis* to DMS). Assuming that all of the DMS produced in this uppermost layer will quickly exchange with the atmosphere, we acknowledge that this is a very rough *upper* estimate. When integrating over greater depths, e.g. the mixed layer, our assumption will be less justified, as DMS produced at greater depths will at least partly be degraded before escaping to the atmosphere (see e.g. Stefels et al., 2007). However, as we are currently not explicitly modeling the cycling of DMS in ROMS-BEC, we have decided to only present the calculation for the uppermost ocean layer, where the assumption of quick exchange is most justified.

**SC75:** Line 640: These topics should be discussed in your methods. The rationale for excluding solitary Phaeocystis needs to be justified.
Please see our answers to comments SC15 & SC18 above.

**SC76:** Line 669: These non-Redfieldian dynamics should be straightforward to implement in the model, and would be an interesting sensitivity study.
We completely agree with the reviewer in that the assessment of non-Redfieldian stoichiometry of diatoms and *Phaeocystis* for the simulated Southern Ocean-wide nutrient distributions and export fluxes should be a key focus of the modeling community. However, unfortunately, we do not agree with the reviewer in the amount of work its implementation would involve. While the implementation of non-Redfieldian stoichiometry in phytoplankton nutrient uptake might seem rather easy (and is currently already done e.g. for the Fe:C uptake ratio, see GC1 above), it requires the implementation of several additional biogeochemical tracers to track the non-Redfieldian nutrient uptake stoichiometry not only through phytoplankton biomass, but also through all particle classes and zooplankton. As a result, this exercise certainly goes beyond a simple sensitivity experiment and unfortunately beyond the scope of this work.

**SC77:** Line 689: It may be worth testing a range of parameters to get a general sense of the sensitivity of export. I'd imagine this would be useful for many people to see
We agree with the reviewer that a thorough assessment of the sensitivity of the simulated carbon routing to the chosen model parameters would be interesting, but note that this analysis would be computationally expensive and is beyond the scope of this study. For this study, parameters were chosen to best fit the observed pattern and magnitude of export fluxes.

**Cited literature**

Alvain, S., Moulin, C., Dandonneau, Y., & Loisel, H. (2008). Seasonal distribution and succession of dominant phytoplankton groups in the global ocean: A satellite view. *Global Biogeochemical Cycles*, *22*(3), GB3001. https://doi.org/10.1029/2007GB003154

Arrigo, K. R., van Dijken, G. L., Alderkamp, A., Erickson, Z. K., Lewis, K. M., Lowry, K. E., Joy-Warren, H. L., Middag, R., Nash-Arrigo, J. E., Selz, V., & van de Poll, W. (2017). Early Spring Phytoplankton Dynamics in the Western Antarctic Peninsula. *Journal of Geophysical Research: Oceans*, *122*(12), 9350–9369. https://doi.org/10.1002/2017JC013281

Balch, W. M., Drapeau, D. T., Bowler, B. C., Lyczskowski, E., Booth, E. S., & Alley, D. (2011). The contribution of coccolithophores to the optical and inorganic carbon budgets during the Southern Ocean Gas Exchange Experiment: New evidence in support of the "Great Calcite Belt" hypothesis. *Journal of Geophysical Research*, *116*, C00F06. https://doi.org/10.1029/2011JC006941

Buesseler, K. O., Boyd, P. W., Black, E. E., & Siegel, D. A. (2020). Metrics that matter for assessing the ocean biological carbon pump. *Proceedings of the National Academy of Sciences*, *117*(18), 9679–9687. https://doi.org/10.1073/pnas.1918114117

Bopp, L., Resplandy, L., Orr, J. C., Doney, S. C., Dunne, J. P., Gehlen, M., Halloran, P., Heinze, C., Ilyina, T., Séférian, R., Tjiputra, J., & Vichi, M. (2013). Multiple stressors of ocean ecosystems in the 21st century: projections with CMIP5 models. *Biogeosciences*, *10*(10), 6225–6245. https://doi.org/10.5194/bg-10-6225-2013

Boyd, P. W. (2019). Physiology and iron modulate diverse responses of diatoms to a warming Southern Ocean (supplement). *Nature Climate Change*, *9*(2), 148–152. https://doi.org/10.1038/s41558-018-0389-1

Buma, A. G. J., Bano, N., Veldhuis, M. J. W., & Kraay, G. W. (1991). Comparison of the pigmentation of two strains of the prymnesiophyte Phaeocystis sp. *Netherlands Journal of Sea Research*, *27*(2), 173–182. https://doi.org/10.1016/0077-7579(91)90010-X

Carvalho, F., Kohut, J., Oliver, M. J., Sherrell, R. M., & Schofield, O. (2016). Mixing and phytoplankton dynamics in a submarine canyon in the West Antarctic Peninsula. *Journal of Geophysical Research: Oceans*, *121*(7), 5069–5083. https://doi.org/10.1002/2016JC011650

Dale, A. W., Nickelsen, L., Scholz, F., Hensen, C., Oschlies, A., & Wallmann, K. (2015). A revised global estimate of dissolved iron fluxes from marine sediments. *Global Biogeochemical Cycles*, *29*(5), 691–707. https://doi.org/10.1002/2014GB005017

DiTullio, G. R., Grebmeier, J. M., Arrigo, K. R., Lizotte, M. P., Robinson, D. H., Leventer, A., … Dunbar, R. B. (2000). Rapid and early export of Phaeocystis antarctica blooms in the Ross Sea, Antarctica. *Nature*, *404*(6778), 595–598. https://doi.org/10.1038/35007061

Eppley, R. W. (1972). Temperature and phytoplankton growth in the sea. *Fishery Bulletin*, *70*(4).

Geider, R. J., MacIntyre, H. L., & Kana, T. M. (1998). A dynamic regulatory model of phytoplanktonic acclimation to light, nutrients, and temperature. *Limnology and Oceanography*, *43*(4), 679–694. https://doi.org/10.4319/lo.1998.43.4.0679

Green, S. E., & Sambrotto, R. N. (2006). Plankton community structure and export of C, N, P and Si in the Antarctic Circumpolar Current. *Deep Sea Research Part II: Topical Studies in Oceanography*, *53*(5–7), 620–643. https://doi.org/10.1016/j.dsr2.2006.01.022

Johnson, R., Strutton, P. G., Wright, S. W., McMinn, A., & Meiners, K. M. (2013). Three improved satellite chlorophyll algorithms for the Southern Ocean. *Journal of Geophysical Research: Oceans*, *118*(7), 3694–3703. https://doi.org/10.1002/jgrc.20270

Laufkötter, C., Vogt, M., Gruber, N., Aumont, O., Bopp, L., Doney, S. C., … Völker, C. (2016). Projected decreases in future marine export production: the role of the carbon flux through the upper ocean ecosystem. *Biogeosciences*, *13*(13), 4023–4047. https://doi.org/10.5194/bg-13-4023-2016

Lee, S., Hwang, J., Ducklow, H. W., Hahm, D., Lee, S. H., Kim, D., … Shin, H. C. (2017). Evidence of minimal carbon sequestration in the productive Amundsen Sea polynya. *Geophysical Research Letters*, *44*(15), 7892–7899. https://doi.org/10.1002/2017GL074646

Le Quéré, C., Harrison, S. P., Colin Prentice, I., Buitenhuis, E. T., Aumont, O., Bopp, L., … Wolf-Gladrow, D. (2005). Ecosystem dynamics based on plankton functional types for global ocean biogeochemistry models. *Global Change Biology*, *11*, 2016–2040. https://doi.org/10.1111/j.1365-2486.2005.1004.x

Lima, I. D., Lam, P. J., & Doney, S. C. (2014). Dynamics of particulate organic carbon flux in a global ocean model. *Biogeosciences*, *11*(4), 1177–1198. https://doi.org/10.5194/bg-11-1177-2014

Locarnini, R. A., Mishonov, A. V., Antonov, J. I., Boyer, T. P., Garcia, H. E., Baranova, O. K., … Seidov, D. (2013). *World Ocean Atlas 2013, Volume 1: Temperature* (Vol. 1). Retrieved from http://www.nodc.noaa.gov/OC5/indprod.html

Losa, S. N., Dutkiewicz, S., Losch, M., Oelker, J., Soppa, M. A., Trimborn, S., Xi, H., and Bracher, A.: On modeling the Southern Ocean Phytoplankton Functional Types, Biogeosciences Discussions, https://doi.org/10.5194/bg-2019-289, 2019

Mahowald, N. M., Engelstaedter, S., Luo, C., Sealy, A., Artaxo, P., Benitez-Nelson, C., Bonnet, S., Chen, Y., Chuang, P. Y., Cohen, D. D., Dulac, F., Herut, B., Johansen, A. M., Kubilay, N., Losno, R., Maenhaut, W., Paytan, A., Prospero, J. M., Shank, L. M., & Siefert, R. L. (2009). Atmospheric Iron Deposition: Global Distribution, Variability, and Human Perturbations. *Annual Review of Marine Science*, *1*(1), 245–278. https://doi.org/10.1146/annurev.marine.010908.163727

McGillicuddy, D. J., Sedwick, P. N., Dinniman, M. S., Arrigo, K. R., Bibby, T. S., Greenan, B. J. W., … Dijken, G. L. (2015). Iron supply and demand in an Antarctic shelf ecosystem. *Geophysical Research Letters*, *42*(19), 8088–8097. https://doi.org/10.1002/2015GL065727

Moore, J. K., & Braucher, O. (2008). Sedimentary and mineral dust sources of dissolved iron to the world ocean. *Biogeosciences*, *5*(3), 631–656. https://doi.org/10.5194/bg-5-631-2008

Nissen, C., Vogt, M., Münnich, M., Gruber, N., & Haumann, F. A. (2018). Factors controlling coccolithophore biogeography in the Southern Ocean. *Biogeosciences*, *15*(22), 6997–7024. https://doi.org/10.5194/bg-15-6997-2018

Reigstad, M., & Wassmann, P. (2007). Does Phaeocystis spp. contribute significantly to vertical export of organic carbon? In *Phaeocystis, major link in the biogeochemical cycling of climate-relevant elements* (pp. 217–234). Springer Netherlands. https://doi.org/10.1007/978-1-4020-6214-8_16

Sarthou, G., Timmermans, K. R., Blain, S., & Tréguer, P. (2005). Growth physiology and fate of diatoms in the ocean: a review. *Journal of Sea Research*, *53*(1–2), 25–42. https://doi.org/10.1016/j.seares.2004.01.007

Sauzède, R., Lavigne, H., Claustre, H., Uitz, J., Schmechtig, C., D'Ortenzio, F., Guinet, C., & Pesant, S. (2015). Vertical distribution of chlorophyll a concentration and phytoplankton community composition from in situ fluorescence profiles: a first database for the global ocean. *Earth System Science Data*, *7*(2), 261–273. https://doi.org/10.5194/essd-7-261-2015

Schoemann, V., Becquevort, S., Stefels, J., Rousseau, V., & Lancelot, C. (2005). Phaeocystis blooms in the global ocean and their controlling mechanisms: a review. *Journal of Sea Research*, *53*(1–2), 43–66. https://doi.org/10.1016/j.seares.2004.01.008

Smith, W. O., Dennett, M. R., Mathot, S., & Caron, D. A. (2003). The temporal dynamics of the flagellated and colonial stages of Phaeocystis antarctica in the Ross Sea. *Deep Sea Research Part II: Topical Studies in Oceanography*, *50*(3–4), 605–617. https://doi.org/10.1016/S0967-0645(02)00586-6

Stefels, J., Steinke, M., Turner, S., Malin, G., & Belviso, S. (2007). Environmental constraints on the production and removal of the climatically active gas dimethylsulphide (DMS) and implications for ecosystem modelling. In *Phaeocystis, major link in the biogeochemical cycling of climate-relevant elements* (pp. 245–275). Dordrecht: Springer Netherlands. https://doi.org/10.1007/978-1-4020-6214-8_18

Strzepek, R. F., Boyd, P. W., & Sunda, W. G. (2019). Photosynthetic adaptation to low iron, light, and temperature in Southern Ocean phytoplankton. *Proceedings of the National Academy of Sciences*, *116*(10), 4388–4393. https://doi.org/10.1073/pnas.1810886116

Strzepek, R. F., Maldonado, M. T., Hunter, K. A., Frew, R. D., & Boyd, P. W. (2011). Adaptive strategies by Southern Ocean phytoplankton to lessen iron limitation: Uptake of organically complexed iron and reduced cellular iron requirements. *Limnology and Oceanography*, *56*(6), 1983–2002. https://doi.org/10.4319/lo.2011.56.6.1983

Swan, C. M., Vogt, M., Gruber, N., & Laufkoetter, C. (2016). A global seasonal surface ocean climatology of phytoplankton types based on CHEMTAX analysis of HPLC pigments. *Deep Sea Research Part I: Oceanographic Research Papers*, *109*, 137–156. https://doi.org/10.1016/j.dsr.2015.12.002

Thomalla, S. J., Fauchereau, N., Swart, S., & Monteiro, P. M. S. (2011). Regional scale characteristics of the seasonal cycle of chlorophyll in the Southern Ocean. *Biogeosciences*, *8*(10), 2849–2866. https://doi.org/10.5194/bg-8-2849-2011

Vogt, M., O'Brien, C., Peloquin, J., Schoemann, V., Breton, E., Estrada, M., … Peperzak, L. (2012). Global marine plankton functional type biomass distributions: *Phaeocystis* spp. *Earth System Science Data*, *4*(1), 107–120. https://doi.org/10.5194/essd-4-107-2012

Wang, S., & Moore, J. K. (2011). Incorporating Phaeocystis into a Southern Ocean ecosystem model. *Journal of Geophysical Research*, *116*(C1), C01019. https://doi.org/10.1029/2009JC005817

Yager, P., Sherrell, R., Stammerjohn, S., Ducklow, H., Schofield, O., Ingall, E., … van Dijken, G. (2016). 
[revised manuscript text omitted]

$$\gamma_{\mathrm{g,rel}}^{\mathrm{DPA}} = \log \frac{\frac{\gamma_{\mathrm{g}}^{\mathrm{PA}}}{P^{\mathrm{PA}}}}{\frac{\gamma_{\mathrm{g}}^{\mathrm{D}}}{P^{\mathrm{D}}}} \tag{3}$$

In ROMS-BEC, grazing on each phytoplankton $i$ is calculated using a Holling Type II ingestion function (Nissen et al., 2018). As described in section 2.1, *Phaeocystis* and diatoms in ROMS-BEC do not only differ in parameters describing the zooplankton grazing pressure they experience, but in parameters describing their non-grazing mortality and aggregation losses as well.

320 Therefore, in accordance with the relative grazing ratio defined above, we define the relative mortality ratio ($\gamma_{\mathrm{m,rel}}^{\mathrm{ij}}$) and the relative aggregation ratio ($\gamma_{\mathrm{a,rel}}^{\mathrm{ij}}$) of phytoplankton $i$ and $j$ (e.g. diatoms and *Phaeocystis*) as the ratio of their specific non-grazing mortality rates ($\gamma_{\mathrm{m}}^{\mathrm{i}}$, $\mathrm{d}^{-1}$) and aggregation rates ($\gamma_{\mathrm{a}}^{\mathrm{i}}$, $\mathrm{d}^{-1}$), respectively, following:

$$\gamma_{\mathrm{m,rel}}^{\mathrm{DPA}} = \log \frac{\frac{\gamma_{\mathrm{m}}^{\mathrm{PA}}}{P^{\mathrm{PA}}}}{\frac{\gamma_{\mathrm{m}}^{\mathrm{D}}}{P^{\mathrm{D}}}} \tag{4}$$

325 $$\gamma_{\mathrm{a,rel}}^{\mathrm{DPA}} = \log \frac{\frac{\gamma_{\mathrm{a}}^{\mathrm{PA}}}{P^{\mathrm{PA}}}}{\frac{\gamma_{\mathrm{a}}^{\mathrm{D}}}{P^{\mathrm{
[revised manuscript text omitted]

1200

**Supplementary material**

The supporting information provides additional figures in section S1 with respect to the nutrient limitation of phytoplankton growth in ROMS-BEC (S1), the ecological niche analysis (S2-S3), the data coverage in a SO satellite derived chlorophyll product (S4), the model evaluation (S5-S8), the bloom timing (S9), the competition sensitivity simulations (S10), carbon cycling in the Ross Sea (S11), and the results when using a varying half-saturation constant of iron for *Phaeocystis* growth (S12). In section S2, results of the parameter sensitivity simulations are described (Table S1-S3, Fig. S13).

**S1: Additional figures**

[Figure]

**Figure S1:** Annual mean most limiting nutrient at the surface south of 45° S for growth rates of a) *Phaeocystis* and b) diatoms in the *Baseline* simulation of ROMS-BEC. High-latitude phytoplankton growth in the model is most limited by either iron (green) or silicic acid (yellow, diatoms only). The stippling in panel a) denotes areas where peak monthly mean chlorophyll concentrations of Phaeocystis do not exceed 0.1 mg chl m$^{-3}$.

[Figure]

**Figure S2:** Simulated DJFM average top 50 m average coccolithophore carbon biomass concentrations (mmol C m$^{-3}$) south of 40° S as a function of the simulated temperature (° C) and a) nitrate concentrations (mmol N m$^{-3}$) and b) mixed layer PAR levels (W m$^{-2}$). Overlain are the observed ecological niche centers (median) and breadths (inter quartile ranges) for example taxa from Brun et al. (2015, circle) and as simulated in ROMS-BEC (triangles and dashed lines; area and biomass weighted). The red bars on the axes indicate the simulated range of the respective environmental condition in ROMS-BEC between 60-90° S and averaged over DJFM and the top 50 m.

[Figure]

**Figure S3:** Simulated DJFM average top 50 m average a) *Phaeocystis*, b) diatom, and c) coccolithophore carbon biomass concentrations (mmol C m$^{-3}$) south of 40° S as a function of the simulated a)-c) dissolved iron concentrations ($\mu$mol Fe m$^{-3}$) and mixed layer PAR levels (W m$^{-2}$) and d)-f) temperature (° C) and dissolved silicic acid concentrations [mmol Si m$^{-3}$] in the 5-PFT *Baseline* simulation of ROMS-BEC. Overlain are the simulated area and biomass weighted ecological niche centers (median, triangle) and breadths (inter quartile ranges, dashed lines) for the three functional types.

[Figure]

**Figure S2S4:** Assessment of the SO data coverage in the climatological (1998-2018, i.e. 21 years) daily Globcolor chlorophyll product (Fanton d'Andon et al., 2009; Maritorena et al., 2010): a)-f) Average number of years available for the calculation of the climatological chlorophyll concentration at each grid cell for each of the shown months (October-March), respectively. No minimum number of "days with data coverage" is required for a given month to be counted as "data available" (i.e. one day of data coverage in a month is enough for that month to be counted as "covered" in the respective year). g) Average number of years available for the calculation of the climatological chlorophyll concentration on each day for 10° latitudinal bands across the SO.

[Figure]

**Figure S3S5:** Validation of a) & b) *Phaeocystis*, c) diatom, and d) coccolithophore carbon biomass [mmol C m⁻³]. Panel a) shows the maximum *Phaeocystis* carbon biomass concentrations [mmol C m⁻³] in ROMS-BEC (circles) and in observations (squares, Vogt et al., 2012) for each month between November-February and in the the upper 50 meters of the water column. For panels b)-d), the model output is colocated with observations in space and time, and observational data from all months and from above 1000 m are considered here (Balch et al., 2016; Saavedra-Pellitero et al., 2014; O'Brien et al., 2013; Vogt et al., 2012; Leblanc et al., 2012; Tyrrell and Charalampopoulou, 2009; Gravalosa et al., 2008; Cubillos et al., 2007). For more details on the biomass evaluation, see Nissen et al. (2018). The dotted line shows the perfect linear 1:1 fit, whereas the solid line is the actual fit of the data (linear regression). Pearson correlation coefficients of these regressions are given in the top right, those for *Phaeocystis* and coccolithophores are statistically significant (p<0.05). Points are color-coded according to the sampling latitude.

[Figure]

**Figure S4S6:** a)-c) Relative contribution of the five phytoplankton PFTs to total chlorophyll biomass [mg chl m$^{-3}$] for a) 30-90° S, b) 60-90° S, and c) the Ross Sea. The top pie charts denote the climatological mixed layer average community composition suggested by CHEMTAX analysis of HPLC pigments for spring, summer, and fall, respectively (the total number of available observations for a given region and season is given at the lower left side, Swan et al., 2016), and the lower pie charts denote the corresponding community structure in the top 50 m in ROMS-BEC in the 5-PFT setup (middle row, same as in Fig. 2 in the main text) and in the 4-PFT setup (lowest row, no *Phaeocystis*, Nissen et al., 2018), respectively. Note that the categories in the CHEMTAX analysis are not 100% equivalent to the model PFTs, and here, "Hapto-8 reassigned" corresponds to the contribution of Hapto-6 where the temperature is <2°C (see also section 2.3.1 in the main text).

[Figure]

**Figure S7:** Annual mean bias (*Baseline* simulation minus observations) of a) total surface chlorophyll concentrations [g chl m$^{-3}$], b) total vertically integrated NPP [mg C m$^{-2}$ d$^{-1}$], d) surface nitrate concentrations [mmol m$^{-3}$], and e) surface silicic acid concentrations [mmol m$^{-3}$]. The panels c) & f) denote the temporal evolution of the model bias of c) total surface chlorophyll concentration (red) and total NPP (blue), as well as f) surface nitrate concentrations (red), and silicic acid concentrations (blue) in the 5-PFT setup of ROMS-BEC between 30-60° S (solid) and 60-90° S (dashed), respectively. For comparison, the model bias obtained with the 4 PFT setup of ROMS-BEC is included in both panels in green (chlorophyll and nitrate) and yellow (NPP and silicic acid), respectively (see also supplement in Nissen et al., 2018).

[Figure]

**Figure S6S8:** Annual mean top 100 m average a) Si* [mmol m$^{-3}$], which is defined as the difference in concentration between silicic acid and nitrate (Freeman et al., 2018), in the *Baseline* simulation of the 5-PFT setup of ROMS-BEC (colors). The contours denote the latitude of the silicate front, i.e. where Si*=0, in data from the World Ocean Atlas (green, Garcia et al., 2014) and in the *Baseline* simulation of the 5-PFT setup (light blue) and the 4-PFT setup (black, Nissen et al., 2018) of ROMS-BEC, respectively. b) zonal average Si* [mmol m$^{-3}$], colors are the same as the contours in panel a).

[Figure]

**Figure** S9: a) Same as Fig. 3 in the main text, Hovmoller plots south of 50° S of the day of maximum total chlorophyll concentrations in a satellite product (black line, Globcolor climatology from 1998-2018 based on the daily 25 km chlorophyll product, see Fanton d'Andon et al., 2009; Maritorena et al., 2010), the *Baseline* simulation of this study (solid blue line), the *Baseline* simulation of Nissen et al. (2018, dashed blue line; without *Phaeocystis*). Additionally, two sensitivity simulations in the 4 PFT setup from Nissen et al. (2018) are shown here to show the impact of biases in the simulated physical fields on phytoplankton phenology: The simulations TEMP (dashed red line) and MLD (dashed green line) correct for the simulated average temperature and MLD biases, respectively, within the biological subroutine of the model. b) Difference in day of bloom peak between *Phaeocystis* and diatoms, based on chlorophyll concentrations in the 5-PFT *Baseline* simulation. Stippling indicates locations where maximum chlorophyll concentrations never exceed 0.1 mg chl m$^{-3}$ for *Phaeocystis* (orange) and diatoms (green), respectively. White areas correspond to areas where the peak total chlorophyll concentrations do not exceed 0.5 mg chl m$^{-3}$.

[Figure]

**Figure** S10:  Diatom (red) and *Phaeocystis* (blue) surface carbon biomass concentrations mmol C m$^{-3}$~~) south of 40° S as a function of the simulated temperature (° C) and a) nitrate concentrations (mmol N m$^{-3}$) and b) mixed layer PAR levels (W m$^{-2}$) . Overlain are the observed ecological niche centers (median)and breadths (inter quartile ranges) for example taxa from Brun et al. (2015, circles and solid lines) and as simulated in ROMS-BEC (triangles and dashed lines; area and biomass weighted) . The red bars on the axes indicate the simulated range of the respective environmental condition in ROMS-BEC betweenS and averaged over DJFM and the top 50 m.~~ S and those on the right for the Ross Sea. Light blue area indicate times of the year when *Phaeocystis* biomass is larger than diatom biomass.

[Figure]

**Figure S9:** **Figure S11:** Carbon cycling in the Ross Sea: a) Pathways of particulate organic carbon (POC) formation in the *Baseline* simulation of ROMS-BEC averaged annually over the Ross Sea. The green and yellow boxes show the relative contribution (%) of *Phaeocystis*, diatoms, coccolithophores, small phytoplankton (SP), and zooplankton (Zoo) to the combined phytoplankton and zooplankton biomass (green) and total POC production (yellow) in the top 100 m, respectively. The arrows denote the relative contribution of the different POC production pathways associated with each PFT (black = grazing by zooplankton, grey = aggregation, blue = non-grazing mortality), given as % of total NPP in the top 100 m. Numbers are printed if ≥0.1% and rounded to the nearest integer if >1%. The sum of all arrows gives the POC production efficiency, i.e., the fraction of NPP which is converted into sinking POC upon biomass loss (p ratio). Note that diazotrophs are not included in this figure due to their minor contribution to NPP in the model domain. b)-d) Simulated vertically integrated production of particulate organic carbon (POC) b) as a function of  time [mmol C m$^{-2}$ d$^{-1}$], c) cumulative over time (absolute production in Pg C yr$^{-1}$ on the left axis and relative to annually integrated production on the right axis), and d) as a function of time via grazing and aggregation, respectively. The colors correspond to the different PFTs in ROMS-BEC, and the panels correspond to averages or integrals over the Ross Sea.

[Figure]

**Figure S12:** Results from the simulation VARYING_kFE (see section 2.2 in the main text): Varying half-saturation constant of iron of *Phaeocystis* (k_Fe, red, left y axis) and PAR (yellow, right y axis) as a function of time (x axis) for the surface (solid) and averaged over the top 50 m (dashed) for a) between 60-90° S and b) in the Ross Sea. Black lines indicate the constant k_Fe of *Phaeocystis* (dashed) and diatoms (dotted) used in the *Baseline* simulation of this study. c) Difference in days in the timing of the bloom peak of diatoms and *Phaeocystis* for each latitude, with negative values denoting a succession from *Phaeocystis* to diatoms throughout the season. d) Difference in day of bloom peak between *Phaeocystis* and diatoms. Stippling indicates locations where maximum chlorophyll concentrations never exceed 0.1 mg chl m$^{-3}$ for *Phaeocystis* (orange) and diatoms (green), respectively. White areas correspond to areas where the peak total chlorophyll concentrations do not exceed 0.5 mg chl m$^{-3}$.

**S2: Parameter sensitivity experiments**

**Table S1:** Overview of parameter sensitivity simulations, varying the respective parameter by $\pm 50\%$. PA=*Phaeocystis*, D=diatoms. See also Table 1 & Table 2 in the main text.

| Run Name  | Description | |
|---|---|---|
| Topt150 | Increase $T_{\mathrm{opt}}^{\mathrm{PA}}$ by 50% | ⎫ Param_Topt |
| Topt50 | Decrease $T_{\mathrm{opt}}^{\mathrm{PA}}$ by 50% | ⎭ |
| kFe150 | Increase $k_{\mathrm{Fe}}^{\mathrm{PA}}$ by 50% | ⎫ Param_kFe |
| kFe50 | Decrease $k_{\mathrm{Fe}}^{\mathrm{PA}}$ by 50% | ⎭ |
| alphaPI150 | Increase $\alpha_{\mathrm{PI}}^{\mathrm{PA}}$ by 50% | ⎫ Param_alphaPI |
| alphaPI50 | Decrease $\alpha_{\mathrm{PI}}^{\mathrm{PA}}$ by 50% | ⎭ |
| mortality150 | Increase $\gamma_{\mathrm{m,0}}^{\mathrm{PA}}$ by 50% | ⎫ Param_mortality |
| mortality50 | Decrease $\gamma_{\mathrm{m,0}}^{\mathrm{PA}}$ by 50% | ⎭ |
| aggregation150 | Increase $\gamma_{\mathrm{a,0}}^{\mathrm{PA}}$ by 50% | ⎫ Param_aggregation |
| aggregation50 | Decrease $\gamma_{\mathrm{a,0}}^{\mathrm{PA}}$ by 50% | ⎭ |
| grazing150 | Increase $\gamma_{\mathrm{g,max}}^{\mathrm{PA}}$ by 50% | ⎫ Param_grazing |
| grazing50 | Decrease $\gamma_{\mathrm{g,max}}^{\mathrm{PA}}$ by 50% | ⎭ |
| thetaNmax50 | Increase $\theta_{\mathrm{chl:N,max}}^{\mathrm{PA}}$ by 50% | ⎫ Param_thetaNmax |
| thetaNmax50 | Decrease $\theta_{\mathrm{chl:N,max}}^{\mathrm{PA}}$ by 50% | ⎭ |

In order to more systematically quantify the sensitivity of simulated distributions of *Phaeocystis* and diatoms and integrated estimates of NPP and POC export in ROMS-BEC to *Phaeocystis* model parameter choices, we have performed a set of model parameter sensitivity experiments. To that aim, we have systematically increased/decreased all key *Phaeocystis* parameters by 50%, allowing for an objective ranking of model sensitivities. We varied the following seven parameters of *Phaeocystis*, resulting in a total of 14 simulations: the temperature optimum, the half-saturation constant of iron, $\alpha_{\mathrm{PI}}$, the maximum chl:N ratio $\theta_{\mathrm{chl:N,max}}$, the linear mortality rate, the quadratic mortality rate (aggregation), and the maximum grazing rate of zooplankton on *Phaeocystis*  see Table S1).

We then quantify the sensitivity $S$ of any target variable $A$ (here $A$ being one of the following targets: total phytoplankton, *Phaeocystis*, and diatom chlorophyll concentrations, total NPP, and POC export across 100 m) to changes in the parameter $X$ as follows, allowing for a ranking of the seven sets of simulations by the magnitude of the sensitivity (see Table S1):

$$S_{\mathrm{X}}^{A} = 100 \cdot \frac{A_{\mathrm{X150}} - A_{\mathrm{X50}}}{A_{\mathrm{XBaseline}}} \tag{1}$$

As expected (see also Nissen et al., 2018), we find that both total chlorophyll concentrations and chlorophyll levels of *Phaeocystis* and diatoms are highly sensitive to parameters describing the growth and loss of *Phaeocystis* biomass, with increases of up to 700% (grazing50) and declines of up to >90% (Topt50, thetaNmax50) in *Phaeocystis* biomass between 60-90° S for a 50% change in the associated parameters (see Fig. S13). In general, any decline/increase in *Phaeocystis* chlorophyll biomass is associated with an increase/decline in diatom chlorophyll biomass, pointing to the direct competition for resources of these two phytoplankton types at high SO latitudes. Yet, the biomass compensation is not always complete due to non-linearities in the model system (e.g. food web feedbacks), resulting in changes of up to 70% (grazing150) in total chlorophyll levels upon changes in *Phaeocystis* parameters. The ranking of model sensitivities between 60-90° S reveals the highest sensitivity of *Phaeocystis*

[Figure]

and diatom chlorophyll concentrations to the maximum grazing rate $\gamma_{\mathrm{g,max}}^{\mathrm{PA}}$, the maximum chl:N ratio $\theta_{\mathrm{chl:N,\ max}}^{\mathrm{PA}}$, the initial slope of the photosynthesis-irradiance curve ($\alpha_{\mathrm{PI}}^{\mathrm{PA}}$), and the temperature optimum $T_{\mathrm{opt}}$ of *Phaeocystis* growth (Param_grazing, Param_thetaNmax, Param_alphaPI, Param_Topt in Table S1 & S2). In comparison, the opposed changes in *Phaeocystis* and diatom chloro- **Figure S13:** Annual mean surface chlorophyll concentrations of all phytoplankton (*total Chl*), *Phaeocystis* (*PA*), and diatoms (*D*) in the parameter sensitivity simulations (see Table S1) relative to the *Baseline* simulation. The model output is averaged over a) 60-90° S and  b) the Ross Sea.

phyll levels (see Fig. S13) result in lower sensitivities of total chlorophyll levels to changes in *Phaeocystis*  parameters in general and a lower ranking of the temperature optimum and thetaNmax experiments in particular (Param_Topt and Param_thetaNmax in Table S2).

In comparison to the ranking of model experiments for total chlorophyll, the model sensitivities for NPP and POC export across 100 m are similar in magnitude both between 60-90° S and in the Ross Sea (20-90%, compare Table S2 & Table S3). Additionally, the ranking of model experiments for NPP and POC export reveals only small differences to the ranking of model sensitivities for total chlorophyll: While the experiments Param_alphaPI and Param_grazing consistently rank amongst the top two most sensitive experiments for NPP and POC export and between 60-90° S for total chlorophyll concentrations, the experiments Param_mortality/Param_Topt are less/more important for NPP and POC than for total chlorophyll levels in ROMS-BEC (compare Table S2 & S3). In summary, this demonstrates the large model sensitivity of bulk biogeochemical quantities to parameter choices describing the temperature and light dependence of *Phaeocystis* growth and zooplankton grazing.

**Table S2:** Ranking of the parameter sensitivity experiments by the absolute sensitivity of annual mean total surface chlorophyll ($|S_X^{\text{Chl}}|$), *Phaeocystis* chlorophyll ($|S_X^{\text{Chl}^{\text{PA}}}|$), and diatom chlorophyll ($|S_X^{\text{Chl}^{\text{D}}}|$) to a $\pm 50\%$ change in the model parameter $X$ relative to the *Baseline* setup of ROMS-BEC between 60-90°S and in the Ross Sea, respectively. The sensitivity $S$ (%) is quantified using Eq. 1. See Table S1 for details on the experimental setup and Fig. S13 for details on the resulting chlorophyll fields in ROMS-BEC in each experiment. Note that the simulated changes in carbon biomass fields are qualitatively similar to those of chlorophyll (not shown) and that the ranking shown here is therefore insensitive to the choice of chlorophyll in the analysis.

|  | Ranking ($|\mathbf{S}_X^{\text{Chl}}|$ in %) | Ranking ($|\mathbf{S}_X^{\text{Chl}^{\text{PA}}}|$ in %) | Ranking ($|\mathbf{S}_X^{\text{Chl}^{\text{D}}}|$ in %) |
|---|---|---|---|
| **60-90°S** | | | |
| | 1. Param_alphaPI (63.6) | 1. Param_grazing (693.1) | 1. Param_alphaPI (153.4) |
| | 2. Param_grazing (48.3) | 2. Param_thetaNmax (390.9) | 2. Param_thetaNmax (149.6) |
| | 3. Param_mortality (40.6) | 3. Param_Topt (306.8) | 3. Param_Topt (132.7) |
| | 4. Param_kFe (39.8) | 4. Param_alphaPI (259.4) | 4. Param_grazing (128.3) |
| | 5. Param_Topt (37.5) | 5. Param_kFe (209.1) | 5. Param_kFe (109.6) |
| | 6. Param_thetaNmax (33.0) | 6. Param_mortality (178.0) | 6. Param_mortality (101.8) |
| | 7. Param_aggregation (6.4) | 7. Param_aggregation (65.1) | 7. Param_aggregation (10.2) |
| **Ross Sea** | | | |
| | 1. Param_alphaPI (76.3) | 1. Param_grazing (360.3) | 1. Param_thetaNmax (189.1) |
| | 2. Param_mortality (53.3) | 2. Param_thetaNmax (288.9) | 2. Param_alphaPI (189.1) |
| | 3. Param_thetaNmax (46.4) | 3. Param_Topt (194.2) | 3. Param_Topt (142.1) |
| | 4. Param_Topt (41.6) | 4. Param_alphaPI (188.3) | 4. Param_grazing (129.8) |
| | 5. Param_kFe (41.3) | 5. Param_kFe (126.2) | 5. Param_mortality (126.7) |
| | 6. Param_grazing (19.2) | 6. Param_mortality (114.8) | 6. Param_kFe (114.3) |
| | 7. Param_aggregation (12.3) | 7. Param_aggregation (59.5) | 7. Param_aggregation (9.0) |

**Table S3:** Ranking of the parameter sensitivity experiments by the absolute sensitivity of annually integrated NPP ($|S_X^{\mathrm{NPP}}|$) and POC export across 100 m ($|S_X^{\mathrm{POC_{100m}}}|$) to a $\pm50\%$ change in the model parameter $X$ relative to the *Baseline* setup of ROMS-BEC between 60-90°S and in the Ross Sea, respectively. The sensitivity $S$ (%) is quantified using Eq. 1. See Table S1 for the experimental setup.

| | **Ranking ($\|S_X^{\mathrm{NPP}}\|$ in %)** | **Ranking ($\|S_X^{\mathrm{POC_{100m}}}\|$ in %)** |
|---|---|---|
| **60-90°S** | | |
| | 1. Param_grazing (68.4) | 1. Param_grazing (86.4) |
| | 2. Param_alphaPI (46.7) | 2. Param_alphaPI (35.4) |
| | 3. Param_Topt (43.6) | 3. Param_Topt (26.7) |
| | 4. Param_kFe (23.6) | 4. Param_mortality (12.9) |
| | 5. Param_thetaNmax (23.4) | 5. Param_kFe (11.6) |
| | 6. Param_mortality (11.6) | 6. Param_thetaNmax (10.7) |
| | 7. Param_aggregation (7.6) | 7. Param_aggregation (1.4) |
| | | |
| **Ross Sea** | | |
| | 1. Param_grazing (55.6) | 1. Param_grazing (71.9) |
| | 2. Param_alphaPI (48.5) | 2. Param_alphaPI (39.0) |
| | 3. Param_Topt (44.0) | 3. Param_Topt (26.9) |
| | 4. Param_thetaNmax (24.7) | 4. Param_thetaNmax (11.9) |
| | 5. Param_kFe (20.4) | 5. Param_kFe (10.5) |
| | 6. Param_aggregation (11.6) | 6. Param_mortality (10.2) |
| | 7. Param_mortality (8.3) | 7. Param_aggregation (2.6) |

**References**

[revised manuscript text omitted]